# MAKING PRE-TRAINED LANGUAGE MODELS GREAT ON TABULAR PREDICTION

**Jiahuan Yan**[1,2,*], **Bo Zheng**[2], **Hongxia Xu**[2], **Yiheng Zhu**[2], **Danny Z. Chen**[3], **Jimeng Sun**[4], **Jian Wu**[1,2,†], **Jintai Chen**[4,†]
[1]The Second Affiliated Hospital Zhejiang University School of Medicine [2]Zhejiang University
[3]University of Notre Dame [4]University of Illinois at Urbana-Champaign
{jyansir,zjuzhengbo,einstein,zhuyiheng2020,wujian2000}@zju.edu.cn, dchen@nd.edu, jimeng@illinois.edu, jtchen721@gmail.com

## ABSTRACT

The transferability of deep neural networks (DNNs) has made significant progress in image and language processing. However, due to the heterogeneity among tables, such DNN bonus is still far from being well exploited on tabular data prediction (e.g., regression or classification tasks). Condensing knowledge from diverse domains, language models (LMs) possess the capability to comprehend feature names from various tables, potentially serving as versatile learners in transferring knowledge across distinct tables and diverse prediction tasks, but their discrete text representation space is inherently incompatible with numerical feature values in tables. In this paper, we present *TP-BERTa*, a specifically pre-trained LM for tabular data prediction. Concretely, a novel *relative magnitude tokenization* converts scalar numerical feature values to finely discrete, high-dimensional tokens, and an *intra-feature attention* approach integrates feature values with the corresponding feature names. Comprehensive experiments demonstrate that our pre-trained TP-BERTa leads the performance among tabular DNNs and is competitive with Gradient Boosted Decision Tree models in typical tabular data regime.

## 1 INTRODUCTION

Tabular data, a common data form, is pivotal in various fields such as medical trial predictions (Hassan et al., 2020) and financial risk detection (Aziz et al., 2022). The remarkable successes of deep neural networks (DNNs) in computer vision (CV) and natural language processing (NLP) have spurred interest in applying DNNs to tabular data for tasks like classification or regression (Popov et al., 2020; Song et al., 2019; Wang et al., 2021; Chen et al., 2023b), which also pave the road for cross-modality processing. However, most current research on tabular data relies on fully supervised paradigms (Arik & Pfister, 2021; Gorishniy et al., 2021; Somepalli et al., 2022; Li et al., 2023; Hollmann et al., 2023), and with typically limited data available for DNNs in this regime, Gradient Boosted Decision Trees (GBDTs) (Chen & Guestrin, 2016; Ke et al., 2017; Prokhorenkova et al., 2018) continue to outperform these paradigms (Grinsztajn et al., 2022).

As widely evidenced in the CV and NLP fields, the transferability of DNNs consistently brought about substantial performance boosts and decreased data demands in downstream tasks (Devlin et al., 2018; Xie et al., 2020; He et al., 2020). However, how to utilize the transferability of DNNs on tabular data is still much under-explored. One major obstacle is the feature heterogeneity among tables (Borisov et al., 2022; Yan et al., 2023; Chen et al., 2022). Unlike images, which often exhibit similar feature distributions (e.g., consistent pixel intensity ranges and color distributions) (Chen et al., 2023a), structured tables inherently contain diverse columns and feature spaces, leading to considerable heterogeneity and feature space shifts between pre-training and downstream datasets.

**Related Work.** Recent studies highlight the importance of tabular transfer learning, with initial efforts like TransTab (Wang & Sun, 2022) and XTab (Zhu et al., 2023) utilizing shared Transformer

[*]Work under partial support from the Second Affiliated Hospital Zhejiang University School of Medicine.
[†]Corresponding authors. Codes are available at https://github.com/jyansir/tp-berta.

blocks in the FT-Transformer architecture (Gorishniy et al., 2021) for cross-table learning. TransTab focused on clinical trial tables with common feature names, facilitating partially overlapped feature embeddings, whereas XTab explored a broader domain with dataset-specific encoders. However, neither achieved comprehensive knowledge transfer, resulting in moderate pre-training performance. The advancements in language models (LMs) have demonstrated their capability to act as commonsense knowledge bases (Petroni et al., 2019; Jiang et al., 2020; Gao et al., 2021; Zha et al., 2023). Through self-supervised pre-training on extensive domain-agnostic corpora, LMs can implicitly capture associations among different words or phrases, showcasing potential as tabular transfer agents with their inherent support for feature name processing within a unified language space. Despite this potential, early attempts of applying LMs to tabular prediction were limited to synthetic table generation (e.g., missing value imputation) and faced challenges. GReaT (Borisov et al., 2023) and TapTap (Zhang et al., 2023) fine-tuned GPT-2 (Radford et al., 2019) on simply templated table texts, treating numerical values as strings, which led to insensitivity to such values (Qian et al., 2023). A contemporary work (Ye et al., 2024) developed a BERT-based model (CT-BERT) using a large tabular database and similar techniques to TransTab. However, these studies overlooked the customization of LMs for understanding continuous numerical values, which is a critical aspect of tables and presents challenges to LMs due to their inherent complexity and rarity (Qian et al., 2023).

To unlock LMs' power and take a pioneering step on LM-based tabular transfer learning, in this paper, we propose a tailored pre-trained LM for tabular prediction based on RoBERTa (Liu et al., 2019), called the ***T**abular **P**rediction adapted **BERT** **a**pproach (TP-BERTa)*. TP-BERTa maintains the strengths of LMs as well as possessing the sensitivity to numeric features. Specifically, TP-BERTa discretizes numerical feature values as relative magnitude tokens (RMT) in order to treat them as some meaningful words in the LM's vocabulary. The design of relative magnitude tokens enables the LM to perceive relative value magnitudes in the language space. In this way, we decouple the representations of feature names and numerical values (compared to FT-Transformer, TransTab, and CT-BERT), preserving the semantic signal of feature names. Further, we develop a shared intra-feature attention (IFA) module to attentively fuse the embeddings of a feature's name and value into a single vector. IFA retains the text order in a feature name, and outputs a vector for each feature name-value pair to the subsequent LM process to achieve feature order-agnostic prediction.

We pre-train TP-BERTa on numerous large tabular datasets (101 binary classification and 101 regression datasets), and provide three versions (i.e., pre-trained on only classification tasks, or only regression tasks, or both). We conduct evaluations on extensive downstream datasets: (1) performance comparison with classical GBDTs, advanced deep tabular models, and cross-table models shows that our TP-BERTa (the pre-trained versions on a single task type, with default hyperparameters) outperforms the other tabular DNNs and is competitive with GBDTs in the overall rank on 145 downstream datasets; (2) comparison with two existing numerical encoding strategies (Borisov et al., 2023; Ye et al., 2024) shows that our RMT adaption achieves average AUC improvements of 12.45% and 3.44% on significantly changed (i.e., with AUC variation over 0.5%) downstream binary classification datasets, respectively; (3) ablation on table-specific designs for LM adaption.

**Contributions.** In a nutshell, our work offers: (1) **A pre-trained LM for tabular data:** dealing with fundamental difficulties in LM adaption to tabular data (i.e., numeric feature handling and tabular feature organization), we develop LM-based tabular DNNs and pre-train a tabular-data-tailored LM called TP-BERTa; (2) **superior performances:** comparisons with various existing methods on 145 downstream datasets demonstrate that pre-trained LMs can outperform common tabular DNNs and are competitive with GBDTs in typical tabular regime; (3) **in-depth analysis:** multi-facet comparison implies that TP-BERTa has a data appetite of informative discrete features, and key ablation experiments show that our RMT and IFA adaptions are successful.

## 2 TP-BERTa: Tabular Prediction adapted BERT approach

Our proposed TP-BERTa is built on the basis of RoBERTa (Liu et al., 2019) as default. Its model architecture and key components (the *relative magnitude tokenization* approach and *intra-feature attention* module) are shown in Fig. 1. Below we introduce our novel (i) relative magnitude tokenization (RMT) for numerical value representation, (ii) intra-feature attention (IFA) module for feature name-value matching before the LM processing, and (iii) the overall pre-training paradigm.

### 2.1 Relative Magnitude Tokenization

Tabular features can be roughly categorized into continuous type (i.e., numerical features) and discrete type (categorical, binary, or string features). Although discrete feature values with clear semantics (e.g., "male" and "female" are values of a discrete feature "gender") can be naturally understood by LMs, it is still difficult to make numerical features fully understandable to LMs due to their wide range of values and counter-intuitive meanings of exact numerical values. In this section, we present a novel Relative Magnitude Tokenization (RMT) approach to boost numerical value understanding.

**Numerical Discretization.** Our RMT process is inspired by classical works on feature binning (Dougherty et al., 1995; Gorishniy et al., 2022) that utilized discretization techniques for numerical features. To deal with diverse labeled tabular datasets, we adopt a target-aware binning method similar to (Gorishniy et al., 2022). Specifically, the "C4.5 Discretization" algorithm (Kohavi & Sahami, 1996) is applied to each numerical feature by recursively splitting its value range guided by its label. This process is equivalent to building a decision tree, and continuous values are grouped into corresponding tree leaves. The boundary values of all the leaves are used to split the value range into multiple bins. Each numerical value is converted to its bin index after discretization, as:

$$\boldsymbol{e}^{(i)} = \text{C4.5}(\boldsymbol{x}^{(i),\text{train}}, \boldsymbol{y}^{(i),\text{train}}), \tag{1}$$

$$\text{BinIndex}(x_j^{(i)}) \equiv k, \tag{2}$$

where $\boldsymbol{x}^{(i),\text{train}}$ is the vector of the $i$-th numerical feature values in the training set, $\boldsymbol{y}^{(i),\text{train}}$ is the corresponding labels, $\boldsymbol{e}^{(i)}$ denotes the vector of leaf node boundary values (in ascending order), $x_j^{(i)}$ is the $i$-th feature value of sample $j$, and $k$ is its bin index if $e_k^{(i)} \leq x_j^{(i)} < e_{k+1}^{(i)}$. In TP-BERTa, we set the maximum numerical bin (magnitude token) number (denoted as $n_{\text{bin}}$) to 256 (i.e., $0 \leq k < 256$), unless otherwise specified. A bin index represents a relative magnitude in the value range.

**Magnitude Tokenization.** To transform numerical values into the language space, we treat the numerical bins as new words. Specifically, $n_{\text{bin}}$ additional tokens are added to the RoBERTa vocabulary with randomly initialized token embeddings. Each numerical value is discretized with a feature-specific C4.5 process and mapped to these shared magnitude tokens. Since there may be a large number of values in a single numerical bin, the final token embedding of a numerical value is its corresponding bin token embedding multiplied with the value itself, i.e., $\text{RMT}(x_j^{(i)}) \equiv \boldsymbol{E}_{:,k}^{\text{extra}} \times x_j^{(i)}$, where $\boldsymbol{E}_{:,k}^{\text{extra}}$ denotes the $k$-th embedding of the RoBERTa additional vocabulary for the numerical magnitude. These embeddings are shared across any numerical features or datasets that purely represent relative magnitudes with word vectors. Just as LMs show general capability of language modeling based on reasonable pair-wise word similarity, we seek to make the designed "magnitude embeddings" follow a similar relationship. Hence, we devise a magnitude-aware triplet loss to regularize the learning process of the magnitude embeddings. We formulate the regularization as:

$$\begin{aligned} L_{\text{reg}} = \max(d(f(k_1), f(k_2)) - d(f(k_1), f(k_3)) + m(k_1, k_2, k_3), 0), \\ s.t. \mid k_1 - k_2 \mid < \mid k_1 - k_3 \mid, \end{aligned} \tag{3}$$

$$f(k) = \text{LayerNorm}(\text{Linear}(\boldsymbol{E}_{:,k}^{\text{extra}})), \tag{4}$$

$$m(k_1, k_2, k_3) = \frac{\mid k_1 - k_3 \mid - \mid k_1 - k_2 \mid}{n_{\text{bin}}}, \tag{5}$$

where $k_1, k_2$, and $k_3$ are three bin indices, and $d(\boldsymbol{x}, \boldsymbol{y})$ is the $L_2$ distance between vectors $\boldsymbol{x}$ and $\boldsymbol{y}$. In a nutshell, this regularization process assists to pull the embedding of a bin close to the embedding of a nearby one, while pushing away from the embedding of a bin far away from it, serving as an auxiliary loss to help embedding learning for magnitude tokens.

**Tabular Feature Pre-processing.** A tabular sample may contain features of different types. We process each feature $i$ by simply concatenating the embeddings of its feature name ($E_i^{\text{name}} \in \mathbb{R}^{l_1 \times d}$) and value ($E_i^{\text{value}} \in \mathbb{R}^{l_2 \times d}$), i.e., $E_i = E_i^{\text{name}} \otimes E_i^{\text{value}}$, where $d$ is the hidden dimension of the RoBERTa embeddings, $l_1$ is the token length of the feature name, and $l_2$ is the length of the feature value. Notably, $l_2 \equiv 1$ for numerical features. As for categorical features, we convert their values into structured texts (e.g., value "0" of the feature "gender" is mapped to "male"). Note that we do not distinguish binary and categorical ones in this paper since they are both converted to meaningful

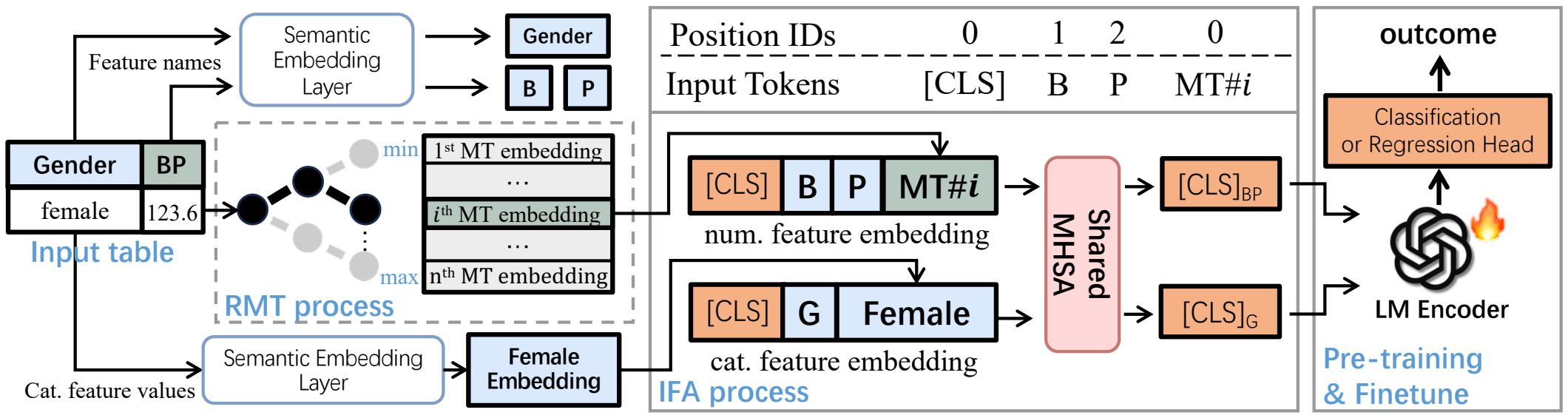

Figure 1: Illustrating the TP-BERTa workflow. "BP" in the input table denotes the feature name text "blood pressure". The rectangles with "B", "P", and "Gender" ("G") represent word embedding of "blood", "pressure", and "gender", respectively. In the RMT process, numerical values are discretized by the feature-specific C4.5 decision tree. In the IFA process, "MT#$i$" indicates the $i$-th magnitude token. All numerical features share these MT embeddings for magnitude representation. "MHSA" is a shared multi-head self-attention across all features for feature refinement.

texts. Some datasets contain string features, such as a feature "movie comment" with unstructured texts. We process the values of these feature types in the same way as for feature names.

### 2.2 Intra-Feature Attention Module

Previous attempts of using LMs to process tables still face three lingering issues. (1) Targets in tabular predictions are independent of feature permutations, while LMs inherently process texts with positional encoding since positions of linguistic units matter. (2) When we simply feed all feature values with names into a vanilla LM (e.g., "[Gender] is female, [Blood Pressure] is 123.8"), it likely increases the training difficulty of LMs since they have to understand the correctly matched name-value pairs of features and learn to alleviate interference from other features. However, fully connected attention mechanism (commonly adopted in auto-encoder LMs) makes it inevitable to generate mismatched name-value signal. (3) Feeding the whole templated text can incur computation burden caused by excessively long sequences when the feature amount is large. Recently, a solution was given for issue (1) by augmenting a sample with copies of different feature permutations (Borisov et al., 2023), and position encoding was directly dropped and text order of feature names was ignored (Ye et al., 2024). But, they all neglected issues (2) and (3). Hence, we develop the intra-feature attention (IFA) module for feature refinement before feeding features to the LM.

IFA is essentially a single multi-head self-attention (MHSA) module shared across all features and datasets. It accepts embeddings of a feature name-value pair and fuses them into a single vector. We formulate the process of IFA fusion on a single feature $i$ as:

$$\boldsymbol{H}^{(i)} = \boldsymbol{e}^{\text{CLS}} \otimes \boldsymbol{E}^{(i)}, \tag{6}$$

$$\boldsymbol{Q}^{(i)} = \boldsymbol{W}_{\text{q}}^{\text{T}}(\boldsymbol{H}^{(i)} + \boldsymbol{P}^{(i)}),\ \ \boldsymbol{K}^{(i)} = \boldsymbol{W}_{\text{k}}^{\text{T}}(\boldsymbol{H}^{(i)} + \boldsymbol{P}^{(i)}),\ \ \boldsymbol{V}^{(i)} = \boldsymbol{W}_{\text{v}}^{\text{T}}\boldsymbol{H}^{(i)}, \tag{7}$$

$$\hat{\boldsymbol{H}}^{(i)} = \text{MHSA}(\boldsymbol{Q}^{(i)}, \boldsymbol{K}^{(i)}, \boldsymbol{V}^{(i)}),\ \ \hat{\boldsymbol{h}}^{(i)} \equiv \hat{\boldsymbol{H}}^{(i)}_{:,\text{Index(CLS)}}, \tag{8}$$

where $\boldsymbol{E}^{(i)} \in \mathbb{R}^{(l_1+l_2)\times d}$ is concatenation of name-value embeddings, $\boldsymbol{e}^{\text{CLS}} \in \mathbb{R}^{1\times d}$ is the [CLS] embedding, $\boldsymbol{W}_{\text{q}}, \boldsymbol{W}_{\text{k}}$, and $\boldsymbol{W}_{\text{v}} \in \mathbb{R}^{d\times d}$ are transformations for query, key, and value vectors, and $\boldsymbol{P}^{(i)} \in \mathbb{R}^{(1+l_1+l_2)\times d}$ is position embeddings. IFA uses the output vector at the [CLS] position $\hat{\boldsymbol{h}}^{(i)}$ as refined feature information and feeds it to the subsequent RoBERTa. It can be clearly seen that information from both the name and value is included in $\hat{\boldsymbol{h}}^{(i)}$, and information from other feature names or values cannot corrupt feature $i$'s representation. As shown in Fig. 1(IFA process), the positions of the [CLS] token and magnitude token are assigned to id 0, and those of feature names are from 1 to $l_1$. This design aims to make the [CLS] token pay more attention to values (which are probably more important for prediction) as well as keeping the text order of feature names. Notably, we remove position encoding on value vectors (see Eq. (7)); the key reason for this is to protect magnitude token embeddings from the impact of embeddings at a constant id position (e.g., position id 0). Since magnitude embeddings are randomly initialized and intentionally regularized to represent the meaning of the relative magnitude carefully, a constant signal may distort the representations and thus make the embedding learning process more difficult.

### 2.3 OVERALL TRAINING PARADIGM

After features are processed by the IFA module, an $n$-feature sample is organized as the concatenation of feature vectors and a [CLS] embedding to be the RoBERTa input, i.e., $\boldsymbol{X} \equiv \boldsymbol{e}^{\text{CLS}} \otimes \hat{\boldsymbol{h}}_1 \otimes \hat{\boldsymbol{h}}_2 \otimes \cdots \otimes \hat{\boldsymbol{h}}_n \in \mathbb{R}^{(1+n)\times d}$, which is computation-friendly. Since the text order of feature names has been considered in $\hat{h}_i$, we can avoid position encoding in this step, and achieve feature order-agnostic prediction. The prediction is based on the [CLS] output of the *RoBERTa-Encoder*, as:

$$\hat{\boldsymbol{y}}_m = \text{PredictionHead}^{(m)}(\textit{RoBERTa-Encoder}(\boldsymbol{X}^{(m)})_{:,\text{Index(CLS)}}), \tag{9}$$

$$\text{PredictionHead}(\boldsymbol{x}) = \text{Dropout}(\text{Linear}_1(\text{Tanh}(\text{Linear}_2(\boldsymbol{x})))), \tag{10}$$

where $\boldsymbol{X}^{(m)}$ represents the input from the $m$-th task (dataset), and we use task-specific prediction heads PredictionHead$^{(m)}$, the shared RoBERTa, and the IFA module (constituting TP-BERTa) to perform supervised pre-training on extensively large tabular datasets. The final pre-training loss consists of supervised loss and regularization loss (see Eq. (3)), as:

$$L = L_{\text{sup}} + \lambda L_{\text{reg}}, \tag{11}$$

where for the supervised loss $L_{\text{sup}}$, we use binary cross entropy loss for binary classification tasks and mean squared error loss for regression tasks. We keep a constant weight $\lambda \equiv 0.1$ in pre-training. For downstream tasks, ordinary finetune is adopted only with $L_{\text{sup}}$. We exclude multi-class datasets in this work as in (Grinsztajn et al., 2022), for the reasons: (1) they can be decomposed into multiple binary classification tasks, (2) the trends on binary classification can essentially reflect the classification ability, and (3) multi-class datasets are not very common in tabular dataset collections.

## 3 EXPERIMENTS

We first compare our TP-BERTa with classical and advanced tabular prediction models, including (1) the dominating GBDTs, (2) advanced deep tabular models, and (3) recent open-source cross-table models or pre-trained tabular models. We utilize extensive downstream datasets, and analyze the huge potential of our pre-trained LM, TP-BERTa, as a powerful tabular prediction learner from the data perspective (Sec. 3.2). Based on that, we further demonstrate how the encoding strategy of numerical values impacts the LMs' performances, and discuss why they were neglected in previous tabular prediction research (Sec. 3.3). Transferability evaluations (Sec. 3.4) and design ablations (Sec. 3.5) are conducted to reflect the generalization capability and rational adaption of TP-BERTa.

### 3.1 EXPERIMENTAL DETAILS

**Datasets.** We leverage the high-quality large semantic tabular database TabPertNet (Ye et al., 2024). Datasets with at least 10,000 samples and no more than 32 features are taken for pre-training, and datasets with fewer than 10,000 samples are collected as downstream tasks (following the same "typical tabular data" settings of "medium-sized dataset regime" and "not high dimensional" in (Grinsztajn et al., 2022)). We strictly remove the same datasets in the database and make sure that no subset of pre-training datasets (e.g., a small version of a large dataset) appears in the downstream ones. Since LMs are fueled by meaningful texts, we manually exclude datasets with uninformative feature names (e.g., feature names like "v1, v2, x1, x2") or unmappable categorical features (e.g., a feature "job" with values "0, 1, 2"). Note that those excluded datasets can still benefit from our model with simple feature preprocessing with their corresponding data dictionaries. In total, our pre-training datasets consist of 101 binary classification datasets and 101 regression datasets with about 10 million samples, and our downstream datasets consist of 80 binary classification datasets and 65 regression datasets. Detailed dataset statistics are provided in Appendix B.

**Pre-training Details.** Since our work does not focus on the curriculum learning issue, we warp all the datasets into a large data-loader, which provides a data batch from a randomly selected dataset per training step. Each dataset is learned with a dataset-specific prediction head and the shared TP-BERTa (see Sec. 2.3). Because the massive LM is likely to overfit a single dataset, we use 5% of the training data as the validation set. For binary classification, we keep the same label distributions for the training set and validation set. Pre-training is conducted on four NVIDIA A100 Tensor Core GPUs, with a total batch size of 512 per step. We reuse the weights of the RoBERTa-base as the

Table 1: The average values (standard deviations) of all method ranks on the dataset collections of two task types. "(d)" in the "Baselines" means using default hyperparameters, and "(t)" for using tuned ones. "Ours$_j$" is TP-BERTa pre-trained on both binary classification and regression tasks, and "Ours$_s$" contains two models pre-trained on the corresponding single-type tasks separately. "All" denotes rank information calculated on all the datasets, $\alpha$ is the amount ratio of categorical features and numerical ones in a dataset, and $\beta$ is the ratio of the Shapley value sums between the two feature types. $\alpha$ or $\beta$ provides a reference on the dominating feature type in tabular data: "$\alpha \geq 1$" represents that only the datasets with their $\alpha \geq 1$ are considered (similar denotations are for the others). The top performances are marked in **bold**, and the second best ones are underlined. We present feature type distribution statistics, $\alpha$ and $\beta$ formulation, and the original performances in Appendix B.

| Baselines | 80 downstream binary classification tasks | | | | | | 65 downstream regression tasks | | | | | |
|---|---|---|---|---|---|---|---|---|---|---|---|---|
| | All | $\alpha > 0$ | $\alpha \geq 1$ | $\alpha = 0$ | $\beta > 0$ | $\beta > 0.5$ | All | $\alpha > 0$ | $\alpha \geq 1$ | $\alpha = 0$ | $\beta > 0$ | $\beta > 0.5$ |
| XGBoost(d) | 7.7(4.0) | 7.8(4.1) | 9.2(4.0) | 6.8(3.5) | 8.2(4.1) | 8.3(3.9) | 7.7(4.4) | 7.7(4.6) | 7.3(4.1) | 7.8(4.0) | 8.0(4.7) | 9.2(4.3) |
| CatBoost(d) | 6.7(4.1) | 6.8(4.0) | 7.4(4.0) | 6.0(4.6) | 7.0(4.1) | 6.8(4.2) | 5.5(2.7) | 5.5(2.6) | 5.5(2.7) | 5.6(3.0) | 5.5(2.7) | 5.8(3.2) |
| FTT(d) | 7.1(3.5) | 7.0(3.5) | 6.6(3.5) | 6.9(3.6) | 6.9(3.6) | 7.2(3.6) | 7.8(2.7) | 7.8(2.5) | 8.2(3.0) | 7.6(3.2) | 8.0(2.6) | 8.3(1.3) |
| TransTab(d) | 11.0(4.5) | 11.2(4.5) | 11.2(4.1) | 10.2(4.6) | 11.6(4.3) | 11.7(4.2) | 12.1(4.0) | 12.1(3.8) | 13.3(2.2) | 12.4(4.5) | 12.0(4.0) | 13.6(1.2) |
| XGBoost(t) | 6.2(4.1) | 6.3(4.1) | 6.5(4.3) | 5.9(4.2) | 6.5(4.2) | 6.7(4.5) | 4.5(3.7) | 4.3(3.8) | **3.3(3.3)** | 5.0(3.5) | 4.7(3.9) | 4.1(3.2) |
| CatBoost(t) | 5.9(3.8) | 6.3(3.9) | 7.1(4.1) | **4.9(3.1)** | 6.4(3.9) | 6.4(4.1) | 5.5(3.6) | 5.7(3.6) | 5.8(3.5) | 4.9(3.7) | 5.7(3.7) | 6.1(3.8) |
| MLP(t) | 8.6(4.0) | 8.9(3.9) | 8.7(4.1) | 8.5(4.1) | 8.5(3.9) | 8.3(4.1) | 8.5(3.6) | 8.8(3.4) | 9.3(3.2) | 7.6(4.1) | 9.0(3.4) | 7.5(3.8) |
| AutoInt(t) | 8.0(3.5) | 7.8(3.3) | 7.4(3.4) | 8.6(4.0) | 7.7(3.4) | 7.7(3.2) | 8.3(3.0) | 8.6(3.0) | 8.5(2.7) | 7.4(3.1) | 8.3(3.0) | 8.2(3.2) |
| DCNv2(t) | 7.9(3.9) | 8.0(3.9) | 8.4(3.8) | 7.9(4.0) | 7.7(3.9) | 8.8(3.3) | 8.4(3.4) | 8.4(3.5) | 8.5(3.1) | 8.5(3.2) | 8.4(3.5) | 7.2(3.5) |
| TabNet(t) | 12.1(3.5) | 12.4(3.3) | 12.7(2.7) | 11.5(4.2) | 12.3(3.4) | 12.3(3.8) | 12.6(3.6) | 13.2(2.6) | 13.1(2.4) | 10.5(5.1) | 13.5(1.9) | 14.1(1.4) |
| SAINT(t) | 8.2(3.8) | 8.0(3.7) | 8.1(4.1) | 8.7(4.2) | 7.9(3.8) | 7.5(3.9) | 7.6(3.8) | 7.3(3.9) | 7.7(3.3) | 8.4(3.7) | 6.6(3.6) | 7.2(3.0) |
| FTT(t) | 6.8(3.5) | 6.8(3.6) | 6.5(3.4) | 6.2(3.3) | 6.9(3.6) | 6.9(3.9) | 7.9(3.4) | 7.6(3.3) | 7.7(3.1) | 9.0(3.4) | 7.2(3.0) | 6.8(3.2) |
| XTab(t) | 9.8(4.0) | 9.7(4.0) | 8.9(3.8) | 10.5(4.1) | 9.4(4.0) | 9.9(3.7) | 12.4(2.8) | 12.5(2.8) | 13.3(1.6) | 12.0(3.0) | 12.4(2.9) | 13.1(1.8) |
| Ours$_j$(d) | 8.4(4.5) | 7.7(4.5) | 7.0(5.0) | 9.9(4.1) | 7.9(4.6) | 7.0(4.7) | 6.9(4.6) | 6.3(4.4) | 4.8(3.9) | 8.5(5.0) | 6.5(4.5) | 5.2(3.9) |
| Ours$_s$(d) | **5.8(4.0)** | **5.1(3.9)** | **4.4(3.3)** | 7.5(3.7) | **5.2(4.1)** | **4.5(3.4)** | **4.3(2.8)** | **4.1(2.6)** | 3.9(2.4) | **4.8(3.4)** | **4.3(2.7)** | **3.6(2.8)** |

starting point, and follow similar pre-training settings of RoBERTa (Liu et al., 2019): We use a total of 30 training epochs, with a linear warm-up for the first 6% of steps, followed by a linear decay to 0. The best checkpoint is saved by the average validation loss over all the datasets. We provide three TP-BERTa versions: pre-trained on only binary classification tasks, or only regression tasks, or both types. More detailed pre-training information and analysis are given in Appendix D.

**Compared Methods.** We compare our TP-BERTa with (1) the representative non-deep learning models XGBoost (Chen & Guestrin, 2016) and CatBoost (Prokhorenkova et al., 2018); (2) known DNNs including MLP, TabNet (Arik & Pfister, 2021), AutoInt (Song et al., 2019), DCNv2 (Wang et al., 2021), FT-Transformer (FTT) (Gorishniy et al., 2021), and SAINT (Somepalli et al., 2022); (3) the recent open-source cross-table model TransTab (Wang & Sun, 2022) and pre-trained model XTab (Zhu et al., 2023). We split each finetune dataset ((64%, 16%, 20%) for training, validation, and testing separately), and keep the same label distribution in each split on binary classification.

**Hyperparameter Tuning & Finetune.** We implement our TP-BERTa with PyTorch and the HuggingFace Transformers package on Python 3.8. All the models are finetuned on NVIDIA RTX 3090. In training, we uniformly use a training batch size of 64 for all the DNNs. Since the LM takes an increased training time, we directly set fixed hyperparameters on the pre-trained TP-BERTa across all the downstream datasets without tuning. For the other DNNs, the optimizer is AdamW (Loshchilov & Hutter, 2019) with the default configuration except for the learning rate and weight decay rate. We follow the hyperparameter spaces from the original work for SAINT. For TransTab, we use its default hyperparameters without cross-table pre-training because it originally required partially overlapped medical tables. For XTab, we follow its settings that report the score of the best pre-trained checkpoint on the validation set, with other hyperparameters kept fixed. For XGBoost, CatBoost, and the other DNNs, we follow the default (for GBDTs and FT-Transformer) and tuning settings provided in (Gorishniy et al., 2021). Hyperparameter search is performed with the Optuna library (Akiba et al., 2019). More detailed information of hyperparameters is provided in Appendix E.

### 3.2 Are Pre-trained TP-BERTa Great Tabular Prediction Learners?

**Overall Comparison.** Table 1 reports the means and standard deviations of model ranks on two dataset collections. As expected, a similar trend as shown in (Grinsztajn et al., 2022) is attained: GBDTs (i.e., XGBoost and CatBoost) still outperform classical and advanced DNNs in typical tabu-

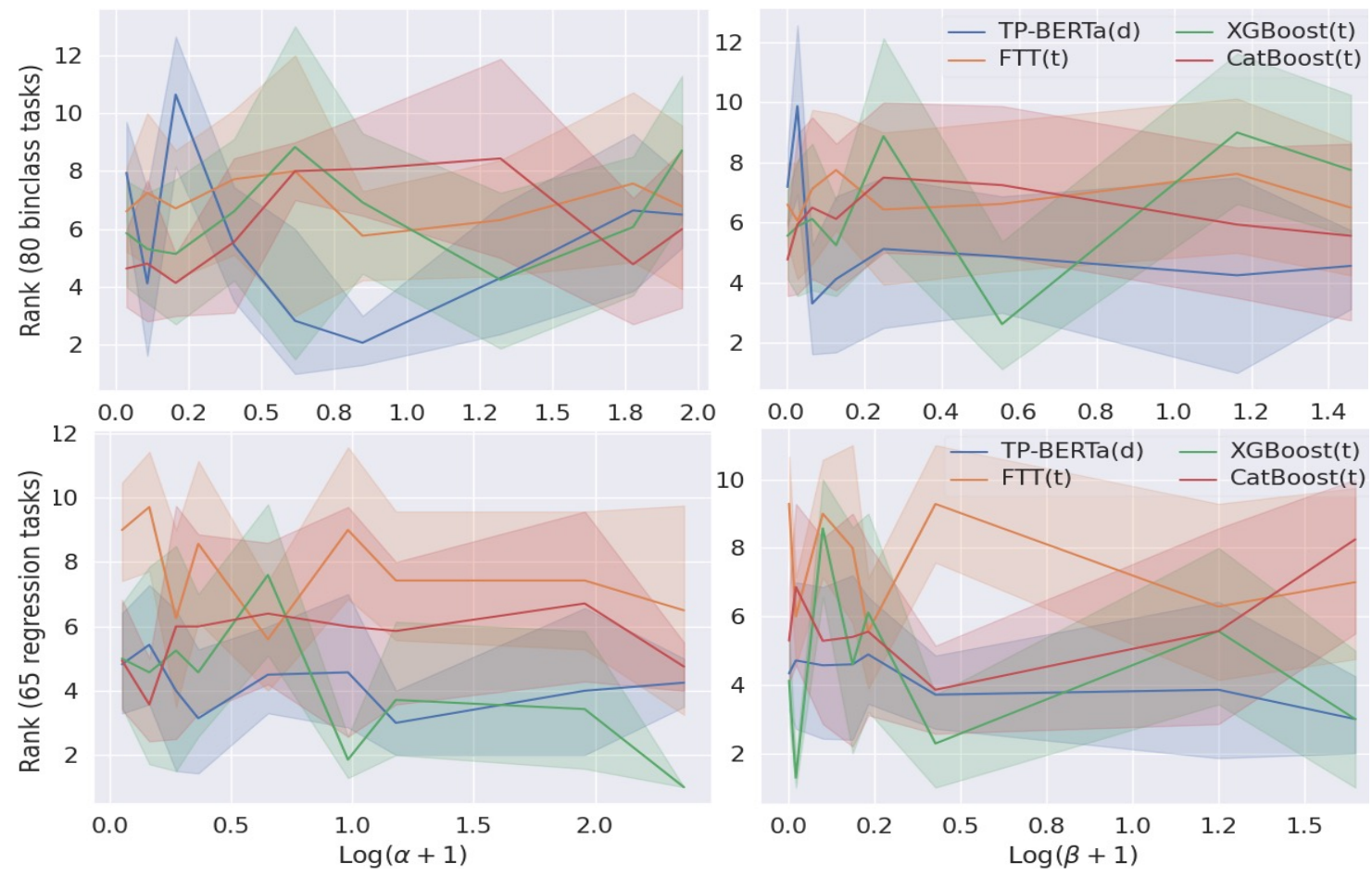

Figure 2: Rank variation curve plots of several representative models with respect to variations of some feature type characteristics. Each point represents a set of datasets in a range of $\alpha$ or $\beta$.

lar regime (specified in "**Datasets**"of Sec. 3.1). Yet, it is worth noting that the pre-trained TP-BERTa exhibits a significantly different progress and competitive performances. This notable improvement may be attributed to the generalization ability of the pre-trained LMs (e.g., GPT-3 (Brown et al., 2020)). A medium-sized dataset (with $<$ 10K points and low-dimensional features) may not have sufficient information for non-pre-trained DNNs, while LMs are able to leverage semantic information from feature names and structured values. Besides, our RMT approach further enables the LMs to handle numerical values in the language space (Sec. 3.3 discusses the necessity of RMT). Few previous deep tabular models were evaluated in such data settings, and this is the first time an extensive comparison on typical tabular data is brought to the forefront. As for cross-table models, TransTab was inspired by overlapped columns between pre-training datasets and downstream ones, which can benefit on domain datasets (e.g., medical tables), but general tables can contain many features from various domains, thus constraining its application. XTab adopted dataset-specific featurizers, though a Transformer backbone is shared; it misses the inherent relationship between features of different datasets and learns the feature embeddings from scratch, which may be trapped in insufficiently generalized data patterns. TP-BERTa is able to exploit feature semantics, e.g., the patterns learned on feature values "male & female" in pre-training can be inherently transferred to "boy & girl" by LMs without compulsory need for overlapped features or dataset-specific encoders.

**Comparison from the Feature Perspectives.** Since LMs typically operate on discrete texts, we further investigate from the perspective of feature type distributions. We report ranks among the datasets with various feature type distributions in Table 1. One can see that TP-BERTa achieves stably better performances (both "Ours$_j$" and "Ours$_s$') when the categorical feature type gradually becomes dominating (a larger $\alpha$ or $\beta$). This can be intuitively explained by LMs' ability to understand meaningful structured values (as discrete strings). Even among the datasets with at least one categorical feature ($\alpha > 0$), TP-BERTa still leads the performances on both task types (also shown in Fig. 2). However, if all features are numerical ($\alpha = 0$), TP-BERTa performs inferiorly. This may be due to its LM nature that precision loss in numerical representation is inevitable. For a more detailed illustration, we show in Fig. 2 rank variation curve plots across datasets grouped in different ranges of feature type distributions. Overall, TP-BERTa is stably promising when discrete features begin to dominate in the datasets, while for purely numerical datasets, GBDTs or FTT are still better choices (especially for classification tasks). Since there can exist useless tabular features, we introduce the ratio of Shapley value sums between categorical and numerical features (i.e., $\beta$, the right column of Fig. 2); an expected smoother trend that TP-BERTa performs better on datasets with larger $\beta$ is observed. Additionally, we empirically observe: (1) XGBoost highly relies on hyperparameter tuning and thus performs unstably (shown in its standard deviations and Fig. 2); (2) in contrast, CatBoost, just using default hyperparameters, is often a good choice, especially on datasets in which categorical features dominate (the same was suggested in (Prokhorenkova et al., 2018)).

**Comparison from the Data Volume Perspective.** We show rank variations on data volumes in the Appendices (Fig. 5). Similar to studying the feature type distribution effects, we examine the trend in two dataset groups ($\beta < 0.1$ and $\beta \geq 0.1$). In the typical tabular regime, the choices are mostly influenced by the distributions, while from the data scale dimension, no special trend is observed.

Table 2: Performance changes on encoding strategy substitution and IFA ablation using 80 binary classification datasets. The column "$|\Delta| \leq 0.5\%$" denotes the number of datasets with AUC variation less than 0.5% (these datasets are called "insignificantly changed datasets" due to different random seeds); the other "$\Delta$" columns use similar denotations. "Avg. diff." means the average performance difference on significantly changed datasets. "Avg. training time ratio" is the average ratio of training time compared to using the IFA module. Appendix 11 gives more detailed performances.

| Comparison (numerical encoding strategies) | | | | |
|---|---|---|---|---|
| Substitution | $\|\Delta\| \leq 0.5\%$ | $\Delta < -0.5\%$ | $\Delta > 0.5\%$ | Avg. diff. |
| Value2Str (Borisov et al., 2023) | 16 | 54 | 10 | -12.45% |
| VMFE (Ye et al., 2024) | 34 | 36 | 10 | -3.44% |
| **Ablation (w/o IFA module)** | | | | |
| Avg. training time ratio | $\|\Delta\| \leq 0.5\%$ | $\Delta < -0.5\%$ | $\Delta > 0.5\%$ | Avg. diff. |
| 1.32 | 14 | 52 | 14 | -4.17% |

**Joint Task Type Pre-training.** A more expensive TP-BERTa version is conducted on "Ours$_j$" by pre-training on binary classification and regression tasks jointly. A similar trend as "Ours$_s$" is observed, with a stably inferior performance. This may be due to: (1) incompatible natures of classification and regression tasks, (2) the dataset-specific head pre-training strategy is unsuitable for the combined patterns of classification and regression, or (3) a more powerful base LM is needed for such complicated data configuration. This will become a part of our future study.

**Takeaway.** We empirically show a strong potential for well-adapted LMs to fill the void of previous DNN-based tabular learners under typical tabular data settings, and demonstrate the capability of TP-BERTa to handle tabular prediction tasks, especially those with informative categorical features, which can help architecture selection based on feature type characteristics in future studies.

### 3.3 Why Were LMs Neglected on Tabular Prediction?

Only a few previous studies directly employed LM-based learners for tabular prediction tasks. Their numerical encoding strategies can be categorized into: (1) value string methods (directly treating numerical values as strings, e.g., GReaT (Borisov et al., 2023) and TapTap (Zhang et al., 2023)); (2) value-multiplied feature name embeddings (e.g., CT-BERT (Ye et al., 2024), FTT (Gorishniy et al., 2021), and TransTab (Wang & Sun, 2022)). Besides, they all fed a longer templated table text to LMs, which may further incur a heavy training burden. In this section, we compare our RMT approach with two other strategies, and conduct ablation study on the intra-feature attention (IFA) module to demonstrate that refining single-feature information before the LM processing is a better and computation-friendly adaption. Since each pre-training round is costly and is equivalent to evaluation under the same condition, we use the non-pre-trained TP-BERTa (initialized with the RoBERTa weights) for the subsequent comparison and ablation studies (marked in the Appendix tables). To show a more clear and quantifiable comparison, we conduct our analysis on the binary classification datasets (Table 2). To alleviate the impact of performance fluctuations caused by random seeds, we exclude the datasets on which the performance changes are insignificant (the column "$|\Delta| \leq 0.5\%$") in average difference calculation (the column "Avg. diff."). The following sections use a similar analysis method. We present the detailed results in the Appendices (Table 11).

**Numerical Encoding Strategy Comparison.** After directly substituting RMT with "value string" (Value2Str) or "value-multiplied feature name embeddings" (VMFE), changes in performance are observed (the upper half of Table 2). Both the previous strategies hurt AUC scores on most significantly changed datasets with average declines of 12.45% and 3.44%, respectively. There are still 10 datasets with better performances on both substitutions. This is probably due to insufficient embedding learning: Since in non-pre-trained TP-BERTa, magnitude embeddings are randomly initialized, direct finetune on downstream tasks faces a risk of data inadequacy to learn precise representations.

**IFA Module Ablation.** Performances and training time changes are reported in the lower half of Table 2, by removing IFA and directly feeding all feature names and values to the LM as done in previous works. A noticeable performance degradation occurs on 52 datasets ($\Delta < -0.5\%$) with an average AUC decline of 4.17% (on $52 + 14 = 66$ datasets). This indicates that LMs are likely to be confused when they process a pile of unmatched feature name-value texts, giving

Table 3: Performance changes by comparing the pre-trained TP-BERTa with (1) TP-BERTa randomly initialized and (2) TP-BERTa initialized with the RoBERTa weights. "Avg. diff." is calculated by excluding the datasets with $|\Delta| \leq 0.5\%$.

| Comparison (w/ no pre-training) using 80 binary classification datasets | | | | | | |
|---|---|---|---|---|---|---|
| Initialization | $\|\Delta\| \leq 0.5\%$ | $\Delta < -0.5\%$ | $\Delta > 0.5\%$ | $\Delta < -3\%$ | $\Delta > 3\%$ | Avg. diff. |
| Random | 29 | 41 | 10 | 26 | 5 | -3.16% |
| RoBERTa | 26 | 35 | 19 | 21 | 6 | -2.79% |

them additional burden to learn correct matchings while fully connected attention in Transformer-based LMs interacts a name with values from other features. The IFA module explicitly fuses the name-value pair of a single feature into a vector before passing it to the LM, guarding against noisy interactions from other features. Besides, a shorter input sequence length (equal to the feature amount) accelerates learning (e.g., see the 1.32 average training time ratio without IFA in Table 2).

Since TP-BERTa adopts both magnitude-aware numerical encoding and intra-feature pre-processing before the LM process, it acquires a significantly better ability as well as friendly computation cost, becoming competitive with GBDTs and other deep models. Our comparison shows that simply treating tables as normal texts can pose difficulties for LMs to understand structured tabular features, thus decreasing their potential on high-precision demanding tasks such as tabular predictions.

### 3.4 TP-BERTa Transferability on Tabular Data

Table 3 reports performance changes by comparing the non-pre-trained TP-BERTa (initialized by random weights or RoBERTa weights) with the pre-trained one ("Ours$_s$" in Table 1). Overall, over 3% AUC increase is attained on 26 (comparing to random weights) and 21 (comparing to RoBERTa weights) datasets using pre-training, and the average improvement on significantly changed datasets is 3.16% and 2.79%, respectively. It seems that using the RoBERTa weights is better than random weights, as LM weights have inherently entailed meaningful semantic knowledge. A more significant leap can be achieved by further pre-training on extensive tabular data. This indicates that LMs are also effective in transferring tabular data knowledge and suitable for cross-table pre-training.

### 3.5 The Necessity of Other Design Details

Since there are several key differences in our TP-BERTa design compared to the common Transformer-based LMs, we further examine their necessity by ablation studies. By evaluating different **magnitude token numbers** ($n_{\text{bin}} = 256$ as default), we find that using 128 and 32 tokens yields an average AUC decline of 2.06% and 3.59%, respectively. This indicates that a larger $n_{\text{bin}}$ for numerical value representation benefits performances on most datasets, while a few tables favor a smaller $n_{\text{bin}}$, which may be due to over-representation of excessive magnitude tokens. The detailed results with analysis are presented in Appendix C, which further discusses the regularization efforts on magnitude embeddings of the **magnitude-aware triplet loss function** (Eq. (3)) and the function of **removing position encoding for value vectors** (Eq. (7)). We find that the magnitude-aware triplet loss function potentially facilitates fast convergence and over-fitting reduction.

## 4 Conclusions and Future Work

This paper undertook the first study of the substantial difficulties of continuous value representation and tabular feature organization in building LM-based tabular DNNs. We designed and deployed two bespoke adaptions, relative magnitude tokenization and intra-feature attention, to explore the possibilities of using pre-trained LMs on tabular prediction tasks. Our proposed TP-BERTa exhibits unprecedented progress over various non-LM DNNs, and is competitive with GBDTs under the typical tabular prediction regime, contributing a powerful DNN alternative for typical tabular data.

While our approach has significantly improved the performance of language models in handling numerical features in tables, TP-BERTa currently excels more on tables dominated by categorical features. Besides, it was witnessed that some tables prefer a small magnitude token number. We will conduct more endeavors on better numerical representation in future tabular prediction studies.

## ACKNOWLEDGMENTS

The research of Jiahuan Yan, Hongxia Xu and Jian Wu was supported in part by the National Natural Science Foundation of China under grants No. 62176231 and No. 82202984, and the Zhejiang Key R&D Program of China under grant No. 2023C03053. The research of Jintai Chen and Jimeng Sun was supported by NSF award SCH-2205289, SCH-2014438, and IIS-2034479.

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

## A LIMITATIONS

Since our TP-BERTa relies on the semantic knowledge of LMs to transfer feature patterns from pre-training feature names to downstream ones, it implicitly requires meaningful and clear feature semantics. However, in the real world, there always exist tables with unclear feature names or values, such as quantum physics experiment tables containing lots of uncommon quantum descriptor feature names and medical domain substituting specific feature values with meaningless encoding to protect patient privacy. This suggests that LM-based tabular DNNs cannot take their inherent advantages of feature semantic understanding in privacy-sensitive (e.g., federated learning) or semantic-incomplete (missing original meanings in data collection) tables. For an uncommon domain, LMs pre-trained on domain corpora may be utilized as base models. Besides, LMs own a larger space of parameters and hyperparameters, making them more time-consuming in hyperparameter tuning with a potentially higher performance ceiling compared to non-LM tabular DNNs. Hence, we directly finetune TP-BERTa with default hyperparameters for fairness of time, in which case it can achieve adequate results with less time than other tuned methods.

## B DATASET INFORMATION AND MAIN EXPERIMENTAL DETAILS

We provide detailed information of the pre-training datasets in Table 6 and detailed information of the downstream datasets in Table 7 and Table 9. Detailed baseline performances are given in Table 8 and Table 10. To represent distributions of feature types in a dataset, we define $\alpha$ as the feature amount ratio between the categorical type and numerical type. Since there exist datasets with usefulness features, we further define $\beta$ as the Shapley value (calculated with default XGBoost in Appendix E) sum ratio between categorical features and numerical features. These are used as references of feature type characteristics. Specifically, $\alpha$ and $\beta$ of the $i$-th dataset are formulated as:

$$\alpha_i = \frac{\#Cat.^{(i)}}{\#Num.^{(i)}}, \tag{12}$$

$$\beta_i = \frac{\sum_{f \in Cat.^{(i)}} \text{Shapley value}(f)}{\sum_{f \in Num.^{(i)}} \text{Shapley value}(f)}. \tag{13}$$

We exclude datasets with pure categorical features to avoid zero division, while as an LM, TP-BERTa can inherently handle discrete features. We provide the dataset frequency in different feature type distribution ranges in Table 4.

Table 4: Dataset frequency of two downstream collections in several feature type distribution ranges.

| Collection | $\beta = 0$ | $\beta \in (0, 0.5)$ | $\beta \in [0.5, 1.0)$ | $\beta \geq 1.0$ |
|---|---|---|---|---|
| 80 binary classification datasets | 24 | 33 | 7 | 16 |
| 65 regression datasets | 24 | 28 | 4 | 9 |

## C RESULTS AND ANALYSIS OF OTHER DESIGN COMPARISONS

**The Number of Magnitude Tokens.** In Sec. 2.1, we set the maximum number of magnitude tokens, $n_{\text{bin}} = 256$, for TP-BERTa. In fact, the C4.5 decision tree splits the value range in a greedy fashion, and the actual number of leaves can be less than 256 (e.g., a small dataset with less than 256 points). Hence, we choose a conservative method to balance between "not too many new words to learn" and "enough precision for relative magnitude representation". In Table 5, we present the performance changes by setting $n_{\text{bin}}$ to 32 and 128 separately. As expected, the overall performance gradually drops when using a smaller token number to represent the numerical value magnitude. We find that some datasets favor a smaller $n_{\text{bin}}$, which may be attributed to over-representation of too many magnitude tokens. Thus, a better solution is to set a large $n_{\text{bin}}$ for pre-training in order to enhance the upper limit of magnitude representation capability, and search for a reasonable dataset-specific $n_{\text{bin}}$ on downstream tasks.

**Regularization on Magnitude Embeddings.** Since all the magnitude embeddings are randomly initialized, in Sec. 2.1, we propose a magnitude-aware triplet loss (see Eq. (3)) to assist the learning process. Fig. 3 presents the validation AUC curves of several datasets on using our regularization or not using it during finetuning, which shows that the designed regularization provides a potential of fast convergence and overfitting reduction. We use the triplet loss only in pre-training for a smoother and accelerated learning, and exclude it in actual finetune because the loss has converged in pre-training.

**No Position Encoding for Value Vectors.** In Eq. (7), we explicitly remove position encoding in value vectors of self-attention since position embeddings may distort randomly initialized magnitude embedding learning. Table 5 shows a major performance decline when adding position encoding. The reason for this probably lies in semantic corruption of magnitude tokens, since numerical values need precise representations to convey magnitude information.

Table 5: Performance changes with respect to using different magnitude token numbers (the default is $n_{\text{bin}} = 256$; see Sec. 2.1) and position encoding in value vectors (see Eq. (7)).

| Comparison (magnitude token numbers) using 80 binary classification datasets | | | | |
|---|---|---|---|---|
| Substitution | $\|\Delta\| \le 0.5\%$ | $\Delta < -0.5\%$ | $\Delta > 0.5\%$ | Avg. diff. |
| $n_{\text{bin}} = 32$ | 27 | 40 | 13 | -3.59% |
| $n_{\text{bin}} = 128$ | 42 | 26 | 12 | -2.06% |
| **Ablation (w/ value vector position encoding)** | | | | |
| 80 binary classification datasets | $\|\Delta\| \le 0.5\%$ | $\Delta < -0.5\%$ | $\Delta > 0.5\%$ | Avg. diff. |
| | 36 | 31 | 13 | -2.35% |

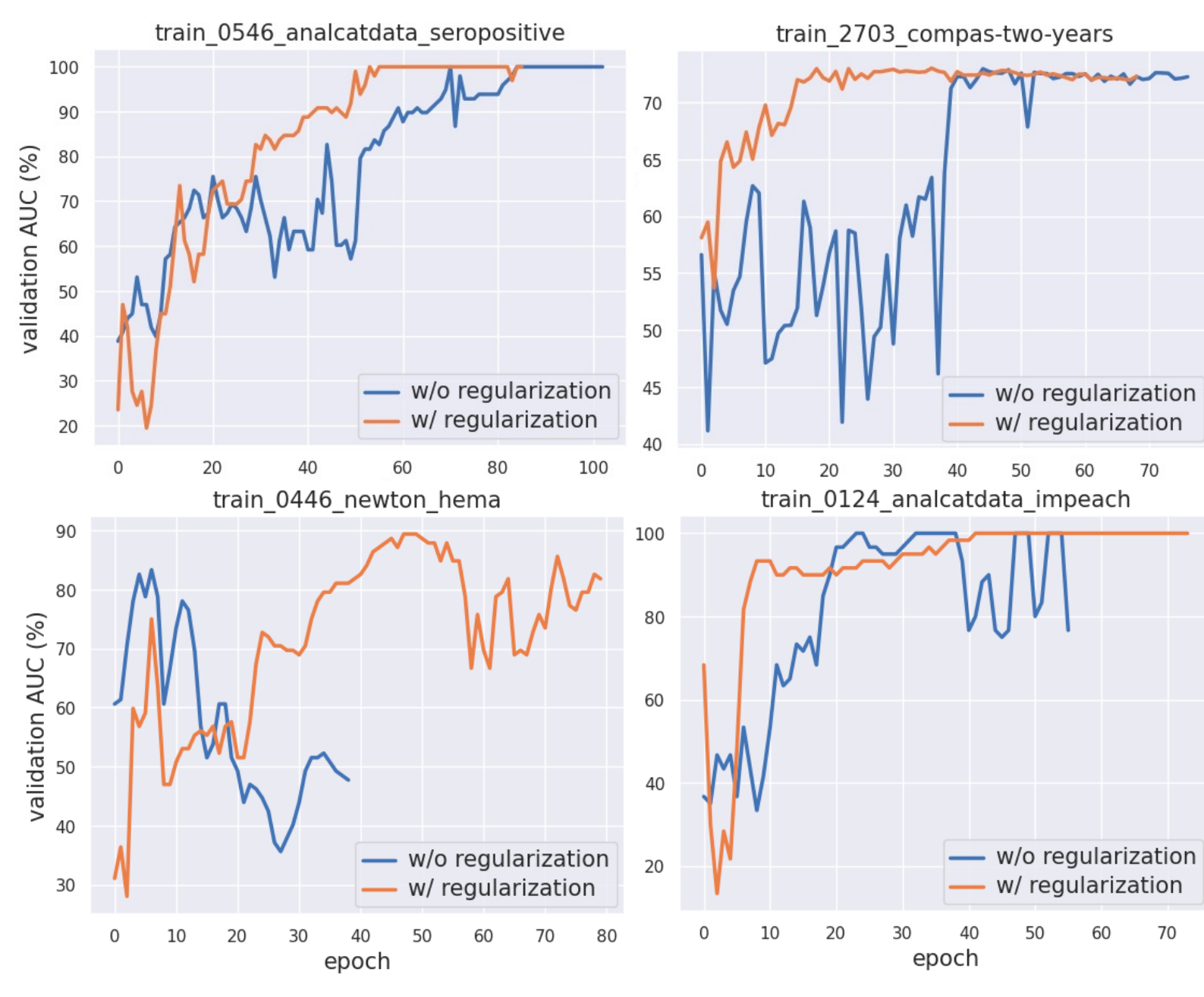

Figure 3: Comparison of using regularization or not using it during finetuning on the non-pre-trained TP-BERTa. The validation AUC curves of several representative binary classification datasets show that the effect of the magnitude-aware triplet loss (see Eq. (3)) is to help quick convergence and avoid potential overfitting of TP-BERTa. In experiments, we use this regularization only in pre-training to smooth and accelerate the learning process.

# D PRE-TRAINING DETAILS

**Starting Point.** We reuse the weights of the RoBERTa-base with the HuggingFace Transformers API. The additional $n_{\text{bin}}$ magnitude embeddings and the IFA module are randomly initialized.

**Runtime Environment.** Pre-training is conducted with PyTorch version 1.9.0, CUDA version 11.3, and HuggingFace Transformers package version 4.18.0, using 4 NVIDIA A100 PCIe 40GB and 2 Intel 6248R 48C@3.0GHz.

**Pre-training Process.** We pre-train three versions of TP-BERTa: pre-training only on binary classification datasets, or only on regression datasets (these two versions constitute "Ours$_s$" in Table 1), or jointly pre-training on both types ("Ours$_j$" in Table 1). These three versions share the same maximum epoch number of 30 and the total batch size of 512, using the best average validation loss across all the datasets to save the checkpoint. The same pre-training learning rate and linear decay in (Liu et al., 2019) are used. The pre-training on binary classification tasks and regression tasks took 72 hours and 98 hours respectively, and the one on both types of tasks took 143 hours. We provide several loss curves during pre-training in Fig. 6 and validation metric curves on several pre-training datastes in Fig. 7.

**Analysis.** In most cases, the TP-BERTa version pre-trained jointly on both task types yields sightly lower validation scores on the pre-training datasets compared to the two TP-BERTa versions pre-trained on a single task type (see Fig. 7), and the gap is more noticeable on binary classification datasets. This probably leads to a smaller overall rank difference between Ours$_j$ and Ours$_s$ on regression tasks (see Table 1).

# E HYPERPARAMETER TUNING

For the baselines of XGBoost, CatBoost, MLP, AutoInt, DCNv2, TabNet, and FT-Transformer, we reuse the implementations, default settings, and hyperparameter search spaces in (Gorishniy et al., 2021). For SAINT, the hyperparameter space in (Somepalli et al., 2022) is used. For TransTab and XTab, we follow the same settings as in (Zhu et al., 2023), using the default hyperparameters in (Wang & Sun, 2022) for TransTab and the best checkpoint on the validation set for XTab. As for TP-BERTa (including the joint pre-trained version "Ours$_j$" and the single pre-trained one "Our$_s$"), we keep the default hyperparameters of 1e-5 learning rate without weight decay. All the baselines use AdamW (Loshchilov & Hutter, 2019) as the optimizer and the Optuna-driven tunning.

# F INTERPRETABILITY OF RMT

In Fig. 4, we visualize the TP-BERTa's 256 magnitude tokens by directly applying t-SNE algorithm using scikit-learn package (with default function parameters). Interestingly, even if we have randomly placed them in the language space at first, after pre-training a clear inherent distribution is captured, all tokens lie on a highly regular manifold and successfully maintain an intuitive assumption that the embedding of a numerical value should close to the embedding of a nearby one. This empirically demonstrates the TP-BERTa is sensitive to the numerical value magnitudes and benefits from the captured regular relationship among the relative magnitudes, which interprets its significant progress on the existing LM-based tabular models (e.g., directly treating values as raw strings).

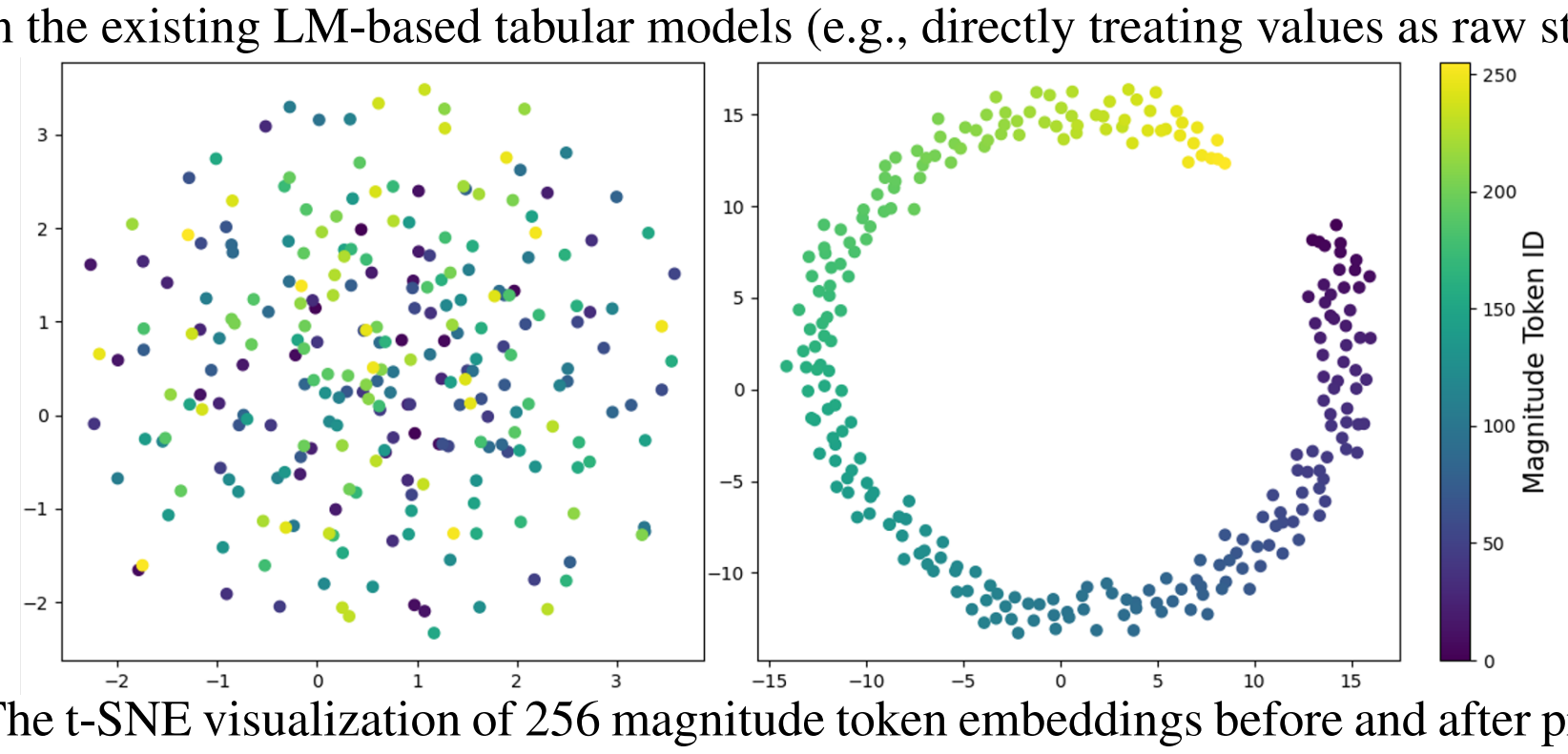

Figure 4: The t-SNE visualization of 256 magnitude token embeddings before and after pre-training.

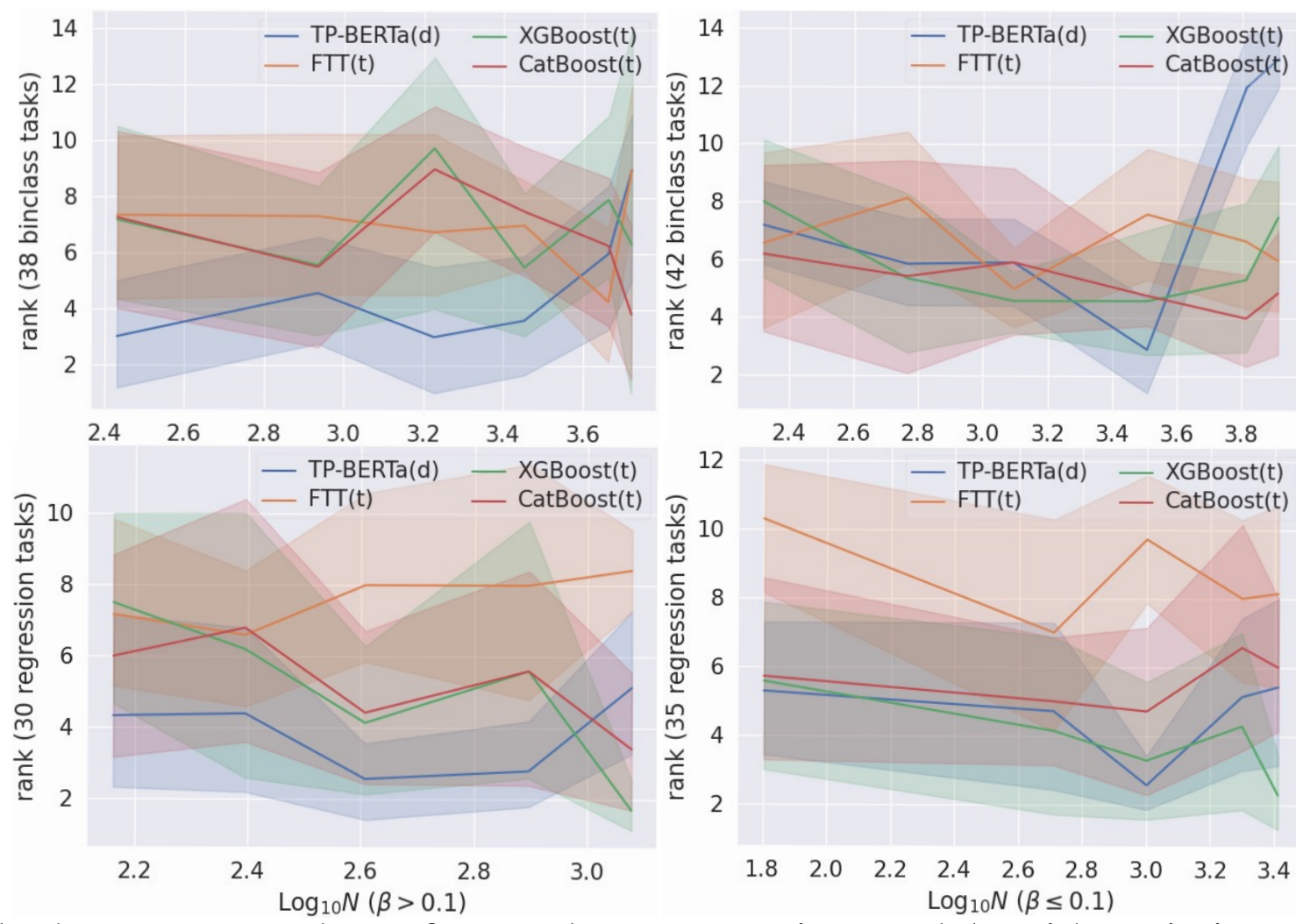

Figure 5: Rank change curve plots of several representative models with variations of data volume ($N$). We divide the datasets into two groups (the first column is for "$\beta > 0.1$" and the second column is for "$\beta \leq 0.1$") to alleviate the impact from the feature type distributions. The split value 0.1 is chosen by keeping a roughly equal number of datasets in both groups.

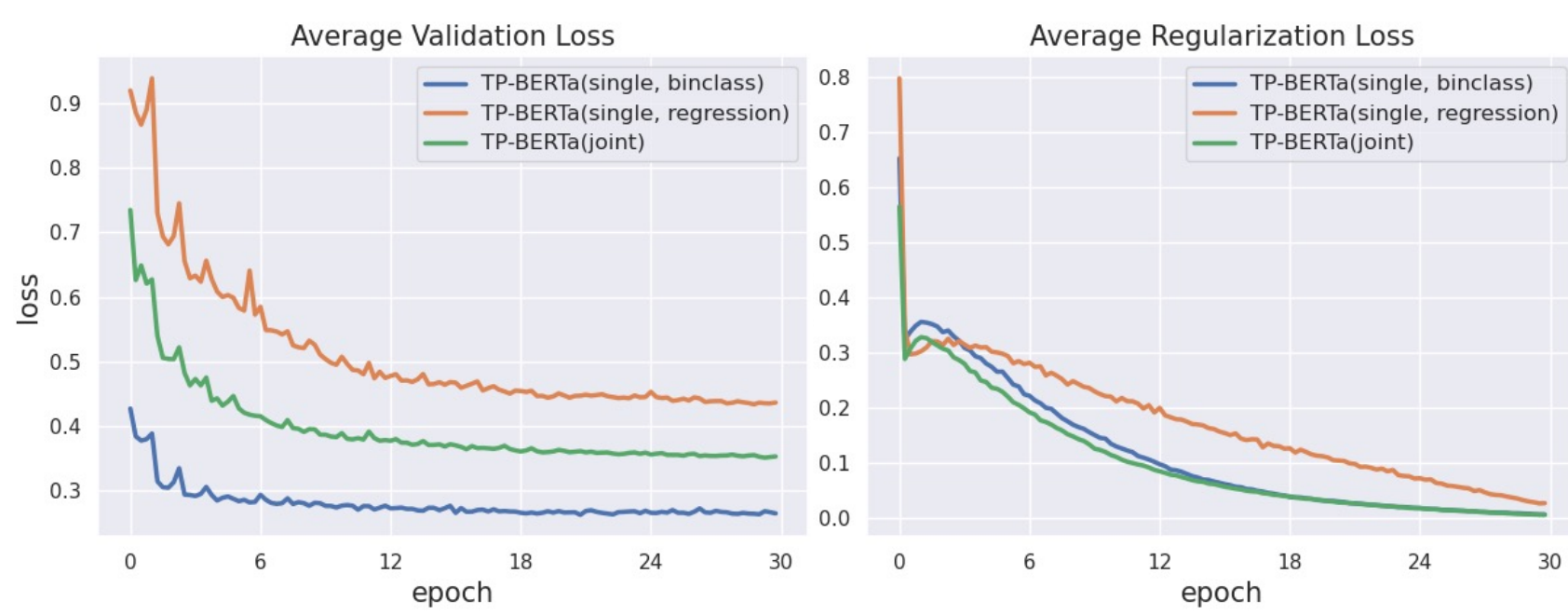

Figure 6: The average validation loss curves and regularization loss (Eq. (3)) curves in pre-training. "TP-BERTa(single, binclass)" and "TP-BERTa(single, regression)" are the two versions separately pre-trained on binary classification datasets and regression ones (constituting "Ours$_s$" in Table 1); "TP-BERTa(joint)" is the version pre-trained on both task types ("Ours$_j$" in Table 1).

# G Evaluations on Other Data Scenarios

## G.1 Imbalanced Label Distribution

To inspect the performances on imbalanced datasets, we create pivot tables by filtering datasets whose minor-class proportions, i.e., $p = \min(\#positive, \#negative)/\#sample$, are less than 1/3 (32 datasets), 1/5 (18 datasets), 1/8 (12 datasets), 1/20 (4 datasets) from 80 binary classification datasets. The results are reported in the upper part of Table 15. It is obvious that TP-BERTa outperforms baselines in moderate class-imbalance situations. In the extremely imbalanced situations (4 datasets, with $p < 0.05$), GBDTs showcase dominating performances, but TP-BERTa still outperforms most DNN approaches. Perhaps this is attributed to TP-BERTa leveraging the transferability and semantic understanding capabilities of the language model, thus resulting in consistent performance across various levels of data imbalance.

## G.2 Multi-class Classification

We additionally experiment on 32 downstream multi-class datasets from the database, the data statistics and results are reported in Table 12 and the middle part of Table 15 respectively, and the TP-BERTa used here is the version pre-trained on 101 binary classification datasets.

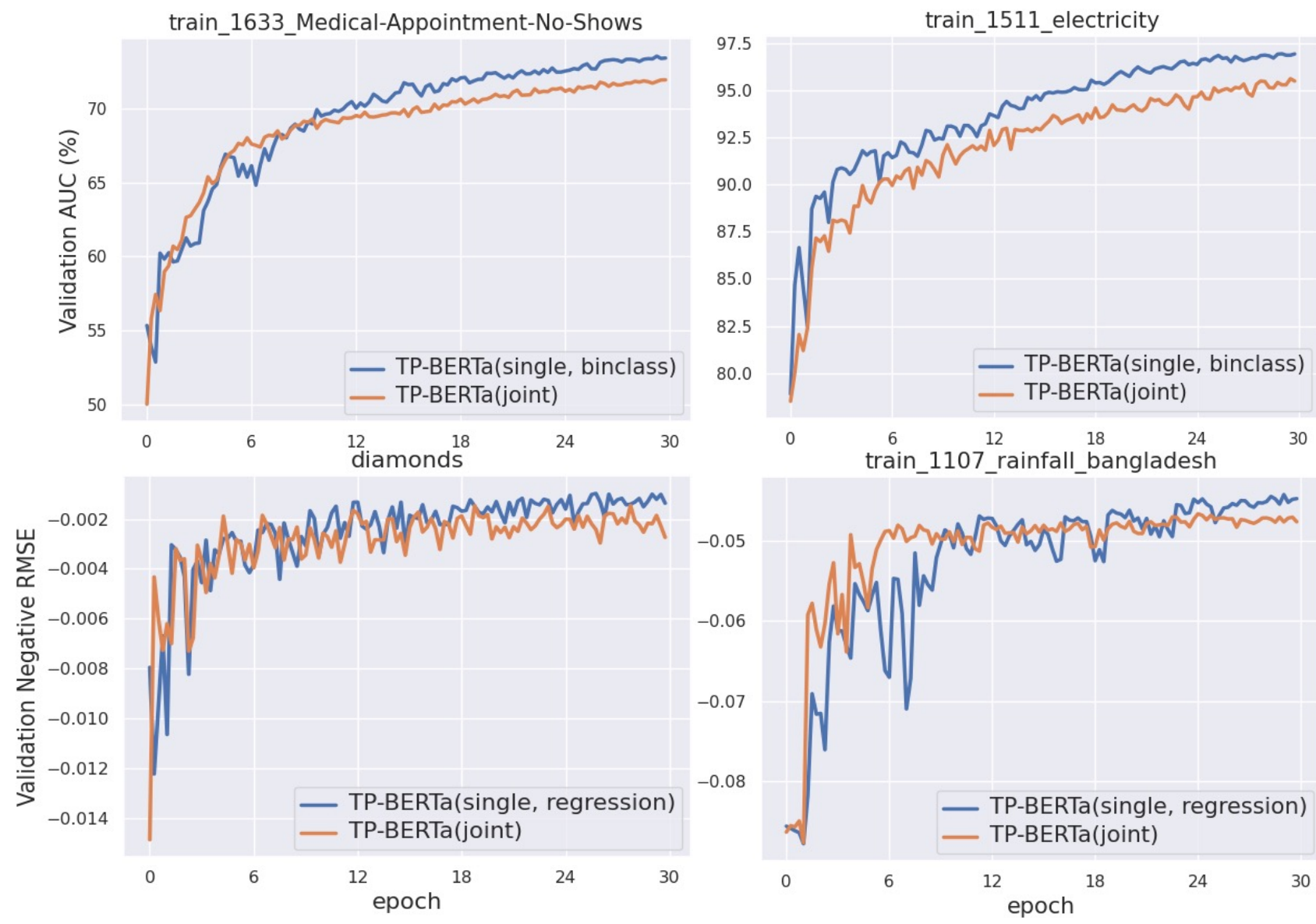

Figure 7: Validation score curves in AUC and RMSE on several pre-training datasets.

### G.3 Medical Applications

Medical data and labels hold inherently greater value than those from other domains. Therefore, we further inspect whether our pre-trained model can achieve notable performance on medical domain tasks, such as patient risk prediction and clinical trial outcome prediction, by leveraging knowledge learned and transferred from diverse domains, which typically offer more cost-effective data sources. We identified and filtered out all 25 medical tasks from 145 main experiment downstream datasets (statistics and results are given in Table 13 and Table 14). Refer to the bottom row of Table 14, the results indicate that the pre-trained TP-BERTa model significantly outperforms Gradient Boosting Decision Trees (GBDTs) and other Deep Neural Networks (DNNs) that are trained in the supervision manner and undergo meticulous hyperparameter tuning. However, our TP-BERTa achieved this without the need for hyperparameter tuning.

The noteworthy performance suggests that we can harness more affordable data sources to alleviate reliance on costly healthcare data in clinical practice. This illuminates a promising pathway towards reducing the need for extensive medical data and expediting the development of algorithms for healthcare applications.

Table 6: Statistics of 101 binary classification datasets and 101 regression datasets for pre-training.

| Task Type | Binary classification | | | | Regression | | | |
|---|---|---|---|---|---|---|---|---|
| ID | Dataset name | # samples | # num. features | # cat. features | Dataset name | # samples | # num. features | # cat. features |
| 0 | 1969_CPS1988 | 28155 | 3 | 3 | Aemf1 | 41714 | 7 | 11 |
| 1 | ai4i2020 | 10000 | 5 | 6 | 1656_Candy-crush | 16865 | 1 | 2 |
| 2 | Rain in Australia | 100000 | 13 | 7 | 2137_house_sales | 21613 | 12 | 3 |
| 3 | airline_passenger | 100000 | 4 | 18 | avocado_sales | 18249 | 9 | 4 |
| 4 | AV Healthcare | 100000 | 4 | 11 | 2198_turing_binary | 10000 | 19 | 3 |
| 5 | 0074_BNG(tic-tac | 39366 | 0 | 9 | BNG(echoMonths) | 17496 | 6 | 3 |
| 6 | Bank Customer Churn | 10000 | 4 | 6 | 2664_diamonds | 53940 | 6 | 3 |
| 7 | Bank Marketing | 45211 | 7 | 9 | BNG(lowbwt) | 31104 | 2 | 7 |
| 8 | Travel Insurance | 63326 | 4 | 5 | 1295_delays_zurich | 27327 | 9 | 7 |
| 9 | bank | 11162 | 7 | 9 | Brazilian_houses | 10692 | 5 | 3 |
| 10 | 0634_mozilla4 | 15545 | 3 | 1 | CPS1988 | 28155 | 2 | 4 |
| 11 | bank_customer_survey | 45211 | 7 | 9 | 1728_HSI-Futures | 87645 | 5 | 0 |
| 12 | 0080_BNG(vote) | 100000 | 0 | 16 | Customer-Churn | 10000 | 4 | 10 |
| 13 | Bank_marketing_data | 45211 | 7 | 9 | 2672_kings_county | 21613 | 13 | 8 |
| 14 | 2149_electricity | 38474 | 7 | 1 | dataset_sales | 10738 | 4 | 10 |
| 15 | BNG(breast-w) | 39366 | 9 | 0 | 2707_seattlecrime6 | 52031 | 3 | 1 |
| 16 | 2668_cps88wages | 28155 | 3 | 3 | diamonds | 53940 | 6 | 3 |
| 17 | campaign33 | 12870 | 6 | 9 | elevators | 16599 | 14 | 4 |
| 18 | 0677_COMET_MC_SAMPLE | 89640 | 2 | 1 | fifa | 18063 | 5 | 0 |
| 19 | Candidate Selection | 73147 | 1 | 11 | houses | 20640 | 8 | 0 |
| 20 | 2683_electricity | 38474 | 7 | 0 | 1781_SDSS-16 | 100000 | 11 | 3 |
| 21 | Cardio Disease | 70000 | 6 | 6 | house_sales_reduced | 21613 | 12 | 6 |
| 22 | 0710_Agrawal1 | 100000 | 6 | 3 | 1799_NSE-Stocks-Data | 100000 | 8 | 1 |
| 23 | Car_Insurance_Claim | 10000 | 3 | 14 | transactions | 100000 | 2 | 2 |
| 24 | 2687_Diabetes130US | 71090 | 5 | 2 | kc_final | 21613 | 12 | 6 |
| 25 | Churn_Modelling | 10000 | 4 | 6 | WorkersCompensation | 100000 | 4 | 6 |
| 26 | MonkeyPox33 | 25000 | 0 | 9 | MAMe_dataset | 37407 | 2 | 3 |
| 27 | 0948_COMET_MC_SAMPLE | 100000 | 2 | 0 | 1870_product | 11385 | 4 | 15 |
| 28 | Classification | 88858 | 4 | 4 | MiamiHousing2016 | 13932 | 13 | 3 |
| 29 | 2724_shrutime | 10000 | 4 | 6 | NewFuelCar | 36203 | 15 | 2 |
| 30 | classifying_document | 11539 | 4 | 1 | socal2 | 15474 | 3 | 1 |
| 31 | 1249_sf-police | 100000 | 1 | 4 | star_classification | 100000 | 10 | 3 |
| 32 | customer_airways | 50000 | 4 | 7 | stats | 10000 | 7 | 2 |
| 33 | 0137_BNG(labor) | 100000 | 6 | 9 | 0678_BNG(auto_price) | 100000 | 13 | 1 |
| 34 | diabetes_prediction | 100000 | 4 | 4 | 1319_house_sales | 21613 | 12 | 6 |
| 35 | Employee-Turnover-at | 34452 | 8 | 1 | 1946_avocado_sales | 18249 | 9 | 4 |
| 36 | filtered_customer | 49982 | 4 | 7 | 0682_BNG(lowbwt) | 31104 | 2 | 7 |
| 37 | 1294_airlines | 26969 | 3 | 2 | 1362_pm25dataset | 43800 | 5 | 3 |
| 38 | flight_delays_train | 100000 | 2 | 4 | 0684_BNG(autoPrice) | 100000 | 14 | 0 |
| 39 | Warehouse_block | 10999 | 3 | 7 | 2062_black_friday | 100000 | 1 | 8 |
| 40 | Fraud | 100000 | 6 | 2 | 0685_BNG(pharynx) | 100000 | 1 | 9 |
| 41 | 1366_bankmarketing | 41188 | 5 | 15 | 0688_BNG(echoMonths) | 17496 | 6 | 3 |
| 42 | fusion_experiment | 100000 | 16 | 2 | 1509_california | 20640 | 8 | 0 |
| 43 | Phishing websites | 95910 | 5 | 5 | 0690 | 100000 | 2 | 7 |
| 44 | Health Insurance Lead | 50882 | 5 | 7 | 1510_fifa | 18063 | 5 | 0 |
| 45 | 0158_BNG(heart | 100000 | 6 | 7 | 2131_houses | 20640 | 8 | 0 |
| 46 | Horse Racing | 38248 | 1 | 3 | 0875_nfl_games | 16274 | 6 | 2 |
| 47 | 1414_AI4I2020 | 10000 | 5 | 6 | 1576_Earthquakes | 20648 | 3 | 0 |
| 48 | Hotel Reservations | 36275 | 5 | 12 | 0880_dataset_sales | 10738 | 4 | 10 |
| 49 | 1465_credit | 16714 | 7 | 3 | 1587_elevators | 16599 | 14 | 2 |
| 50 | HR Analysis Case | 54808 | 4 | 8 | 0932_mlr_rpart_rng | 92067 | 7 | 2 |
| 51 | 1511_electricity | 38474 | 7 | 1 | 1595_Oranges-vs. | 10000 | 4 | 1 |
| 52 | income_evaluation | 32561 | 5 | 9 | 0940_seattlecrime6 | 52358 | 2 | 4 |
| 53 | Preprocessed Shopee | 73539 | 14 | 6 | 1596_Cinema-Tickets | 100000 | 10 | 2 |
| 54 | Janatahack cross | 100000 | 5 | 5 | 1030 | 100000 | 4 | 6 |
| 55 | PS_20174392719 | 100000 | 6 | 2 | 1639_Melbourne | 13580 | 8 | 9 |
| 56 | JanataHack Machine | 100000 | 5 | 6 | 1107_rainfall | 16755 | 1 | 2 |
| 57 | pulsar_data_train | 12528 | 8 | 0 | 2134_Brazilian | 10692 | 5 | 3 |
| 58 | L&T Vehicle Loan | 100000 | 18 | 13 | 1140_exercises | 15000 | 5 | 1 |
| 59 | 0160_BNG(hepatitis) | 100000 | 6 | 13 | 1644_Credit-Risk | 32581 | 6 | 5 |
| 60 | law-school-admission | 20800 | 3 | 8 | 1245_Production | 50625 | 12 | 0 |
| 61 | new_train | 32950 | 4 | 11 | 1368_IMDb-Ratings | 67408 | 1 | 0 |
| 62 | 1550_credit | 16714 | 7 | 3 | 1645_Sloan-Digital | 100000 | 12 | 3 |
| 63 | League of Legends | 48651 | 10 | 4 | 1415_beijing-pm2.5 | 43824 | 8 | 3 |
| 64 | mlbootcamp5_train | 70000 | 5 | 6 | 1649_Tamilnadu-Crop | 13266 | 2 | 4 |
| 65 | Run_or_walk | 88588 | 6 | 0 | 1452_gender-by-name | 100000 | 1 | 1 |
| 66 | salary | 32561 | 5 | 9 | 1466_post-operative | 65532 | 10 | 1 |
| 67 | 0474_houses | 20640 | 8 | 0 | 1453_metro | 48204 | 3 | 4 |
| 68 | Server Logs | 100000 | 4 | 2 | 1465_internet | 65532 | 10 | 1 |
| 69 | 0046 | 100000 | 0 | 16 | 1552 | 100000 | 6 | 0 |
| 70 | Success of Bank | 30477 | 1 | 6 | 1561_Complete | 100000 | 6 | 0 |
| 71 | term_deposit | 31647 | 7 | 9 | 1587_Intel-Stock | 10361 | 5 | 0 |
| 72 | 0050_BNG(breast | 100000 | 0 | 9 | 1611_Advertising | 16288 | 3 | 11 |
| 73 | 0077_BNG(heart | 100000 | 0 | 13 | 1612_Historical | 100000 | 5 | 1 |
| 74 | 0079 | 100000 | 0 | 19 | 1613_COVID-19 | 70464 | 7 | 0 |
| 75 | 1020_Run_or_walk | 88588 | 6 | 0 | 1646_COVID19-Dataset | 34862 | 11 | 1 |
| 76 | 1254_Case-Study | 20000 | 4 | 5 | 1671_COVID-19-Mexico | 92320 | 3 | 1 |
| 77 | 1375_MAGIC-Gamma | 19020 | 9 | 0 | 1675_COVID19-cases-by | 26144 | 3 | 0 |
| 78 | 1406_law-school | 20800 | 3 | 8 | 1679_COVID-19-Rio-de | 37272 | 2 | 1 |
| 79 | 1569_COVID-19-World | 97606 | 1 | 1 | 1697_AMD-Stock-Prices | 10361 | 5 | 0 |
| 80 | 1571_Temperature | 97606 | 1 | 1 | 1704_House-Rent-in | 10692 | 4 | 7 |
| 81 | 1584_Towards-Data | 46079 | 3 | 0 | 1760_BSE-30-daily | 73066 | 5 | 1 |
| 82 | 1598_Churn-for-Bank | 10000 | 4 | 6 | 1783_Covid-19 | 56717 | 7 | 0 |
| 83 | 1621_The-Bread | 20507 | 1 | 2 | 1830_Football | 37147 | 1 | 4 |
| 84 | 1630_Default-of | 30000 | 14 | 9 | 1853_Crowdedness-at | 62184 | 3 | 6 |
| 85 | 1633_Medical | 100000 | 1 | 9 | 1858_Worldwide-Meat | 13760 | 1 | 3 |
| 86 | 1662_Toronto-COVID-19 | 14911 | 0 | 12 | 1860_Worldwide-Crop | 21165 | 1 | 3 |
| 87 | 1668_Census | 100000 | 6 | 8 | 1904_Apple-Complete | 10015 | 5 | 0 |
| 88 | 1690_Malware | 43293 | 4 | 0 | 1905_New-Delhi-Rental | 17890 | 7 | 4 |
| 89 | 1707_Employee | 34452 | 8 | 1 | 2135_Bike_Sharing | 17379 | 4 | 2 |
| 90 | 1761_Binary-Dataset | 11000 | 9 | 5 | 2136_nyc-taxi-green | 100000 | 8 | 1 |
| 91 | 1832_Bank-Marketing | 11162 | 7 | 9 | 2140 | 13932 | 12 | 1 |
| 92 | 1919_Diabetes-130 | 100000 | 2 | 22 | 2145_Brazilian_houses | 10692 | 5 | 3 |
| 93 | 1934_Diabetes-130 | 100000 | 2 | 19 | 2167_Intersectional | 10000 | 13 | 5 |
| 94 | 2176_NewspaperChurn | 15855 | 2 | 14 | 2654_naval_propulsion | 11934 | 12 | 2 |
| 95 | 2178_Churn_Telco | 100000 | 12 | 6 | 2659_video | 68784 | 10 | 8 |
| 96 | 2181_Bank_marketing | 45211 | 7 | 9 | 2673_brazilian | 10692 | 3 | 6 |
| 97 | 2701_BitcoinHeist | 24780 | 7 | 0 | 2704_Intersectional | 11000 | 13 | 6 |
| 98 | 2726_Insurance | 23548 | 3 | 7 | 2711_medical_charges | 100000 | 3 | 0 |
| 99 | 2734_airlines | 100000 | 3 | 2 | 2746_click_prediction | 39926 | 2 | 3 |
| 100 | 2736_Shipping | 10999 | 2 | 7 | 2749_amazon_employee | 32769 | 7 | 1 |

Table 7: Statistics of 80 downstream binary classification datasets.

| ID | Dataset name | # samples | # num. | # cat. | $\alpha$ | Num. Shapely sum | Cat. Shapely sum | $\beta$ |
|---|---|---|---|---|---|---|---|---|
| 0 | BankNoteAuthentication | 1372 | 4 | 0 | 0.00 | 9.21 | 0.00 | 0.00 |
| 1 | bt_dataset_t3 | 1644 | 17 | 0 | 0.00 | 6.05 | 0.00 | 0.00 |
| 2 | 0292_cpu_small | 8192 | 12 | 0 | 0.00 | 7.61 | 0.00 | 0.00 |
| 3 | 0312_cpu_act | 8192 | 21 | 0 | 0.00 | 7.39 | 0.00 | 0.00 |
| 4 | 0345_delta_ailerons | 7129 | 5 | 0 | 0.00 | 3.98 | 0.00 | 0.00 |
| 5 | 0356_delta_elevators | 9517 | 6 | 0 | 0.00 | 3.83 | 0.00 | 0.00 |
| 6 | 0406_visualizing | 111 | 3 | 0 | 0.00 | 0.69 | 0.00 | 0.00 |
| 7 | 0419_pm10 | 500 | 7 | 0 | 0.00 | 1.06 | 0.00 | 0.00 |
| 8 | 0435_strikes | 625 | 6 | 0 | 0.00 | 4.73 | 0.00 | 0.00 |
| 9 | 0437_quake | 2178 | 3 | 0 | 0.00 | 0.33 | 0.00 | 0.00 |
| 10 | 0445_arsenic-male | 559 | 4 | 0 | 0.00 | 1.35 | 0.00 | 0.00 |
| 11 | 0446_arsenic-female | 559 | 4 | 0 | 0.00 | 1.28 | 0.00 | 0.00 |
| 12 | 0447_arsenic-female | 559 | 4 | 0 | 0.00 | 1.70 | 0.00 | 0.00 |
| 13 | 0509_pollen | 3848 | 5 | 0 | 0.00 | 0.11 | 0.00 | 0.00 |
| 14 | 1201_Gender | 3168 | 20 | 0 | 0.00 | 6.34 | 0.00 | 0.00 |
| 15 | 1600_VulNoneVul | 5692 | 16 | 0 | 0.00 | 0.91 | 0.00 | 0.00 |
| 16 | 1006_Titanic | 2201 | 3 | 0 | 0.00 | 1.57 | 0.00 | 0.00 |
| 17 | 1592_Diabetes-Data | 768 | 8 | 0 | 0.00 | 1.69 | 0.00 | 0.00 |
| 18 | 0424_autoPrice | 159 | 14 | 1 | 0.07 | 2.99 | 0.00 | 0.00 |
| 19 | audit_data | 776 | 23 | 3 | 0.13 | 5.29 | 0.00 | 0.00 |
| 20 | audit_risk | 776 | 23 | 3 | 0.13 | 5.29 | 0.00 | 0.00 |
| 21 | new_model | 400 | 10 | 3 | 0.30 | 5.83 | 0.00 | 0.00 |
| 22 | trial | 776 | 7 | 10 | 0.39 | 5.29 | 0.00 | 0.00 |
| 23 | 1333_ricci_vs | 118 | 3 | 2 | 0.67 | 3.66 | 0.00 | 0.00 |
| 24 | 1619_NBA-2k20-player | 439 | 4 | 10 | 2.50 | 1.51 | 0.01 | 0.01 |
| 25 | 1458_kdd_ipums_la_97 | 5188 | 17 | 3 | 0.18 | 4.73 | 0.03 | 0.01 |
| 26 | 1736_combined-wine | 6497 | 10 | 2 | 0.20 | 11.77 | 0.09 | 0.01 |
| 27 | 1759_Red–White-wine | 6497 | 10 | 2 | 0.20 | 11.77 | 0.09 | 0.01 |
| 28 | 1578_kdd_ipums_la_97 | 5188 | 17 | 3 | 0.18 | 5.10 | 0.05 | 0.01 |
| 29 | 0526_colleges_aaup | 1161 | 13 | 2 | 0.15 | 8.10 | 0.09 | 0.01 |
| 30 | 1408_national | 4908 | 6 | 10 | 1.67 | 7.25 | 0.10 | 0.01 |
| 31 | 0284_bank8FM | 8192 | 7 | 1 | 0.14 | 6.78 | 0.15 | 0.02 |
| 32 | Customer_Behaviour | 400 | 2 | 1 | 0.50 | 2.94 | 0.15 | 0.05 |
| 33 | 2306_electricity_seed | 2000 | 7 | 1 | 0.14 | 3.94 | 0.22 | 0.06 |
| 34 | 0546_analcatdata | 132 | 2 | 1 | 0.50 | 3.81 | 0.22 | 0.06 |
| 35 | 2304_electricity_seed | 2000 | 7 | 1 | 0.14 | 3.25 | 0.23 | 0.07 |
| 36 | 2308_electricity_seed | 2000 | 7 | 1 | 0.14 | 4.37 | 0.32 | 0.07 |
| 37 | 1461_heart-failure | 299 | 7 | 5 | 0.71 | 4.47 | 0.37 | 0.08 |
| 38 | 2392_airlines_seed_3 | 2000 | 3 | 4 | 1.33 | 0.45 | 0.04 | 0.09 |
| 39 | 2390_airlines_seed_1 | 2000 | 3 | 4 | 1.33 | 0.32 | 0.03 | 0.09 |
| 40 | 0124_analcatdata | 100 | 2 | 7 | 3.50 | 3.11 | 0.27 | 0.09 |
| 41 | 2305_electricity_seed | 2000 | 7 | 1 | 0.14 | 4.08 | 0.37 | 0.09 |
| 42 | 2389_airlines_seed_0 | 2000 | 3 | 4 | 1.33 | 0.34 | 0.04 | 0.11 |
| 43 | 2393_airlines_seed_4 | 2000 | 3 | 4 | 1.33 | 0.44 | 0.05 | 0.12 |
| 44 | 0541_plasma_retinol | 315 | 10 | 3 | 0.30 | 2.62 | 0.37 | 0.14 |
| 45 | NFL | 3477 | 10 | 7 | 0.70 | 1.69 | 0.25 | 0.15 |
| 46 | 1142_Sick_numeric | 3772 | 6 | 23 | 3.83 | 4.07 | 0.68 | 0.17 |
| 47 | 2703_compas-two | 4966 | 2 | 9 | 4.50 | 0.86 | 0.16 | 0.19 |
| 48 | 0885_compas-two | 5278 | 2 | 11 | 5.50 | 1.00 | 0.19 | 0.19 |
| 49 | 2391_airlines_seed_2 | 2000 | 3 | 4 | 1.33 | 0.20 | 0.04 | 0.19 |
| 50 | 0400_analcatdata | 4052 | 1 | 6 | 6.00 | 4.62 | 1.01 | 0.22 |
| 51 | 2621_sf-police | 2000 | 4 | 4 | 1.00 | 0.49 | 0.11 | 0.23 |
| 52 | b_depressed | 1429 | 12 | 9 | 0.75 | 0.89 | 0.25 | 0.28 |
| 53 | 2619_sf-police | 2000 | 4 | 4 | 1.00 | 0.83 | 0.23 | 0.28 |
| 54 | 2620_sf-police | 2000 | 4 | 4 | 1.00 | 0.99 | 0.34 | 0.34 |
| 55 | Breast_Cancer | 4024 | 5 | 10 | 2.00 | 2.15 | 0.81 | 0.37 |
| 56 | 1512_eye_movements | 7608 | 17 | 5 | 0.29 | 1.94 | 0.74 | 0.38 |
| 57 | 0472_analcatdata | 364 | 23 | 9 | 0.39 | 1.21 | 0.68 | 0.56 |
| 58 | 0555_socmob | 1156 | 1 | 4 | 4.00 | 2.69 | 1.51 | 0.56 |
| 59 | 2622_sf-police | 2000 | 4 | 4 | 1.00 | 0.55 | 0.36 | 0.66 |
| 60 | loan_train | 614 | 4 | 7 | 1.75 | 0.34 | 0.22 | 0.67 |
| 61 | TravelInsurancePrediction | 1987 | 1 | 7 | 7.00 | 0.78 | 0.56 | 0.72 |
| 62 | 1742_Loan | 614 | 4 | 7 | 1.75 | 0.60 | 0.48 | 0.79 |
| 63 | 1635_Is-this-a-good | 1723 | 4 | 9 | 2.25 | 0.62 | 0.56 | 0.89 |
| 64 | 0948_Ishwar | 2205 | 17 | 4 | 0.24 | 3.34 | 3.70 | 1.11 |
| 65 | piracydataset | 1423 | 1 | 3 | 3.00 | 0.18 | 0.27 | 1.47 |
| 66 | 1752_Wisconsin | 699 | 2 | 8 | 4.00 | 1.85 | 3.06 | 1.66 |
| 67 | 1413_shill-bidding | 6321 | 7 | 4 | 0.57 | 1.59 | 3.58 | 2.25 |
| 68 | Employee | 500 | 1 | 11 | 11.00 | 0.16 | 0.44 | 2.81 |
| 69 | 0446_newton_hema | 140 | 2 | 1 | 0.50 | 0.34 | 1.00 | 2.92 |
| 70 | Bank_Personal_Loan | 5000 | 6 | 6 | 1.00 | 0.23 | 0.68 | 2.93 |
| 71 | UniversalBank | 5000 | 6 | 6 | 1.00 | 0.23 | 0.68 | 2.93 |
| 72 | 1011_cleve | 303 | 5 | 8 | 1.60 | 0.84 | 2.76 | 3.29 |
| 73 | 1898_Personal-Loan | 5000 | 6 | 6 | 1.00 | 0.19 | 0.72 | 3.85 |
| 74 | 1564_Mammographic | 830 | 1 | 4 | 4.00 | 0.48 | 2.22 | 4.65 |
| 75 | 1692_Gender | 5001 | 1 | 6 | 6.00 | 0.55 | 6.31 | 11.57 |
| 76 | diabetes_data_upload | 520 | 1 | 15 | 15.00 | 0.24 | 5.48 | 22.64 |
| 77 | 1451_early-stage | 520 | 1 | 15 | 15.00 | 0.22 | 5.75 | 26.47 |
| 78 | 1774_Early-Stage | 520 | 1 | 15 | 15.00 | 0.22 | 5.75 | 26.47 |
| 79 | 0408_pharynx | 195 | 2 | 8 | 4.00 | 0.04 | 1.32 | 33.20 |

Table 8: AUC scores (the higher the better) of the baselines on 80 binary classification datasets.

| ID | XGB(d) | Cat(d) | FTT(d) | Trans(d) | XGB(t) | Cat(t) | MLP(t) | Auto(t) | DCN(t) | Tab(t) | SAI(t) | FTT(t) | XTab(t) | Ours$_j$(d) | Ours$_s$(d) |
|---|---|---|---|---|---|---|---|---|---|---|---|---|---|---|---|
| 0 | 1.0000 | 1.0000 | 1.0000 | 1.0000 | 1.0000 | 1.0000 | 0.9958 | 1.0000 | 1.0000 | 0.9999 | 0.9983 | 1.0000 | 0.9977 | 1.0000 | 1.0000 |
| 1 | 0.9955 | 0.9973 | 0.9981 | 0.8258 | 0.9981 | 0.9991 | 0.9976 | 0.9934 | 0.9951 | 0.9968 | 0.9945 | 0.9959 | 0.9849 | 0.9878 | 0.9964 |
| 2 | 0.9800 | 0.9810 | 0.9763 | 0.9763 | 0.9781 | 0.9790 | 0.9769 | 0.9756 | 0.9769 | 0.9702 | 0.9769 | 0.9779 | 0.9711 | 0.9688 | 0.9698 |
| 3 | 0.9857 | 0.9880 | 0.9854 | 0.6031 | 0.9856 | 0.9869 | 0.9869 | 0.9855 | 0.9859 | 0.9782 | 0.9851 | 0.9849 | 0.9797 | 0.9836 | 0.9811 |
| 4 | 0.9801 | 0.9806 | 0.9771 | 0.9787 | 0.9802 | 0.9803 | 0.9788 | 0.9777 | 0.9793 | 0.9759 | 0.9780 | 0.9785 | 0.9771 | 0.9769 | 0.9680 |
| 5 | 0.9379 | 0.9488 | 0.9496 | 0.7148 | 0.9427 | 0.9477 | 0.9470 | 0.9474 | 0.9437 | 0.9478 | 0.9508 | 0.9490 | 0.9496 | 0.9413 | 0.9426 |
| 6 | 0.6894 | 0.6439 | 0.6818 | 0.7121 | 0.6629 | 0.7083 | 0.6553 | 0.6288 | 0.6326 | 0.7879 | 0.8636 | 0.7955 | 0.7197 | 0.7045 | 0.6818 |
| 7 | 0.6238 | 0.6331 | 0.5882 | 0.5550 | 0.6327 | 0.5532 | 0.5986 | 0.5794 | 0.5302 | 0.5038 | 0.4478 | 0.5862 | 0.5458 | 0.5822 | 0.5886 |
| 8 | 0.9946 | 1.0000 | 0.9898 | 0.9429 | 0.9995 | 1.0000 | 0.9562 | 0.9626 | 0.9749 | 0.9501 | 0.9997 | 0.9985 | 0.7788 | 0.9954 | 0.9905 |
| 9 | 0.5351 | 0.5499 | 0.5395 | 0.5288 | 0.5588 | 0.5397 | 0.5343 | 0.5225 | 0.5377 | 0.5187 | 0.5356 | 0.5429 | 0.5214 | 0.4909 | 0.5510 |
| 10 | 0.8000 | 0.9009 | 0.7514 | 0.8860 | 0.8636 | 0.9009 | 0.6897 | 0.9028 | 0.6374 | 0.5495 | 0.5617 | 0.7626 | 0.8262 | 0.5477 | 0.7607 |
| 11 | 0.8333 | 0.7747 | 0.7682 | 0.8216 | 0.7435 | 0.8542 | 0.7650 | 0.7878 | 0.8333 | 0.7041 | 0.7266 | 0.7643 | 0.5798 | 0.7904 | 0.7806 |
| 12 | 0.8738 | 0.8912 | 0.9144 | 0.8981 | 0.9398 | 0.9062 | 0.8275 | 0.9815 | 0.8333 | 0.9537 | 0.8241 | 0.7708 | 0.8796 | 0.7593 | 0.9248 |
| 13 | 0.5112 | 0.4660 | 0.4777 | 0.4768 | 0.4578 | 0.4918 | 0.5220 | 0.5112 | 0.4778 | 0.4840 | 0.4866 | 0.5238 | 0.5026 | 0.4802 | 0.4966 |
| 14 | 0.9922 | 0.9942 | 0.9933 | 0.9947 | 0.9953 | 0.9945 | 0.9925 | 0.9912 | 0.9951 | 0.9906 | 0.9942 | 0.9932 | 0.9935 | 0.9863 | 0.9927 |
| 15 | 0.7458 | 0.7384 | 0.7232 | 0.5608 | 0.8093 | 0.7780 | 0.8552 | 0.7804 | 0.8424 | 0.6157 | 0.7254 | 0.7974 | 0.7878 | 0.7742 | 0.7615 |
| 16 | 0.7582 | 0.7510 | 0.7613 | 0.6336 | 0.7597 | 0.7520 | 0.7605 | 0.7451 | 0.7606 | 0.7170 | 0.7536 | 0.7528 | 0.7450 | 0.7609 | 0.7609 |
| 17 | 0.7672 | 0.7556 | 0.8031 | 0.3363 | 0.7778 | 0.8134 | 0.7887 | 0.7830 | 0.8002 | 0.7087 | 0.7894 | 0.7957 | 0.7841 | 0.7604 | 0.7913 |
| 18 | 0.9870 | 0.9827 | 0.9827 | 0.9957 | 0.9805 | 0.9957 | 0.8788 | 0.9654 | 0.9740 | 0.5390 | 0.9827 | 0.9913 | 0.9177 | 0.9913 | 0.9827 |
| 19 | 1.0000 | 1.0000 | 1.0000 | 0.9679 | 1.0000 | 0.9918 | 0.9991 | 0.9997 | 0.9995 | 0.9995 | 0.9995 | 1.0000 | 0.9848 | 0.9998 | 1.0000 |
| 20 | 1.0000 | 1.0000 | 1.0000 | 0.9638 | 1.0000 | 1.0000 | 0.9988 | 0.9997 | 0.9993 | 0.9862 | 0.9995 | 1.0000 | 0.9848 | 0.9998 | 1.0000 |
| 21 | 0.9997 | 0.9970 | 0.9840 | 0.9847 | 1.0000 | 0.9980 | 0.9687 | 0.9833 | 0.9753 | 0.9280 | 0.9807 | 0.9673 | 0.9773 | 0.9807 | 0.9933 |
| 22 | 1.0000 | 1.0000 | 0.9986 | 1.0000 | 1.0000 | 1.0000 | 0.9921 | 1.0000 | 0.9798 | 0.9901 | 0.9993 | 1.0000 | 1.0000 | 1.0000 | 1.0000 |
| 23 | 1.0000 | 1.0000 | 0.9860 | 1.0000 | 1.0000 | 1.0000 | 0.9091 | 0.8951 | 0.9930 | 0.8182 | 0.9930 | 0.9930 | 0.5664 | 1.0000 | 1.0000 |
| 24 | 0.7500 | 1.0000 | 0.9244 | 0.9186 | 1.0000 | 0.6047 | 0.8314 | 0.8488 | 0.8721 | 0.6105 | 0.8721 | 0.8779 | 0.8953 | 0.9767 | 0.9826 |
| 25 | 0.9421 | 0.9355 | 0.9346 | 0.3702 | 0.9441 | 0.9428 | 0.9437 | 0.9433 | 0.9432 | 0.9381 | 0.9427 | 0.9420 | 0.9400 | 0.9333 | 0.9332 |
| 26 | 0.9997 | 0.9986 | 0.9997 | 0.3612 | 0.9971 | 0.9995 | 0.9948 | 0.9992 | 0.9998 | 0.9974 | 0.9973 | 0.9950 | 0.9965 | 0.9921 | 0.9934 |
| 27 | 0.9997 | 0.9986 | 0.9997 | 0.6377 | 0.9966 | 0.9987 | 0.9974 | 0.9936 | 0.9997 | 0.9991 | 0.9978 | 0.9982 | 0.9965 | 0.9921 | 0.9934 |
| 28 | 0.9447 | 0.9521 | 0.9461 | 0.4090 | 0.9499 | 0.9535 | 0.9477 | 0.9501 | 0.9481 | 0.9310 | 0.9504 | 0.9524 | 0.9478 | 0.9478 | 0.9480 |
| 29 | 0.9989 | 0.9988 | 0.9999 | 0.5117 | 0.9989 | 0.9989 | 0.9982 | 0.9972 | 0.9999 | 0.9892 | 0.9996 | 0.9998 | 0.9947 | 0.9980 | 0.9982 |
| 30 | 0.9992 | 1.0000 | 1.0000 | 0.6133 | 1.0000 | 1.0000 | 0.9998 | 0.9999 | 0.9999 | 0.9994 | 1.0000 | 1.0000 | 1.0000 | 0.9991 | 1.0000 |
| 31 | 0.9871 | 0.9896 | 0.9915 | 0.7284 | 0.9897 | 0.9900 | 0.9900 | 0.9906 | 0.9900 | 0.9884 | 0.9910 | 0.9905 | 0.9908 | 0.9829 | 0.9817 |
| 32 | 0.9097 | 0.9043 | 0.9405 | 0.8864 | 0.9344 | 0.8905 | 0.9412 | 0.9209 | 0.9351 | 0.8627 | 0.9364 | 0.9459 | 0.9486 | 0.9317 | 0.9331 |
| 33 | 0.8896 | 0.8638 | 0.8653 | 0.8750 | 0.8817 | 0.8756 | 0.8447 | 0.8478 | 0.8409 | 0.8174 | 0.8736 | 0.8533 | 0.8258 | 0.8784 | 0.9007 |
| 34 | 0.9568 | 0.9907 | 0.9259 | 0.9753 | 0.9136 | 0.9105 | 0.9383 | 0.9506 | 0.8889 | 0.7469 | 0.8827 | 0.9321 | 0.9630 | 0.9198 | 0.9383 |
| 35 | 0.9048 | 0.9059 | 0.9033 | 0.5467 | 0.8953 | 0.9083 | 0.8733 | 0.8814 | 0.8764 | 0.8606 | 0.8944 | 0.8994 | 0.8431 | 0.8926 | 0.9088 |
| 36 | 0.8625 | 0.8682 | 0.8402 | 0.8474 | 0.8739 | 0.8717 | 0.8542 | 0.8440 | 0.8400 | 0.8266 | 0.8484 | 0.8305 | 0.8386 | 0.8695 | 0.8861 |
| 37 | 0.8440 | 0.8408 | 0.8588 | 0.8691 | 0.8383 | 0.8652 | 0.9076 | 0.8883 | 0.8909 | 0.6059 | 0.8691 | 0.8896 | 0.7343 | 0.8228 | 0.8691 |
| 38 | 0.5884 | 0.6124 | 0.5844 | 0.5935 | 0.6117 | 0.6036 | 0.5811 | 0.5676 | 0.5720 | 0.5644 | 0.5679 | 0.5868 | 0.5962 | 0.6826 | 0.6581 |
| 39 | 0.5332 | 0.5566 | 0.5621 | 0.5359 | 0.6241 | 0.5820 | 0.5359 | 0.5275 | 0.5508 | 0.5357 | 0.5298 | 0.5677 | 0.5522 | 0.6754 | 0.6791 |
| 40 | 0.9375 | 0.8802 | 0.8958 | 0.9479 | 0.9062 | 0.9427 | 0.9375 | 0.9062 | 0.9167 | 0.5365 | 0.9167 | 0.8958 | 0.5938 | 0.9062 | 0.8958 |
| 41 | 0.9066 | 0.9246 | 0.8934 | 0.5660 | 0.9076 | 0.9235 | 0.8915 | 0.9071 | 0.9017 | 0.8829 | 0.8989 | 0.8892 | 0.8824 | 0.8900 | 0.9268 |
| 42 | 0.5892 | 0.5849 | 0.5962 | 0.5770 | 0.6269 | 0.5815 | 0.4876 | 0.6031 | 0.5734 | 0.5798 | 0.5038 | 0.5827 | 0.5801 | 0.6324 | 0.6586 |
| 43 | 0.5224 | 0.5322 | 0.5615 | 0.5583 | 0.5984 | 0.5467 | 0.6129 | 0.5774 | 0.5797 | 0.5412 | 0.5644 | 0.5620 | 0.5621 | 0.6326 | 0.6379 |
| 44 | 0.4393 | 0.5278 | 0.4352 | 0.3796 | 0.4928 | 0.5149 | 0.5319 | 0.4630 | 0.5484 | 0.3858 | 0.4115 | 0.5473 | 0.3405 | 0.4712 | 0.6235 |
| 45 | 0.7199 | 0.7399 | 0.7263 | 0.5494 | 1.0000 | 0.7390 | 0.7392 | 0.7315 | 0.7395 | 0.6997 | 0.7323 | 0.7321 | 0.7247 | 0.9984 | 1.0000 |
| 46 | 0.9749 | 0.9981 | 0.9958 | 0.9895 | 0.9914 | 0.9985 | 0.9169 | 0.9910 | 0.9466 | 0.9389 | 0.9879 | 0.9916 | 0.9238 | 0.9941 | 0.9914 |
| 47 | 0.7327 | 0.7440 | 0.7486 | 0.3895 | 0.7438 | 0.7421 | 0.7480 | 0.7476 | 0.7454 | 0.7351 | 0.7465 | 0.7459 | 0.7362 | 0.7360 | 0.7417 |
| 48 | 0.7277 | 0.7465 | 0.7431 | 0.4310 | 0.7443 | 0.7423 | 0.7419 | 0.7442 | 0.7471 | 0.7202 | 0.7445 | 0.7420 | 0.7403 | 0.7371 | 0.7394 |
| 49 | 0.6119 | 0.6010 | 0.5831 | 0.5516 | 0.6833 | 0.6159 | 0.6005 | 0.6028 | 0.6008 | 0.5267 | 0.5983 | 0.5946 | 0.5699 | 0.6772 | 0.6904 |
| 50 | 0.9895 | 0.9826 | 0.9987 | 0.9984 | 0.9784 | 0.9925 | 0.9983 | 0.9941 | 0.9988 | 0.9981 | 0.9986 | 0.9988 | 0.9974 | 0.9879 | 0.9934 |
| 51 | 0.4981 | 0.5017 | 0.5462 | 0.4683 | 0.5198 | 0.5284 | 0.4549 | 0.5541 | 0.5934 | 0.4994 | 0.4391 | 0.5473 | 0.5629 | 0.4746 | 0.5879 |
| 52 | 0.4950 | 0.4454 | 0.4959 | 0.4417 | 0.4472 | 0.4775 | 0.4767 | 0.4699 | 0.5004 | 0.5194 | 0.5371 | 0.4529 | 0.4813 | 0.4796 | 0.5722 |
| 53 | 0.5554 | 0.5003 | 0.5252 | 0.5308 | 0.5090 | 0.5268 | 0.5216 | 0.5525 | 0.5257 | 0.5099 | 0.5102 | 0.5513 | 0.5690 | 0.4710 | 0.5565 |
| 54 | 0.5750 | 0.4937 | 0.5169 | 0.4915 | 0.4638 | 0.4635 | 0.4691 | 0.5360 | 0.4604 | 0.4976 | 0.4681 | 0.5109 | 0.5251 | 0.5274 | 0.5120 |
| 55 | 0.8442 | 0.8492 | 0.8534 | 0.5585 | 0.8539 | 0.8559 | 0.8478 | 0.8524 | 0.8507 | 0.8341 | 0.8359 | 0.8552 | 0.8532 | 0.8269 | 0.8500 |
| 56 | 0.6902 | 0.6711 | 0.6454 | 0.5343 | 0.7063 | 0.6714 | 0.6461 | 0.6402 | 0.6482 | 0.5782 | 0.6363 | 0.6504 | 0.6067 | 0.6605 | 0.6684 |
| 57 | 0.5330 | 0.4243 | 0.5183 | 0.5443 | 0.5852 | 0.5791 | 0.5087 | 0.4765 | 0.5426 | 0.6304 | 0.5522 | 0.5583 | 0.5617 | 0.5296 | 0.5496 |
| 58 | 0.9660 | 0.9700 | 0.9659 | 0.9749 | 0.9829 | 0.9724 | 0.9689 | 0.9695 | 0.9700 | 0.9634 | 0.9687 | 0.9607 | 0.9748 | 0.9660 | 0.9663 |
| 59 | 0.5465 | 0.4897 | 0.5107 | 0.5335 | 0.6006 | 0.5257 | 0.4395 | 0.5394 | 0.5513 | 0.4878 | 0.4994 | 0.5575 | 0.4985 | 0.5309 | 0.5524 |
| 60 | 0.4938 | 0.5557 | 0.5279 | 0.5452 | 0.7488 | 0.4689 | 0.5638 | 0.5619 | 0.5570 | 0.4949 | 0.5839 | 0.5285 | 0.4833 | 0.7452 | 0.7746 |
| 61 | 0.7901 | 0.7735 | 0.7714 | 0.6366 | 0.7538 | 0.7793 | 0.7891 | 0.7426 | 0.7699 | 0.6888 | 0.7913 | 0.7842 | 0.7664 | 0.8041 | 0.7749 |
| 62 | 0.5556 | 0.4751 | 0.5526 | 0.4780 | 0.7785 | 0.5559 | 0.6361 | 0.5656 | 0.5774 | 0.4913 | 0.5858 | 0.5889 | 0.5553 | 0.7517 | 0.7398 |
| 63 | 0.5738 | 0.6122 | 0.6419 | 0.4686 | 0.7312 | 0.6330 | 0.6010 | 0.6473 | 0.6187 | 0.4893 | 0.6345 | 0.6642 | 0.6382 | 0.7299 | 0.6957 |
| 64 | 0.9964 | 0.9961 | 0.9923 | 0.3536 | 0.9939 | 0.9936 | 0.9882 | 0.9892 | 0.9861 | 0.9774 | 0.9866 | 0.9867 | 0.9708 | 0.9938 | 0.9860 |
| 65 | 0.5359 | 0.5948 | 0.6022 | 0.6065 | 0.5429 | 0.5506 | 0.5948 | 0.6076 | 0.6361 | 0.5353 | 0.6215 | 0.6077 | 0.6092 | 0.6179 | 0.6363 |
| 66 | 0.9955 | 0.9946 | 0.9962 | 0.5170 | 0.9903 | 0.9975 | 0.9966 | 0.9952 | 0.9946 | 0.9665 | 0.9966 | 0.9982 | 0.9921 | 0.9857 | 0.9993 |
| 67 | 0.9996 | 0.9998 | 0.9982 | 0.2496 | 0.9924 | 0.9998 | 0.9994 | 0.9985 | 0.9991 | 0.9975 | 0.9985 | 0.9963 | 0.9966 | 0.9944 | 0.9965 |
| 68 | 0.4410 | 0.5181 | 0.4709 | 0.4035 | 0.4859 | 0.4599 | 0.4990 | 0.5452 | 0.4637 | 0.5873 | 0.4328 | 0.4456 | 0.5725 | 0.5516 | 0.5110 |
| 69 | 0.6429 | 0.6480 | 0.6276 | 0.6837 | 0.6378 | 0.6990 | 0.5204 | 0.6429 | 0.6429 | 0.5561 | 0.5204 | 0.5714 | 0.4847 | 0.6224 | 0.7908 |
| 70 | 0.5824 | 0.6047 | 0.6122 | 0.5235 | 0.5928 | 0.5999 | 0.5915 | 0.6056 | 0.5897 | 0.5733 | 0.6103 | 0.5967 | 0.5956 | 0.5446 | 0.6168 |
| 71 | 0.5824 | 0.6047 | 0.6122 | 0.4710 | 0.6074 | 0.6038 | 0.6127 | 0.5859 | 0.5799 | 0.5750 | 0.5879 | 0.6152 | 0.5956 | 0.5446 | 0.6168 |
| 72 | 0.8290 | 0.8474 | 0.8160 | 0.8398 | 0.7668 | 0.8139 | 0.8409 | 0.8561 | 0.8398 | 0.6742 | 0.8539 | 0.8593 | 0.8182 | 0.8658 | 0.8939 |
| 73 | 0.6363 | 0.6324 | 0.6348 | 0.4712 | 0.6107 | 0.6183 | 0.6421 | 0.6200 | 0.6121 | 0.5887 | 0.6074 | 0.6384 | 0.6179 | 0.6075 | 0.6334 |
| 74 | 0.8780 | 0.8833 | 0.8887 | 0.8718 | 0.8792 | 0.8930 | 0.7521 | 0.8833 | 0.8776 | 0.8417 | 0.8784 | 0.8791 | 0.8630 | 0.8889 | 0.8813 |
| 75 | 0.9978 | 0.9984 | 0.9961 | 0.1167 | 0.9972 | 0.9980 | 0.9964 | 0.9951 | 0.9959 | 0.9950 | 0.9958 | 0.9959 | 0.9960 | 0.9972 | 0.9963 |
| 76 | 0.9973 | 0.9984 | 0.9984 | 0.2641 | 0.9902 | 0.9941 | 0.9891 | 0.9973 | 0.9785 | 0.9031 | 0.9988 | 0.9988 | 0.9742 | 0.9922 | 0.9973 |
| 77 | 0.9842 | 0.9924 | 0.9984 | 0.7633 | 0.9906 | 0.9912 | 0.9723 | 0.9668 | 0.9621 | 0.9676 | 0.9902 | 0.9840 | 0.9828 | 0.9844 | 0.9906 |
| 78 | 0.9842 | 0.9924 | 0.9984 | 0.9859 | 0.9891 | 0.9988 | 0.9777 | 0.9902 | 0.9773 | 0.8660 | 0.9965 | 0.9863 | 0.9828 | 0.9844 | 0.9906 |
| 79 | 0.7667 | 0.7306 | 0.7778 | 0.7639 | 0.7819 | 0.7639 | 0.7778 | 0.6972 | 0.8028 | 0.5306 | 0.6972 | 0.7556 | 0.5194 | 0.8806 | 0.8306 |

Table 9: Statistics of the 65 downstream regression datasets.

| ID | Dataset name | # samples | # num. | # cat. | $\alpha$ | Num. Shapely sum | Cat. Shapely sum | $\beta$ |
|---|---|---|---|---|---|---|---|---|
| 0 | my_csv-3-10-2022-10-35 | 10 | 7 | 1 | 0.14 | 0.95 | 0.00 | 0.00 |
| 1 | 0237_arsenic-female | 559 | 4 | 0 | 0.00 | 0.20 | 0.00 | 0.00 |
| 2 | 0251_arsenic-male | 559 | 4 | 0 | 0.00 | 0.19 | 0.00 | 0.00 |
| 3 | 0258_no2 | 500 | 7 | 0 | 0.00 | 1.46 | 0.00 | 0.00 |
| 4 | 0259_strikes | 625 | 6 | 0 | 0.00 | 0.34 | 0.00 | 0.00 |
| 5 | ph-data | 653 | 3 | 0 | 0.00 | 1.07 | 0.00 | 0.00 |
| 6 | 0925_Concrete_Data | 1030 | 8 | 0 | 0.00 | 1.57 | 0.00 | 0.00 |
| 7 | 1065_hungarian | 522 | 19 | 0 | 0.00 | 1.21 | 0.00 | 0.00 |
| 8 | 1260_optical | 640 | 5 | 4 | 0.80 | 0.23 | 0.00 | 0.00 |
| 9 | 1331_dataset_time_9 | 2178 | 3 | 0 | 0.00 | 0.17 | 0.00 | 0.00 |
| 10 | 1464_dow-jones-index | 750 | 6 | 9 | 1.50 | 0.63 | 0.00 | 0.00 |
| 11 | 1564_Concrete | 1005 | 8 | 0 | 0.00 | 1.56 | 0.00 | 0.00 |
| 12 | 1623_GameStop | 4773 | 5 | 0 | 0.00 | 0.90 | 0.00 | 0.00 |
| 13 | 1624_Alcohol | 1549 | 5 | 1 | 0.20 | 1.07 | 0.00 | 0.00 |
| 14 | 1769_Facebook | 2076 | 5 | 0 | 0.00 | 2.07 | 0.00 | 0.00 |
| 15 | 1845_Predict-Amazon | 349 | 6 | 0 | 0.00 | 0.48 | 0.00 | 0.00 |
| 16 | 1869_Bitcoin-Stock | 2397 | 5 | 0 | 0.00 | 0.98 | 0.00 | 0.00 |
| 17 | 1874_Goodreads | 1234 | 5 | 3 | 0.60 | 0.33 | 0.00 | 0.00 |
| 18 | 1901_Netflix-10-Year | 4581 | 5 | 0 | 0.00 | 3.52 | 0.00 | 0.00 |
| 19 | 2644_concrete | 1030 | 8 | 0 | 0.00 | 1.57 | 0.00 | 0.00 |
| 20 | User Knowledge | 403 | 4 | 0 | 0.00 | 0.85 | 0.00 | 0.00 |
| 21 | wines_SPA | 6331 | 4 | 6 | 1.50 | 0.38 | 0.00 | 0.00 |
| 22 | World Population Live | 234 | 11 | 3 | 0.27 | 0.89 | 0.00 | 0.00 |
| 23 | 1222_premier_league | 2565 | 19 | 0 | 0.00 | 0.73 | 0.00 | 0.00 |
| 24 | real_estate_listings | 4942 | 7 | 2 | 0.29 | 1.66 | 0.00 | 0.00 |
| 25 | 1228_Premier_League | 2961 | 13 | 3 | 0.23 | 0.94 | 0.01 | 0.02 |
| 26 | 1594_Spotify—All-Time | 1994 | 9 | 4 | 0.44 | 0.59 | 0.01 | 0.02 |
| 27 | 1878_COVID | 2580 | 2 | 3 | 1.50 | 0.42 | 0.01 | 0.03 |
| 28 | 1848_Minneapolis-Air | 4790 | 4 | 12 | 3.00 | 0.84 | 0.02 | 0.03 |
| 29 | 1712_Running-Log | 689 | 1 | 15 | 15.00 | 0.74 | 0.02 | 0.03 |
| 30 | 1900_Another-Dataset | 1538 | 4 | 3 | 0.75 | 1.02 | 0.04 | 0.04 |
| 31 | 0988_test_data | 200 | 10 | 1 | 0.10 | 0.99 | 0.04 | 0.04 |
| 32 | 0274_kdd_coil_3 | 316 | 8 | 3 | 0.38 | 0.83 | 0.04 | 0.05 |
| 33 | 0235_plasma_retinol | 315 | 10 | 3 | 0.30 | 1.09 | 0.08 | 0.07 |
| 34 | 0272_kdd_coil_1 | 316 | 8 | 3 | 0.38 | 0.86 | 0.08 | 0.10 |
| 35 | 0279_kdd_coil_5 | 316 | 8 | 3 | 0.38 | 0.42 | 0.05 | 0.12 |
| 36 | 0364_sleuth_case2002 | 147 | 2 | 4 | 2.00 | 0.44 | 0.06 | 0.14 |
| 37 | 0117_fruitfly | 125 | 2 | 2 | 1.00 | 0.18 | 0.03 | 0.16 |
| 38 | 0273_kdd_coil_2 | 316 | 8 | 3 | 0.38 | 0.75 | 0.13 | 0.17 |
| 39 | 1755_Detailed | 148 | 5 | 8 | 1.60 | 0.14 | 0.02 | 0.17 |
| 40 | 1417_ibm-employee | 1470 | 11 | 21 | 1.91 | 1.02 | 0.18 | 0.17 |
| 41 | 1118_jura | 359 | 8 | 9 | 1.12 | 0.88 | 0.18 | 0.20 |
| 42 | 1266_CSM | 196 | 10 | 2 | 0.20 | 0.66 | 0.14 | 0.21 |
| 43 | 0911_forest_fires | 517 | 8 | 4 | 0.50 | 0.48 | 0.11 | 0.24 |
| 44 | 1528 | 1197 | 6 | 8 | 1.33 | 1.14 | 0.30 | 0.27 |
| 45 | 1449_garments-worker | 1197 | 6 | 8 | 1.33 | 1.14 | 0.30 | 0.27 |
| 46 | 0907_UCI-student | 395 | 3 | 29 | 9.67 | 0.90 | 0.24 | 0.27 |
| 47 | 1267_autoMpg | 392 | 4 | 1 | 0.25 | 0.77 | 0.21 | 0.27 |
| 48 | 1787_Lisbon-House | 246 | 5 | 10 | 2.00 | 0.78 | 0.22 | 0.28 |
| 49 | 0149_socmob | 1156 | 1 | 4 | 4.00 | 0.47 | 0.13 | 0.28 |
| 50 | 1640_Calculate | 1030 | 7 | 1 | 0.14 | 1.20 | 0.37 | 0.31 |
| 51 | mechanical_analysis | 927 | 7 | 3 | 0.43 | 0.96 | 0.39 | 0.40 |
| 52 | 1890_ECDC-daily-data | 9370 | 3 | 6 | 2.00 | 0.45 | 0.27 | 0.58 |
| 53 | thyroidDF | 9172 | 7 | 23 | 3.29 | 0.45 | 0.33 | 0.72 |
| 54 | 0261_analcatdata | 108 | 1 | 2 | 2.00 | 0.46 | 0.36 | 0.78 |
| 55 | 2168_Intersectional | 1000 | 13 | 5 | 0.38 | 0.68 | 0.55 | 0.80 |
| 56 | 0130_breastTumor | 286 | 2 | 7 | 3.50 | 0.28 | 0.30 | 1.07 |
| 57 | 1616_myiris | 150 | 2 | 2 | 1.00 | 0.38 | 0.63 | 1.67 |
| 58 | 0226_analcatdata | 468 | 1 | 2 | 2.00 | 0.31 | 0.59 | 1.95 |
| 59 | 1660_Swiss-banknote | 200 | 5 | 1 | 0.20 | 0.44 | 0.89 | 2.00 |
| 60 | 0225_veteran | 137 | 2 | 5 | 2.50 | 0.08 | 0.20 | 2.68 |
| 61 | 1591_Superstore | 9800 | 1 | 15 | 15.00 | 0.06 | 0.25 | 4.19 |
| 62 | 0125_pharynx | 195 | 2 | 8 | 4.00 | 0.14 | 0.93 | 6.76 |
| 63 | 1872_Forest-Surfaces | 8111 | 1 | 2 | 2.00 | 0.03 | 0.21 | 6.83 |
| 64 | 0211_analcatdata | 365 | 1 | 2 | 2.00 | 0.07 | 0.88 | 12.12 |

Table 10: RMSE scores (the lower the better) of the baselines on the 65 regression datasets.

| ID | XGB(d) | Cat(d) | FTT(d) | Trans(d) | XGB(t) | Cat(t) | MLP(t) | Auto(t) | DCN(t) | Tab(t) | SAI(t) | FTT(t) | XTab(t) | $\text{Ours}_j$(d) | $\text{Ours}_s$(d) |
|---|---|---|---|---|---|---|---|---|---|---|---|---|---|---|---|
| 0 | 0.2767 | 0.6728 | 2.2553 | 2.6559 | 0.1703 | 0.7471 | 27.4815 | 3.1905 | 49.2778 | 63.9863 | 2.3226 | 7.0454 | 4.5402 | 0.9395 | 0.2629 |
| 1 | 0.0437 | 0.0210 | 0.0326 | 0.0638 | 0.0469 | 0.0178 | 0.0159 | 0.0121 | 0.0438 | 0.0600 | 0.0402 | 0.0356 | 0.0551 | 0.0239 | 0.0087 |
| 2 | 0.0485 | 0.0230 | 0.0402 | 0.0640 | 0.0293 | 0.0262 | 0.0732 | 0.0292 | 0.0420 | 0.0508 | 0.0540 | 0.0418 | 0.0580 | 0.0095 | 0.0124 |
| 3 | 0.1045 | 0.1003 | 0.1141 | 0.1384 | 0.0966 | 0.1045 | 0.1101 | 0.1085 | 0.1076 | 0.1482 | 0.1098 | 0.1081 | 0.1376 | 0.1021 | 0.0987 |
| 4 | 0.0650 | 0.0559 | 0.0601 | 0.0615 | 0.0568 | 0.0550 | 0.0569 | 0.0587 | 0.0635 | 0.0933 | 0.0601 | 0.0561 | 0.0634 | 0.0662 | 0.0602 |
| 5 | 0.0938 | 0.0843 | 0.0633 | 0.2166 | 0.0757 | 0.0819 | 0.0692 | 0.0683 | 0.0718 | 0.1657 | 0.0679 | 0.0694 | 0.1839 | 0.0717 | 0.0654 |
| 6 | 0.0633 | 0.0645 | 0.0720 | 0.1581 | 0.0583 | 0.0619 | 0.0761 | 0.0695 | 0.0839 | 0.0865 | 0.0775 | 0.0756 | 0.1192 | 0.0604 | 0.0623 |
| 7 | 0.0883 | 0.0793 | 0.0897 | 0.0815 | 0.0738 | 0.0798 | 0.0783 | 0.0815 | 0.0801 | 0.0784 | 0.0772 | 0.0911 | 0.0861 | 0.1109 | 0.0860 |
| 8 | 0.0129 | 0.0103 | 0.0106 | 0.0317 | 0.0099 | 0.0166 | 0.0090 | 0.0135 | 0.0092 | 0.0337 | 0.0421 | 0.0077 | 0.0387 | 0.0221 | 0.0101 |
| 9 | 0.1964 | 0.1963 | 0.1934 | 0.1930 | 0.1944 | 0.1966 | 0.1935 | 0.1937 | 0.1944 | 0.1953 | 0.1937 | 0.1946 | 0.1985 | 0.1946 | 0.1919 |
| 10 | 0.1829 | 0.1872 | 0.1930 | 0.1979 | 0.0256 | 0.1996 | 0.1937 | 0.2054 | 0.1967 | 0.2321 | 0.2005 | 0.2094 | 0.2006 | 0.0284 | 0.0258 |
| 11 | 0.0543 | 0.0533 | 0.0525 | 0.1517 | 0.0522 | 0.0501 | 0.0534 | 0.0724 | 0.0581 | 0.0815 | 0.0616 | 0.0670 | 0.1024 | 0.0652 | 0.0528 |
| 12 | 0.0036 | 0.0048 | 0.0109 | 0.0246 | 0.0056 | 0.0065 | 0.0127 | 0.0147 | 0.0048 | 0.0047 | 0.0110 | 0.0106 | 0.0122 | 0.0125 | 0.0086 |
| 13 | 0.0842 | 0.0815 | 0.0767 | 0.1392 | 0.0504 | 0.0807 | 0.0789 | 0.0798 | 0.0809 | 0.0916 | 0.0885 | 0.0824 | 0.0929 | 0.0516 | 0.0519 |
| 14 | 0.0599 | 0.0586 | 0.0600 | 0.0623 | 0.0603 | 0.0585 | 0.0578 | 0.0592 | 0.0588 | 0.0557 | 0.0599 | 0.0605 | 0.0608 | 0.0612 | 0.0596 |
| 15 | 0.1579 | 0.1512 | 0.1614 | 0.1345 | 0.1577 | 0.1598 | 0.1405 | 0.1407 | 0.1579 | 0.2415 | 0.1330 | 0.1615 | 0.1505 | 0.1675 | 0.1412 |
| 16 | 0.0446 | 0.0457 | 0.0454 | 0.0612 | 0.0450 | 0.0451 | 0.0485 | 0.0454 | 0.0456 | 0.0456 | 0.0476 | 0.0458 | 0.0492 | 0.0441 | 0.0450 |
| 17 | 0.0444 | 0.0551 | 0.0499 | 0.0617 | 0.0453 | 0.0444 | 0.0505 | 0.0595 | 0.0576 | 0.0228 | 0.0635 | 0.0562 | 0.0580 | 0.0500 | 0.0418 |
| 18 | 0.0449 | 0.0461 | 0.0464 | 0.0490 | 0.0447 | 0.0443 | 0.0464 | 0.0460 | 0.0430 | 0.0399 | 0.0479 | 0.0473 | 0.0466 | 0.0469 | 0.0466 |
| 19 | 0.0647 | 0.0663 | 0.0741 | 0.1567 | 0.0636 | 0.0606 | 0.0709 | 0.0722 | 0.1010 | 0.0903 | 0.0788 | 0.0807 | 0.1191 | 0.0607 | 0.0603 |
| 20 | 0.1997 | 0.2029 | 0.2453 | 0.2619 | 0.1955 | 0.2330 | 0.2584 | 0.2631 | 0.2747 | 0.3839 | 0.2695 | 0.2450 | 0.3013 | 0.2358 | 0.2236 |
| 21 | 0.1877 | 0.1851 | 0.1908 | 0.2548 | 0.0000 | 0.1852 | 0.1880 | 0.1912 | 0.1864 | 0.1879 | 0.1898 | 0.1914 | 0.2348 | 0.0022 | 0.0062 |
| 22 | 0.0116 | 0.0188 | 0.0276 | 0.2094 | 0.0067 | 0.0177 | 0.0381 | 0.0534 | 0.0379 | 0.2869 | 0.0668 | 0.0395 | 0.3815 | 0.0291 | 0.0201 |
| 23 | 0.8140 | 0.7873 | 0.7922 | 0.8318 | 0.7909 | 0.7801 | 0.8105 | 0.8053 | 0.8007 | 0.8305 | 0.7969 | 0.7871 | 0.7897 | 0.8205 | 0.8049 |
| 24 | 0.0085 | 0.0113 | 0.0159 | 0.0359 | 0.0081 | 0.0120 | 0.0275 | 0.0118 | 0.0384 | 0.0172 | 0.0042 | 0.0053 | 0.0270 | 0.0234 | 0.0138 |
| 25 | 0.1126 | 0.1134 | 0.1169 | 0.1362 | 0.1096 | 0.1128 | 0.1212 | 0.1208 | 0.1212 | 0.1290 | 0.1201 | 0.1165 | 0.1236 | 0.1223 | 0.1154 |
| 26 | 0.1493 | 0.1459 | 0.1435 | 0.1530 | 0.1197 | 0.1519 | 0.1429 | 0.1447 | 0.1457 | 0.1527 | 0.1452 | 0.1445 | 0.1521 | 0.1401 | 0.1416 |
| 27 | 0.0255 | 0.0292 | 0.0271 | 0.0491 | 0.0076 | 0.0315 | 0.0290 | 0.0256 | 0.0327 | 0.0403 | 0.0282 | 0.0276 | 0.0372 | 0.0098 | 0.0091 |
| 28 | 0.0039 | 0.0065 | 0.0057 | 0.1089 | 0.0001 | 0.0088 | 0.0331 | 0.0266 | 0.0512 | 0.0762 | 0.0560 | 0.0129 | 0.0998 | 0.0078 | 0.0034 |
| 29 | 0.0368 | 0.0383 | 0.0763 | 0.1682 | 0.0173 | 0.0292 | 0.0865 | 0.0778 | 0.1080 | 0.0946 | 0.0803 | 0.0762 | 0.1125 | 0.0222 | 0.0268 |
| 30 | 0.0944 | 0.0931 | 0.0972 | 0.1924 | 0.0891 | 0.0938 | 0.0979 | 0.0966 | 0.1010 | 0.1024 | 0.0925 | 0.0945 | 0.1066 | 0.0964 | 0.0973 |
| 31 | 0.1123 | 0.1369 | 0.1484 | 0.1681 | 0.1435 | 0.1281 | 0.1382 | 0.1513 | 0.1341 | 0.3639 | 0.1392 | 0.1542 | 0.2728 | 0.1286 | 0.1457 |
| 32 | 0.1875 | 0.1706 | 0.1744 | 0.1742 | 0.1751 | 0.1717 | 0.1809 | 0.1686 | 0.1711 | 0.2828 | 0.1689 | 0.1766 | 0.1942 | 0.1882 | 0.1759 |
| 33 | 0.1661 | 0.1376 | 0.1341 | 0.1322 | 0.1374 | 0.1415 | 0.1214 | 0.1428 | 0.1264 | 0.5017 | 0.1280 | 0.1279 | 0.1277 | 0.1328 | 0.1219 |
| 34 | 0.2195 | 0.2013 | 0.2080 | 0.2103 | 0.2011 | 0.2007 | 0.2074 | 0.2088 | 0.2203 | 0.4179 | 0.2053 | 0.2100 | 0.2151 | 0.1942 | 0.2051 |
| 35 | 0.1111 | 0.0899 | 0.0899 | 0.1047 | 0.0929 | 0.0861 | 0.0896 | 0.0895 | 0.0919 | 0.6213 | 0.0929 | 0.0957 | 0.1249 | 0.0886 | 0.0800 |
| 36 | 0.1617 | 0.1259 | 0.1313 | 0.1621 | 0.1541 | 0.1438 | 0.1431 | 0.1602 | 0.1411 | 1.3354 | 0.1253 | 0.1411 | 0.1759 | 0.1402 | 0.1314 |
| 37 | 0.2394 | 0.2299 | 0.2490 | 0.2217 | 0.2324 | 0.2211 | 0.2271 | 0.2317 | 0.2134 | 0.5368 | 0.2154 | 0.2308 | 0.2676 | 0.2171 | 0.2143 |
| 38 | 0.1372 | 0.1265 | 0.1333 | 0.1414 | 0.1259 | 0.1352 | 0.1328 | 0.1382 | 0.1328 | 0.4135 | 0.1348 | 0.1430 | 0.1387 | 0.1392 | 0.1238 |
| 39 | 0.1797 | 0.1785 | 0.1759 | 0.1797 | 0.1757 | 0.1816 | 0.1853 | 0.1807 | 0.1796 | 0.2113 | 0.1782 | 0.1781 | 0.1828 | 0.1740 | 0.1765 |
| 40 | 0.1161 | 0.1092 | 0.1204 | 0.1753 | 0.1126 | 0.1077 | 0.1319 | 0.1234 | 0.1291 | 0.1255 | 0.1374 | 0.1178 | 0.1346 | 0.1074 | 0.1133 |
| 41 | 0.0820 | 0.0860 | 0.1074 | 0.1286 | 0.0779 | 0.0759 | 0.0970 | 0.0872 | 0.1175 | 0.3678 | 0.0901 | 0.0958 | 0.1544 | 0.1232 | 0.0839 |
| 42 | 0.1703 | 0.1688 | 0.1845 | 0.1687 | 0.1992 | 0.1698 | 0.1768 | 0.1738 | 0.1893 | 0.4934 | 0.1632 | 0.1725 | 0.2052 | 0.1765 | 0.1766 |
| 43 | 0.0644 | 0.0352 | 0.0359 | 0.0340 | 0.0414 | 0.0359 | 0.0367 | 0.0381 | 0.0403 | 0.0398 | 0.0332 | 0.0357 | 0.0352 | 0.0480 | 0.0349 |
| 44 | 0.1624 | 0.1648 | 0.1849 | 0.1887 | 0.1598 | 0.1580 | 0.1796 | 0.1707 | 0.1715 | 0.1773 | 0.1709 | 0.1708 | 0.1736 | 0.1779 | 0.1703 |
| 45 | 0.1624 | 0.1617 | 0.1823 | 0.1905 | 0.1606 | 0.1587 | 0.1732 | 0.1780 | 0.1705 | 0.1748 | 0.1680 | 0.1753 | 0.1736 | 0.1797 | 0.1770 |
| 46 | 0.0916 | 0.0868 | 0.1171 | 0.2099 | 0.0854 | 0.0935 | 0.2327 | 0.1063 | 0.1129 | 0.6702 | 0.1000 | 0.0993 | 0.2362 | 0.1064 | 0.0920 |
| 47 | 0.0733 | 0.0688 | 0.0729 | 0.1690 | 0.0743 | 0.0636 | 0.0852 | 0.0890 | 0.0850 | 0.6521 | 0.0761 | 0.0753 | 0.1715 | 0.0673 | 0.0573 |
| 48 | 0.0426 | 0.0341 | 0.0401 | 0.0757 | 0.0335 | 0.0363 | 0.0426 | 0.0376 | 0.0414 | 0.1592 | 0.0384 | 0.0398 | 0.1718 | 0.0322 | 0.0370 |
| 49 | 0.0450 | 0.0506 | 0.0433 | 0.0791 | 0.0275 | 0.0488 | 0.0550 | 0.0539 | 0.0490 | 0.0554 | 0.0542 | 0.0584 | 0.0748 | 0.0388 | 0.0424 |
| 50 | 0.0633 | 0.0645 | 0.0753 | 0.1614 | 0.0709 | 0.0613 | 0.0825 | 0.0815 | 0.0812 | 0.0886 | 0.0830 | 0.0834 | 0.1517 | 0.0649 | 0.0654 |
| 51 | 0.0039 | 0.1243 | 0.0307 | 1.6139 | 0.0287 | 0.0216 | 0.2794 | 0.3480 | 0.2718 | 0.4515 | 0.1540 | 0.0550 | 1.7042 | 0.1288 | 0.0130 |
| 52 | 0.0226 | 0.0203 | 0.1333 | 0.1496 | 0.0006 | 0.0137 | 0.1360 | 0.1362 | 0.1367 | 0.1387 | 0.1349 | 0.1350 | 0.1413 | 0.0040 | 0.0033 |
| 53 | 0.2271 | 0.2252 | 0.2396 | 0.2509 | 0.2226 | 0.2255 | 0.2454 | 0.2406 | 0.2452 | 0.2449 | 0.2399 | 0.2399 | 0.2463 | 0.2267 | 0.2310 |
| 54 | 0.1869 | 0.1830 | 0.2006 | 0.2109 | 0.1860 | 0.1765 | 0.2039 | 0.2036 | 0.1969 | 0.4114 | 0.1936 | 0.2350 | 0.2492 | 0.1827 | 0.1753 |
| 55 | 0.2009 | 0.1874 | 0.1881 | 0.2098 | 0.1863 | 0.1887 | 0.1883 | 0.1917 | 0.1852 | 0.2159 | 0.1894 | 0.1858 | 0.1884 | 0.1878 | 0.1825 |
| 56 | 0.2165 | 0.1997 | 0.1848 | 0.2058 | 0.1931 | 0.2070 | 0.1770 | 0.1876 | 0.1825 | 0.8456 | 0.1792 | 0.1780 | 0.1961 | 0.1847 | 0.1722 |
| 57 | 0.1328 | 0.1093 | 0.1109 | 0.2342 | 0.1176 | 0.1091 | 0.1102 | 0.1070 | 0.1134 | 1.1641 | 0.1028 | 0.1075 | 0.4013 | 0.1070 | 0.1074 |
| 58 | 0.0797 | 0.0861 | 0.1018 | 0.1726 | 0.0821 | 0.0828 | 0.0904 | 0.0873 | 0.0883 | 0.1127 | 0.0923 | 0.0850 | 0.1843 | 0.1020 | 0.0845 |
| 59 | 0.1146 | 0.1056 | 0.1198 | 0.1772 | 0.1073 | 0.1095 | 0.1107 | 0.1339 | 0.1185 | 0.6438 | 0.1195 | 0.1257 | 0.2053 | 0.1371 | 0.1134 |
| 60 | 0.1412 | 0.1102 | 0.1115 | 0.1119 | 0.1108 | 0.1083 | 0.1109 | 0.1073 | 0.1110 | 0.2060 | 0.1116 | 0.1106 | 0.1909 | 0.1101 | 0.1117 |
| 61 | 0.0290 | 0.0279 | 0.0302 | 0.0306 | 0.0194 | 0.0281 | 0.0302 | 0.0301 | 0.0302 | 0.0304 | 0.0301 | 0.0302 | 0.0302 | 0.0201 | 0.0283 |
| 62 | 0.2379 | 0.2117 | 0.2059 | 0.2656 | 0.1957 | 0.2078 | 0.1921 | 0.2054 | 0.2015 | 0.3927 | 0.1880 | 0.1976 | 0.2418 | 0.1778 | 0.1815 |
| 63 | 0.0974 | 0.0959 | 0.0957 | 0.0985 | 0.0005 | 0.0958 | 0.0957 | 0.0961 | 0.0956 | 0.0965 | 0.0956 | 0.0957 | 0.0967 | 0.0033 | 0.0029 |
| 64 | 0.1295 | 0.1128 | 0.1174 | 0.2165 | 0.1118 | 0.1191 | 0.1098 | 0.1176 | 0.1089 | 0.5817 | 0.1180 | 0.1110 | 0.2256 | 0.1256 | 0.1106 |

Table 11: Detailed AUC scores (the higher the better) of key ablations, design comparison, and transferability evaluations on 80 binary classification datasets. "nbin32/128" means the group "$n_{\text{bin}} = 32/128$" in Table 5; "w/ VPos" is for the ablation "w/ Value Vector Position Encoding" in Table 5; "rand. init." and "LM init." denote the non-pre-trained TP-BERTa initialized with random weights or RoBERTa weights in Table 3, respectively.

| ID | $\text{Ours}_s$ | Value2Str | VMFE | w/o IFA | nbin32 | nbin128 | w/ VPos | rand. init. | LM init. |
|---|---|---|---|---|---|---|---|---|---|
| 0 | 1.0000 | 0.4772 | 1.0000 | 0.9927 | 0.9960 | 0.9998 | 1.0000 | 0.9996 | 1.0000 |
| 1 | 0.9964 | 0.9947 | 0.9927 | 0.9908 | 0.9961 | 0.9989 | 0.9992 | 0.9948 | 0.9964 |
| 2 | 0.9698 | 0.9476 | 0.9758 | 0.9641 | 0.9720 | 0.9747 | 0.9716 | 0.9711 | 0.9698 |
| 3 | 0.9811 | 0.9603 | 0.9850 | 0.9705 | 0.9831 | 0.9809 | 0.9829 | 0.9789 | 0.9811 |
| 4 | 0.9680 | 0.9486 | 0.9759 | 0.9684 | 0.9723 | 0.9748 | 0.9727 | 0.9767 | 0.9680 |
| 5 | 0.9426 | 0.9233 | 0.9436 | 0.9273 | 0.9453 | 0.9468 | 0.9476 | 0.9263 | 0.9426 |
| 6 | 0.6818 | 0.3182 | 0.6364 | 0.7955 | 0.7500 | 0.7209 | 0.7348 | 0.6770 | 0.6818 |
| 7 | 0.5886 | 0.5670 | 0.4506 | 0.3345 | 0.3569 | 0.5882 | 0.3645 | 0.5490 | 0.5886 |
| 8 | 0.9905 | 0.9959 | 0.9826 | 0.9864 | 1.0000 | 1.0000 | 0.9980 | 0.9690 | 0.9905 |
| 9 | 0.5510 | 0.5177 | 0.5266 | 0.4991 | 0.4993 | 0.5129 | 0.5102 | 0.5322 | 0.5510 |
| 10 | 0.7607 | 0.9103 | 0.7121 | 0.7944 | 0.7196 | 0.7196 | 0.7252 | 0.7032 | 0.7607 |
| 11 | 0.7806 | 0.7148 | 0.8652 | 0.7617 | 0.7415 | 0.8756 | 0.8639 | 0.8568 | 0.7806 |
| 12 | 0.9248 | 0.8588 | 1.0000 | 0.7500 | 0.8333 | 0.7870 | 0.7778 | 0.7639 | 0.9248 |
| 13 | 0.4966 | 0.5220 | 0.4696 | 0.4848 | 0.4790 | 0.5041 | 0.4982 | 0.4927 | 0.4966 |
| 14 | 0.9927 | 0.8353 | 0.9890 | 0.9716 | 0.9903 | 0.9854 | 0.9893 | 0.9850 | 0.9927 |
| 15 | 0.7615 | 0.7744 | 0.8039 | 0.7576 | 0.7321 | 0.7857 | 0.8199 | 0.7310 | 0.7615 |
| 16 | 0.7609 | 0.6946 | 0.7406 | 0.7522 | 0.7344 | 0.7432 | 0.7042 | 0.7546 | 0.7609 |
| 17 | 0.7913 | 0.4376 | 0.8013 | 0.7593 | 0.7869 | 0.7978 | 0.8019 | 0.8094 | 0.7913 |
| 18 | 0.9827 | 0.7446 | 0.9221 | 0.9134 | 0.9524 | 0.9670 | 0.9784 | 0.9957 | 0.9827 |
| 19 | 1.0000 | 0.9928 | 0.9997 | 0.9991 | 0.9993 | 0.9997 | 1.0000 | 1.0000 | 1.0000 |
| 20 | 1.0000 | 0.9928 | 0.9997 | 0.9991 | 0.9993 | 0.9997 | 1.0000 | 1.0000 | 1.0000 |
| 21 | 0.9933 | 1.0000 | 0.9593 | 0.9573 | 0.9480 | 0.9587 | 0.9820 | 0.9283 | 0.9933 |
| 22 | 1.0000 | 1.0000 | 0.9898 | 0.9958 | 1.0000 | 1.0000 | 1.0000 | 0.9988 | 1.0000 |
| 23 | 1.0000 | 0.5594 | 0.8531 | 0.9091 | 1.0000 | 0.9881 | 0.9860 | 0.9860 | 1.0000 |
| 24 | 0.9826 | 0.1453 | 1.0000 | 0.8488 | 0.9826 | 1.0000 | 0.9709 | 1.0000 | 0.9826 |
| 25 | 0.9332 | 0.9320 | 0.9349 | 0.9235 | 0.9308 | 0.9357 | 0.9367 | 0.9377 | 0.9332 |
| 26 | 0.9934 | 0.9933 | 0.9997 | 0.9955 | 0.9944 | 0.9976 | 0.9980 | 0.9945 | 0.9934 |
| 27 | 0.9934 | 0.9919 | 0.9997 | 0.9955 | 0.9944 | 0.9976 | 0.9980 | 0.9945 | 0.9934 |
| 28 | 0.9480 | 0.9427 | 0.9472 | 0.9296 | 0.9478 | 0.9470 | 0.9447 | 0.9493 | 0.9480 |
| 29 | 0.9982 | 0.9224 | 0.9988 | 0.9815 | 0.9858 | 0.9983 | 0.9983 | 0.9981 | 0.9982 |
| 30 | 1.0000 | 0.9964 | 0.9999 | 1.0000 | 1.0000 | 1.0000 | 1.0000 | 1.0000 | 1.0000 |
| 31 | 0.9817 | 0.6191 | 0.9911 | 0.9428 | 0.9884 | 0.9879 | 0.9875 | 0.9860 | 0.9817 |
| 32 | 0.9331 | 0.6457 | 0.9473 | 0.9513 | 0.9114 | 0.9263 | 0.9432 | 0.8738 | 0.9331 |
| 33 | 0.9007 | 0.6072 | 0.8543 | 0.7766 | 0.8300 | 0.8440 | 0.8586 | 0.8216 | 0.9007 |
| 34 | 0.9383 | 0.2037 | 0.8519 | 0.8457 | 0.7284 | 0.8235 | 0.9321 | 0.9198 | 0.9383 |
| 35 | 0.9088 | 0.7756 | 0.8811 | 0.7911 | 0.8476 | 0.8273 | 0.8705 | 0.8591 | 0.9088 |
| 36 | 0.8861 | 0.5068 | 0.8362 | 0.7728 | 0.8401 | 0.8488 | 0.8478 | 0.8101 | 0.8861 |
| 37 | 0.8691 | 0.6842 | 0.8947 | 0.6932 | 0.8870 | 0.8937 | 0.8845 | 0.8221 | 0.8691 |
| 38 | 0.6581 | 0.4719 | 0.6522 | 0.6273 | 0.6461 | 0.6560 | 0.6193 | 0.6689 | 0.6581 |
| 39 | 0.6791 | 0.5478 | 0.5846 | 0.6393 | 0.6032 | 0.6151 | 0.6410 | 0.6411 | 0.6791 |
| 40 | 0.8958 | 0.6354 | 0.8958 | 0.6979 | 0.6458 | 0.7525 | 0.7292 | 0.8854 | 0.8958 |
| 41 | 0.9268 | 0.5266 | 0.8810 | 0.8151 | 0.8806 | 0.8838 | 0.8807 | 0.8254 | 0.9268 |
| 42 | 0.6586 | 0.4899 | 0.5874 | 0.5691 | 0.6007 | 0.5717 | 0.5934 | 0.6119 | 0.6586 |
| 43 | 0.6379 | 0.5617 | 0.5873 | 0.6137 | 0.5342 | 0.5801 | 0.5762 | 0.6132 | 0.6379 |
| 44 | 0.6235 | 0.5154 | 0.4979 | 0.5288 | 0.4434 | 0.5586 | 0.4733 | 0.5576 | 0.6235 |
| 45 | 1.0000 | 1.0000 | 1.0000 | 0.9997 | 1.0000 | 1.0000 | 1.0000 | 1.0000 | 1.0000 |
| 46 | 0.9914 | 0.6934 | 0.9481 | 0.9643 | 0.9531 | 0.9512 | 0.9592 | 0.9373 | 0.9914 |
| 47 | 0.7417 | 0.6849 | 0.7469 | 0.7299 | 0.6542 | 0.7436 | 0.7464 | 0.7460 | 0.7417 |
| 48 | 0.7394 | 0.6721 | 0.7420 | 0.7268 | 0.7390 | 0.7392 | 0.7415 | 0.7402 | 0.7394 |
| 49 | 0.6904 | 0.5299 | 0.6325 | 0.6442 | 0.6970 | 0.6831 | 0.5610 | 0.6490 | 0.6904 |
| 50 | 0.9934 | 0.9972 | 0.9901 | 0.9944 | 0.9977 | 0.9841 | 0.9929 | 0.9959 | 0.9934 |
| 51 | 0.5879 | 0.5592 | 0.5999 | 0.5600 | 0.5358 | 0.5067 | 0.4994 | 0.5348 | 0.5879 |
| 52 | 0.5722 | 0.5010 | 0.4719 | 0.4118 | 0.3831 | 0.5470 | 0.4023 | 0.5291 | 0.5722 |
| 53 | 0.5565 | 0.5221 | 0.4739 | 0.5545 | 0.5098 | 0.5060 | 0.5286 | 0.5431 | 0.5565 |
| 54 | 0.5120 | 0.5182 | 0.5621 | 0.5671 | 0.4543 | 0.5958 | 0.5408 | 0.5830 | 0.5120 |
| 55 | 0.8500 | 0.8108 | 0.8333 | 0.7965 | 0.8407 | 0.8386 | 0.8425 | 0.8150 | 0.8500 |
| 56 | 0.6684 | 0.5308 | 0.5094 | 0.5293 | 0.5026 | 0.5106 | 0.5141 | 0.6288 | 0.6684 |
| 57 | 0.5496 | 0.4739 | 0.4574 | 0.4565 | 0.4243 | 0.4243 | 0.4870 | 0.4652 | 0.5496 |
| 58 | 0.9663 | 0.9780 | 0.9669 | 0.9588 | 0.9737 | 0.9702 | 0.9520 | 0.9570 | 0.9663 |
| 59 | 0.5524 | 0.4111 | 0.5270 | 0.5027 | 0.4856 | 0.4799 | 0.5246 | 0.4905 | 0.5524 |
| 60 | 0.7746 | 0.5551 | 0.5963 | 0.5845 | 0.6789 | 0.7161 | 0.7872 | 0.8322 | 0.7746 |
| 61 | 0.7749 | 0.7707 | 0.7379 | 0.7933 | 0.7251 | 0.7343 | 0.7881 | 0.7502 | 0.7749 |
| 62 | 0.7398 | 0.7492 | 0.4601 | 0.6762 | 0.7111 | 0.7005 | 0.7513 | 0.7793 | 0.7398 |
| 63 | 0.6957 | 0.7381 | 0.7408 | 0.7421 | 0.7484 | 0.7567 | 0.7252 | 0.7473 | 0.6957 |
| 64 | 0.9860 | 0.9059 | 0.9849 | 0.9752 | 0.9836 | 0.9864 | 0.9823 | 0.9820 | 0.9860 |
| 65 | 0.6363 | 0.5912 | 0.5420 | 0.5906 | 0.6227 | 0.6525 | 0.6200 | 0.5833 | 0.6363 |
| 66 | 0.9993 | 0.9980 | 0.9798 | 0.9973 | 0.9808 | 0.9966 | 0.9989 | 0.9964 | 0.9993 |
| 67 | 0.9965 | 0.9992 | 0.9993 | 0.9954 | 0.9995 | 0.9985 | 0.9996 | 0.9994 | 0.9965 |
| 68 | 0.5110 | 0.4187 | 0.4610 | 0.4530 | 0.6178 | 0.4275 | 0.4241 | 0.4215 | 0.5110 |
| 69 | 0.7908 | 0.4949 | 0.5255 | 0.6276 | 0.4949 | 0.4949 | 0.5204 | 0.5408 | 0.7908 |
| 70 | 0.6168 | 0.5201 | 0.5000 | 0.5383 | 0.4992 | 0.5361 | 0.6095 | 0.6134 | 0.6168 |
| 71 | 0.6168 | 0.5201 | 0.5000 | 0.5383 | 0.4992 | 0.5161 | 0.6095 | 0.6034 | 0.6168 |
| 72 | 0.8939 | 0.6997 | 0.7154 | 0.7208 | 0.7468 | 0.6071 | 0.6814 | 0.8019 | 0.8939 |
| 73 | 0.6334 | 0.5164 | 0.5096 | 0.5081 | 0.5401 | 0.6203 | 0.5632 | 0.6067 | 0.6334 |
| 74 | 0.8813 | 0.8731 | 0.8617 | 0.8586 | 0.8834 | 0.8833 | 0.8784 | 0.8840 | 0.8813 |
| 75 | 0.9963 | 0.9957 | 0.9966 | 0.9934 | 0.9962 | 0.9958 | 0.9966 | 0.9980 | 0.9963 |
| 76 | 0.9973 | 0.9832 | 0.9914 | 0.8734 | 0.9844 | 0.9918 | 0.9917 | 0.9887 | 0.9973 |
| 77 | 0.9906 | 0.9773 | 0.9718 | 0.8812 | 0.9879 | 0.9825 | 0.9802 | 0.9871 | 0.9906 |
| 78 | 0.9906 | 0.9773 | 0.9718 | 0.8812 | 0.9879 | 0.9825 | 0.9802 | 0.9871 | 0.9906 |
| 79 | 0.8306 | 0.5500 | 0.6417 | 0.5361 | 0.7306 | 0.8083 | 0.6333 | 0.8194 | 0.8306 |

Table 12: Statistics of additional 32 downstream multi-class classification datasets.

| ID | Dataset name | # samples | # num. | # cat. |
|---|---|---|---|---|
| 0 | Iris | 150 | 4 | 0 |
| 1 | AI_index_db | 62 | 8 | 3 |
| 2 | all_data_updated | 1275 | 11 | 11 |
| 3 | milknew | 1059 | 2 | 5 |
| 4 | fitz_undersampled | 4515 | 0 | 3 |
| 5 | 0181_bridges | 105 | 2 | 9 |
| 6 | Life_expectancy | 223 | 3 | 1 |
| 7 | 0901_iris-example | 150 | 4 | 0 |
| 8 | 0238_pbcseq | 1945 | 12 | 6 |
| 9 | 1420_burst-header | 1075 | 19 | 2 |
| 10 | 0540_MyIris | 150 | 4 | 0 |
| 11 | 0659 | 151 | 3 | 2 |
| 12 | 1400_iriiiiiis | 150 | 4 | 0 |
| 13 | 0941_TEST10e627dcde | 150 | 4 | 0 |
| 14 | 0829 | 150 | 4 | 0 |
| 15 | 0968_iris | 150 | 4 | 0 |
| 16 | 1261_Heart-Disease | 303 | 5 | 5 |
| 17 | 1748_Sales_DataSet_of | 1000 | 2 | 9 |
| 18 | 2754 | 124 | 0 | 19 |
| 19 | 1310_penguins | 344 | 4 | 2 |
| 20 | 1402_iris_test | 150 | 4 | 0 |
| 21 | 1604_iris_reproduced | 150 | 4 | 0 |
| 22 | 1607_Indian-Liver | 583 | 9 | 2 |
| 23 | 1620_myiris | 150 | 3 | 0 |
| 24 | 2215_Iris | 150 | 4 | 0 |
| 25 | 0882_JuanFeldmanIris | 150 | 4 | 0 |
| 26 | 1191_Students | 480 | 4 | 12 |
| 27 | 1127_mom | 140 | 2 | 1 |
| 28 | 1394_IRIS-flower | 150 | 4 | 0 |
| 29 | 1618_aaaa | 150 | 4 | 0 |
| 30 | 1738_StocksData | 600 | 3 | 2 |
| 31 | 1811_Pokemon-with | 1017 | 9 | 3 |

Table 13: Statistics of 25 medical domain datasets. "bin@10" denotes the dataset corresponds to the one with ID 10 in the downstream binary classification collection, and "reg@1" means the dataset corresponds to the one with ID 1 in the downstream regression collection.

| ID | Dataset name | # samples | # num. | # cat. |
|---|---|---|---|---|
| bin@10 | 0445_arsenic-male | 559 | 4 | 0 |
| bin@11 | 0446_arsenic-female | 559 | 4 | 0 |
| bin@12 | 0447_arsenic-female | 559 | 4 | 0 |
| bin@17 | 1592_Diabetes-Data | 768 | 8 | 0 |
| bin@44 | 0541_plasma_retinol | 315 | 10 | 3 |
| bin@52 | b_depressed | 1429 | 12 | 9 |
| bin@55 | Breast_Cancer | 4024 | 5 | 10 |
| bin@66 | 1752_Wisconsin | 699 | 2 | 8 |
| bin@69 | 0446_newton_hema | 140 | 2 | 1 |
| bin@72 | 1011_cleve | 303 | 5 | 8 |
| bin@74 | 1564_Mammographic | 830 | 1 | 4 |
| bin@76 | diabetes_data_upload | 520 | 1 | 15 |
| bin@77 | 1451_early-stage | 520 | 1 | 15 |
| bin@78 | 1774_Early-Stage | 520 | 1 | 15 |
| bin@79 | 0408_pharynx | 195 | 2 | 8 |
| reg@1 | 0237_arsenic-female | 559 | 4 | 0 |
| reg@2 | 0251_arsenic-male | 559 | 4 | 0 |
| reg@33 | 0235_plasma_retinol | 315 | 10 | 3 |
| reg@41 | 1118_jura | 359 | 8 | 9 |
| reg@52 | 1890_ECDC-daily-data | 9370 | 3 | 6 |
| reg@53 | thyroidDF | 9172 | 7 | 23 |
| reg@55 | 2168_Intersectional | 1000 | 13 | 5 |
| reg@56 | 0130_breastTumor | 286 | 2 | 7 |
| reg@60 | 0225_veteran | 137 | 2 | 5 |
| reg@62 | 0125_pharynx | 195 | 2 | 8 |

Table 14: Results of the baselines on 25 medical domain datasets. The average rank verifies that our approach performs the best.

| ID | XGB(d) | Cat(d) | FTT(d) | Trans(d) | XGB(t) | Cat(t) | MLP(t) | Auto(t) | DCN(t) | Tab(t) | SAI(t) | FTT(t) | XTab(t) | Ours$_s$(d) |
|---|---|---|---|---|---|---|---|---|---|---|---|---|---|---|
| bin@10 | 0.8000 | 0.9009 | 0.7514 | 0.8860 | 0.8636 | 0.9009 | 0.6897 | 0.9028 | 0.6374 | 0.5495 | 0.5617 | 0.7626 | 0.8262 | 0.7607 |
| bin@11 | 0.8333 | 0.7747 | 0.7682 | 0.8216 | 0.7435 | 0.8542 | 0.7650 | 0.7878 | 0.8333 | 0.7041 | 0.7266 | 0.7643 | 0.5798 | 0.7806 |
| bin@12 | 0.8738 | 0.8912 | 0.9144 | 0.8981 | 0.9398 | 0.9062 | 0.8275 | 0.9815 | 0.8333 | 0.9537 | 0.8241 | 0.7708 | 0.8796 | 0.9248 |
| bin@17 | 0.7672 | 0.7556 | 0.8031 | 0.3363 | 0.7778 | 0.8134 | 0.7887 | 0.7830 | 0.8002 | 0.7087 | 0.7894 | 0.7957 | 0.7841 | 0.7913 |
| bin@44 | 0.4393 | 0.5278 | 0.4352 | 0.3796 | 0.4928 | 0.5149 | 0.5319 | 0.4630 | 0.5484 | 0.3858 | 0.4115 | 0.5473 | 0.3405 | 0.6235 |
| bin@52 | 0.4950 | 0.4454 | 0.4959 | 0.4417 | 0.4472 | 0.4775 | 0.4767 | 0.4699 | 0.5004 | 0.5194 | 0.5371 | 0.4529 | 0.4813 | 0.5722 |
| bin@55 | 0.8442 | 0.8492 | 0.8534 | 0.5585 | 0.8539 | 0.8559 | 0.8478 | 0.8524 | 0.8507 | 0.8341 | 0.8359 | 0.8552 | 0.8532 | 0.8500 |
| bin@66 | 0.9955 | 0.9946 | 0.9962 | 0.5170 | 0.9903 | 0.9975 | 0.9966 | 0.9952 | 0.9946 | 0.9665 | 0.9966 | 0.9982 | 0.9921 | 0.9993 |
| bin@69 | 0.6429 | 0.6480 | 0.6276 | 0.6837 | 0.6378 | 0.6990 | 0.5204 | 0.6429 | 0.6429 | 0.5561 | 0.5204 | 0.5714 | 0.4847 | 0.7908 |
| bin@72 | 0.8290 | 0.8474 | 0.8160 | 0.8398 | 0.7668 | 0.8139 | 0.8409 | 0.8561 | 0.8398 | 0.6742 | 0.8539 | 0.8593 | 0.8182 | 0.8939 |
| bin@74 | 0.8780 | 0.8833 | 0.8887 | 0.8718 | 0.8792 | 0.8930 | 0.7521 | 0.8833 | 0.8776 | 0.8417 | 0.8784 | 0.8791 | 0.8630 | 0.8813 |
| bin@76 | 0.9973 | 0.9984 | 0.9984 | 0.2641 | 0.9902 | 0.9941 | 0.9891 | 0.9973 | 0.9785 | 0.9031 | 0.9988 | 0.9988 | 0.9742 | 0.9973 |
| bin@77 | 0.9842 | 0.9924 | 0.9984 | 0.7633 | 0.9906 | 0.9912 | 0.9723 | 0.9668 | 0.9621 | 0.9676 | 0.9902 | 0.9840 | 0.9828 | 0.9906 |
| bin@78 | 0.9842 | 0.9924 | 0.9984 | 0.9859 | 0.9891 | 0.9988 | 0.9777 | 0.9902 | 0.9773 | 0.8660 | 0.9965 | 0.9863 | 0.9828 | 0.9906 |
| bin@79 | 0.7667 | 0.7306 | 0.7778 | 0.7639 | 0.7819 | 0.7639 | 0.7778 | 0.6972 | 0.8028 | 0.5306 | 0.6972 | 0.7556 | 0.5194 | 0.8306 |
| reg@1 | 0.0437 | 0.0210 | 0.0326 | 0.0638 | 0.0469 | 0.0178 | 0.0159 | 0.0121 | 0.0438 | 0.0600 | 0.0402 | 0.0356 | 0.0551 | 0.0087 |
| reg@2 | 0.0485 | 0.0230 | 0.0402 | 0.0640 | 0.0293 | 0.0262 | 0.0732 | 0.0292 | 0.0420 | 0.0508 | 0.0540 | 0.0418 | 0.0580 | 0.0124 |
| reg@33 | 0.1661 | 0.1376 | 0.1341 | 0.1322 | 0.1374 | 0.1415 | 0.1214 | 0.1428 | 0.1264 | 0.5017 | 0.1280 | 0.1279 | 0.1277 | 0.1219 |
| reg@41 | 0.0820 | 0.0860 | 0.1074 | 0.1286 | 0.0779 | 0.0759 | 0.0970 | 0.0872 | 0.1175 | 0.3678 | 0.0901 | 0.0958 | 0.1544 | 0.0839 |
| reg@52 | 0.0226 | 0.0203 | 0.1333 | 0.1496 | 0.0006 | 0.0137 | 0.1360 | 0.1362 | 0.1367 | 0.1387 | 0.1349 | 0.1350 | 0.1413 | 0.0033 |
| reg@53 | 0.2271 | 0.2252 | 0.2396 | 0.2509 | 0.2226 | 0.2255 | 0.2454 | 0.2406 | 0.2452 | 0.2449 | 0.2399 | 0.2399 | 0.2463 | 0.2310 |
| reg@55 | 0.2009 | 0.1874 | 0.1881 | 0.2098 | 0.1863 | 0.1887 | 0.1883 | 0.1917 | 0.1852 | 0.2159 | 0.1894 | 0.1858 | 0.1884 | 0.1825 |
| reg@56 | 0.2165 | 0.1997 | 0.1848 | 0.2058 | 0.1931 | 0.2070 | 0.1770 | 0.1876 | 0.1825 | 0.8456 | 0.1792 | 0.1780 | 0.1961 | 0.1722 |
| reg@60 | 0.1412 | 0.1102 | 0.1115 | 0.1119 | 0.1108 | 0.1083 | 0.1109 | 0.1073 | 0.1110 | 0.2060 | 0.1116 | 0.1106 | 0.1909 | 0.1117 |
| reg@62 | 0.2379 | 0.2117 | 0.2059 | 0.2656 | 0.1957 | 0.2078 | 0.1921 | 0.2054 | 0.2015 | 0.3927 | 0.1880 | 0.1976 | 0.2418 | 0.1815 |
| rank | 8.3(3.0) | 6.2(3.4) | 6.1(2.8) | 10.9(3.7) | 6.5(3.6) | 4.8(3.7) | 8.1(3.8) | 6.6(3.5) | 7.5(3.8) | 12.0(3.1) | 7.7(3.7) | 6.3(3.3) | 10.6(3.0) | **3.5(2.8)** |

Table 15: The average values (standard deviations) of all method ranks under several data scenarios. We use the TP-BERTa version pre-trained on 101 binary classification datasets (i.e., "TP-BERTa(single, binclass)") under imbalanced binary classification and multi-class classification settings.

| Models: | XGB(d) | Cat(d) | FTT(d) | Trans(d) | XGB(t) | Cat(t) | MLP(t) | Auto(t) | DCN(t) | Tab(t) | SAI(t) | FTT(t) | Xtab(t) | Ours(d) |
|---|---|---|---|---|---|---|---|---|---|---|---|---|---|---|
| Imbalanced binary classification | | | | | | | | | | | | | | |
| $p < 1/3$ | 7.9(4.0) | 8.0(4.3) | 6.6(3.6) | 10.6(4.7) | 6.9(4.6) | 6.4(4.2) | 8.5(4.3) | 7.4(3.5) | 6.6(4.0) | 11.2(4.2) | 8.6(3.5) | 6.6(3.7) | 9.0(4.0) | **6.2(4.0)** |
| $p < 1/5$ | 8.1(4.2) | 8.1(4.4) | 6.8(2.9) | 10.3(4.4) | 7.0(5.1) | 6.4(4.8) | 10.1(4.0) | 6.0(3.3) | 7.2(4.3) | 10.9(4.2) | 10.1(3.6) | 7.1(3.6) | 7.8(4.2) | **5.2(3.2)** |
| $p < 1/8$ | 8.0(4.1) | 7.3(4.6) | 6.9(3.2) | 10.2(4.2) | 6.0(4.9) | 6.7(4.7) | 10.1(4.6) | **5.5(3.5)** | 8.3(4.4) | 10.8(3.9) | 11.0(2.3) | 6.8(3.5) | 7.7(4.2) | **5.5(3.1)** |
| $p < 1/20$ | 10.0(2.5) | 5.8(4.5) | 8.3(4.0) | 8.0(4.8) | **3.1(1.4)** | 7.6(5.3) | 9.0(5.4) | 4.8(4.8) | 8.6(4.5) | 11.0(6.0) | 11.9(1.7) | 8.5(4.1) | 6.8(1.7) | 6.3(3.2) |
| 32 additional multi-class classficaition tasks | | | | | | | | | | | | | | |
| rank | 5.8(2.3) | 7.0(3.1) | 6.4(2.9) | 6.7(3.7) | **4.9(3.6)** | 7.1(3.5) | 7.3(4.2) | 8.7(3.2) | 7.1(3.3) | 13.4(1.6) | 7.8(3.3) | 5.9(3.4) | 12.7(2.4) | 5.1(2.2) |