# OpenReview forum: "Making Pre-trained Language Models Great on Tabular Prediction"
_ICLR.cc/2024/Conference — ICLR 2024 spotlight_

### Official Review · Reviewer_e2F6 · 2023-10-25

**Soundness:** 3 good
**Presentation:** 3 good
**Contribution:** 3 good
**Rating:** 8
**Confidence:** 4

**Summary:**

This work introduces TP-BERTa, an LLM for tabular data prediction. TP-BERTa achives strong performance on many regression tasks and binary classification tasks. TP-BERTa incorporates 2 novel changes: (1) relative magnitude tokenization and (2) intra-feature attention.

**Strengths:**

- The model performance is very strong, even outperforming tuned XGBoost and other tabular methods. The paper does a good job comparing against many relevant baselines
- The introduced method architecture changes are both clear and intuitive
- Ablations for these changes and the experiments suggest that they both contribute to TP-BERTa's improved performance.

**Weaknesses:**

- The experiments are restricted to binary classification, would be nice to see experiments for multi-class classification as well
- The method introduces a hyperpameter lambda which is simply fixed to 0.1. It would be nice to see more discussion / ablations of this parameter.
- Authors claim the model will be made available but at this time have not shared any code

**Questions:**

- Is it possible to perform the magitude tokenization without an added loss term and hyperparameter (e.g. by carefull designing the embeddings)?
- Is there a reason to prefer C4.5 discretization over CART?
- Will the authors release all code or only the trained model?

---

> ### Author Response · Authors · 2023-11-20
> **To e2F6 (1/2, Q1 & Q2)**
>
> We profoundly thank you for the positive evaluations on the quality and significance of our paper, and propose constructive questions.
>
> Q1: Can we add experiments on multi-class datasets?
>
> A1: Thank you for your suggestion. We did not include the multi-class dataset for several reasons mentioned in the end of Sec. 2.3 (lines before Sec. 3). We would like to add some experiments on downstream multi-class datasets in the final version. Overall, we find our TP-BERTa outperforms deep model baselines and is comparable with GBDTs on multi-class datasets as well.
> Here are rank results (average values and standard deviations) on 32 extra multi-class datasets from the database, the TP-BERTa here is the version pre-trained on 101 binary classification datasets. Detailed results will be sorted into our final version.
>
> |             | 32 downstream multi-classification tasks |
> |-------------|:----------------------------------------:|
> | XGBoost(d)  | 5.8(2.3)                                 |
> | CatBoost(d) | 7.0(3.1)                                 |
> | FTT(d)      | 6.4(2.9)                                 |
> | TransTab(d) | 6.7(3.7)                                 |
> | XGBoost(t)  | **4.9(3.6)**                             |
> | CatBoost(t) | 7.1(3.5)                                 |
> | MLP(t)      | 7.3(4.2)                                 |
> | AutoInt(t)  | 8.7(3.2)                                 |
> | DCNv2(t)    | 7.1(3.3)                                 |
> | TabNet(t)   | 13.4(1.6)                                |
> | SAINT(t)    | 7.8(3.3)                                 |
> | FTT(t)      | 5.9(3.4)                                 |
> | XTab(t)     | 12.7(2.4)                                |
> | TP-BERTa(d) | $\underline{5.1(2.2)}$      |
>
> ---
> Q2: Is it possible to perform RMT without an added loss (regularization loss) term and hyperparameter ($\lambda$) (e.g., by carefully designing the embeddings)?
>
> A2: In our experiment, the hyperparameter $\lambda$ is fixed at 0.1 during pre-training, in fine-tune stage we did not use the regularization (i.e., $\lambda = 0$ in fine-tune stage). Therefore, strictly conducting ablations on $\lambda$ needs several pre-training groups, which is out of rebuttal time limit and our cost limit, because we pre-trained TP-BERTa on the rented commercial computing server, with each pre-training round taking 72～143 hours and hundreds of dollars (detailed pre-training budget is in Appendix D).
>
> In order to explore the impact of the added loss, we would like to conduct the ablation by fine-tuning a non-pre-trained TP-BERTa with different $\lambda$s in substitution, where all magnitude tokens are randomly initialized at the beginning. Here are the AUC performance changes on 80 binary classification datasets by comparing the group to the one with no added loss (i.e., difference between a group and the group “$\lambda = 0$"), the value in column “$|\Delta| \le 3\\%$" denotes the number of datasets with AUC variation less than 3%.
>
> | $\lambda$ | $ \vert \Delta \vert \le 3\\%$  | $\Delta < -3\\%$  | $\Delta > 3\\%$ |
> |-----------|:----:|:----:|:----:|
> | 0.1       | 54 | 10 | 16 |
> | 0.3       | 50 | 13 | 17 |
> | 0.5       | 51 | 13 | 16 |
>
> It can be seen the most datasets are not significantly impacted by the added loss or the specific value of $\lambda$, but for several datasets (the datasets in the "$\Delta > 3\\%$"  case) a regularization in pre-training is helpful. We further inspect these datasets and find most of them are very small (less than 1000 samples), which may be not sufficient to learn the reasonable magnitude token embeddings from scratch. Therefore, our designed regularization on RMT is to help fast convergence for the magnitude token learning, especially helpful for downstream small-scale datasets which have no sufficient data diversity. For normal-size datasets (in data volume and feature quantity), removing the added loss does not affect the performance.
>
> **Is it possible to drop the regularization (added loss)?** We think it is possible to remove the $\lambda$ by careful initialization for magnitude token embeddings (like the initialization of LM position embedding`[1]`). Besides, we have supplied the t-SNE visualization (in current PDF revision, Appendix F, page 14, Fig. 4) of the 256 magnitude token embeddings, which are placed on a highly regular manifold (refer to Q5/A5 of the first reviewer(AREw, [link]( https://openreview.net/forum?id=anzIzGZuLi&noteId=HtWpGkMMs0)) for detailed analysis). Based on the observed regular pattern of the magnitude token distribution, it is possible to design suitable initialization for these tokens with regular math functions, by adhering to the principle of “the closer two numerical values are, the more similar they should be” rather than using the added loss.

---

> > ### Author Response · Authors · 2023-11-20
> > **To e2F6 (2/2, Q3 & Q4)**
> >
> > Q3: Is there a reason to prefer C4.5 discretization over CART?
> >
> > A3: Considering the cost of pre-training, we made a conservative choice on C4.5 discretization method by following the recent work on tabular numerical embedding`[2]`, and C4.5 is widely regarded as a stable and reliable scheme in classical works of numerical discretization`[3]`. Our primary focus is to convey the numerical processing philosophy of RMT in the LM adaption to tabular data, especially for the high-precision tabular prediction task, which is the first attempt. Using different discretization methods (e.g., CART or other classical binning algorithms) may lead to slight performance changes, while the significant performance boost can be acquired by replacing the previous numerical encoding strategy (e.g., Value2Str in existing tabular LM works`[4, 5, 6]`) with the RMT-like strategy, facilitating the LMs' awareness of numerical values. We hope our design spirit can stimulate broad reflection in this edge-cutting field.
> >
> > ---
> > Q4: Will all codes or only the trained model be released?
> >
> > Certainly, we would like to make all codes (including model adaption, pre-training, fine-tuning and experiment examples) and trained TP-BERTa weights publicly available. Since the amount of implementation and experiment codes are large, we are refactoring the codes into a unified API for convenient model invocation and experiment table reproduction. Therefore, we have not shared the public code link, but private repository has been established, it will be available once the anonymous period ends.
> >
> > ---
> > Reference
> >
> > `[1]` Wang, Benyou, et al. "On position embeddings in bert." ICLR. 2020.
> >
> > `[2]` Gorishniy, Yury, Ivan Rubachev, and Artem Babenko. "On embeddings for numerical features in tabular deep learning." NeurIPS. 2022.
> >
> > `[3]` Dougherty, James, Ron Kohavi, and Mehran Sahami. "Supervised and unsupervised discretization of continuous features." ICML. 1995.
> >
> > `[4]` Borisov, Vadim, et al. "Language Models are Realistic Tabular Data Generators." ICLR. 2022.
> >
> > `[5]` Zhang, Tianping, et al. "Generative Table Pre-training Empowers Models for Tabular Prediction." arXiv. 2023.
> >
> > `[6]` Han, Hongwei, et al. "LUNA: Language Understanding with Number Augmentations on Transformers via Number Plugins and Pre-training." arXiv. 2022.

---

> > > ### Comment · Reviewer_e2F6 · 2023-11-21
> > > **Updated my score to 8**
> > >
> > > The reviewers have sufficiently answered my concerns and I am updating my score to 8.

---

### Official Review · Reviewer_cw3Y · 2023-11-01

**Soundness:** 3 good
**Presentation:** 3 good
**Contribution:** 2 fair
**Rating:** 6
**Confidence:** 4

**Summary:**

The paper proposed a new pretrained model for table prediction called TP-BERTa. TP-BERTa is based on the RoBERTa architecture with two modifications: 1) it discretizes numerical feature values as relative magnitude tokens (RMT) so that the tokens can be treated as meaningful words in the LM’s vocabulary, 2) it adopts the intra-feature attention (IFA) method to attentively fuse the embeddings of feature name and values to a single embedding. TP-BERTa has been trained on a combination of 101 classification datasets and 101 regression datasets. Results show that TP-BERTa outperforms other tabular deep learning models and is comparable to GBDT. Ablation studies show that RMT and IFA boost the performance.

**Strengths:**

1. According to Table 1, the performance of TP-BERTa is strong. It consistently outperforms other tabular DL models and is comparable to XGBoost and CatBoost.
2. The idea of applying relative magnitude tokens (RMT) in tabular DL models is novel according to my knowledge. As pointed out by the "On Embeddings for Numerical Features in Tabular Deep Learning" paper, appropriately embed numerical features is important for tabular DL models. RMT can enhance language models in handling the numerical values that appear often in tabular datasets. As shown in the upper-half of Table-2, RMT significantly outperforms the value2str strategy.

**Weaknesses:**

1. Compared with RMT, the intra-feature attention method is marginally novel and is not showing significant performance boost.
2. The author has not studied the impact of pretrain data diversity in the performance of TP-BERTa. For example, how good will TP-BERTa be if it is only pretrained on 10 classification datasets and 10 regression datasets?

**Questions:**

What's the impact of pretrain data diversity in the performance of TP-BERTa? (See weakness)

---

> ### Author Response · Authors · 2023-11-20
> **To cw3Y (1/1, Q1 & Q2)**
>
> We really thank you for finding our work novel and distinguishable from the traditional tabular deep models, and propose insightful questions!
>
> Q1: The impact of pre-training dataset diversity?
>
> A1: Empirically, when using less diverse pre-training datasets, the model performance is likely to asymptotically degraded to the performance of using RoBERTa weights in Table 3. We additionally conduct TP-BERTa (single task) pre-training with 3, 10, 30 datasets (randomly selected from 101 binary classification datasets) respectively. In 10-dataset pre-training experiment, we repeated 3 groups (group $a$, $b$, $c$) with disjoint 10 datasets in each group. The results are reported as follows, the value in “$|\Delta| \le 0.5\\%$" group denotes the number of datasets with AUC variation less than 0.5%, “Avg. diff.” means the average performance difference on significantly changed datasets, “TP-BERTa($n$)” means the TP-BERTa pre-trained on $n$ datasets.
>
> | Initialization | $ \vert \Delta \vert \le 0.5\\%$  | $\Delta < -0.5\\%$ | $\Delta > 0.5\\%$  | $\Delta < -3\\%$  | $\Delta > 3\\%$ | Avg. diff |
> |----------------|----|----|----|----|---|-----------|
> | Random         | 29 | 41 | 10 | 26 | 5 | -3.16%    |
> | RoBERTa        | 26 | 35 | 19 | 21 | 6 | -2.79%    |
> | TP-BERTa(3)    | 26 | 35 | 19 | 21 | 6 | -2.86%    |
> | TP-BERTa(10)-$a$ | 31 | 32 | 17 | 18 | 5 | -2.34%    |
> | TP-BERTa(10)-$b$ | 37 | 28 | 15 | 15 | 5 | -2.18%    |
> | TP-BERTa(10)-$c$ | 33 | 30 | 17 | 18 | 5 | -2.41%    |
> | TP-BERTa(30)   | 48 | 21 | 11 | 12 | 3 | -1.67%    |
>
> When pre-training dataset diversity is insufficient (3-dataset group), the TP-BERTa performance is almost close to the non-pre-trained one (RoBERTa initialization). Even all use 10 datasets, the 3 10-dataset groups exhibit different downstream performances since their different seen features in pre-training, some of which hold more benefit in downstream datasets. Besides, randomly sampled pre-training datasets varies in data volumes, i.e., different 10-dataset groups are still different in data diversity. Therefore, our primary contribution focuses on “making” LMs great on the tabular prediction (in the title), i.e., RMT and IFA adaption techniques, in which RMT elevates LM-based tabular models to an entirely new level (compared to the numerical processing strategy of existing LM-based tabular models`[1, 2, 3]`), especially in high-precision tabular prediction tasks. We hope the design of TP-BERTa can stimulate broad reflection in this edge-cutting field and tabular foundation models (e.g., tabular LLM design, LLM enhancement on table processing).
>
> ---
> Q2: Compared to RMT, the IFA method seems marginally novel and not showing significant performance boost?
>
> A2: Thank you for finding our IFA novel! The function of IFA is to reduce theory computational complexity introduced by the large language model and provide potential advantage of fast convergence (for detailed complexity and advantage analysis, please refer to the Q2/A2 of the reviewer AREw, which is also important under the conditions of limited computation resource or time, making it reproduction-friendly and easy to follow. It takes 32% more average training time after removing IFA (lower-half of the Table 2).
>
> ---
> Reference
>
> `[1]` Borisov, Vadim, et al. "Language Models are Realistic Tabular Data Generators." ICLR. 2022.
>
> `[2]` Zhang, Tianping, et al. "Generative Table Pre-training Empowers Models for Tabular Prediction." arXiv. 2023.
>
> `[3]` Han, Hongwei, et al. "LUNA: Language Understanding with Number Augmentations on Transformers via Number Plugins and Pre-training." arXiv. 2022.

---

### Official Review · Reviewer_v1fL · 2023-11-01

**Soundness:** 2 fair
**Presentation:** 2 fair
**Contribution:** 1 poor
**Rating:** 6
**Confidence:** 2

**Summary:**

The paper presents pre-training methods for the tabular data. Unlike the traditional text-based pre-training such as masked LM or language modeling, tabular data does not have common token distributions and is not suitable for the traditional pretraining mechanism. The paper introduces a way to discretize the continuous values to discrete tokens based on their relative magnitude. Doing so allows posing a pretraining objective over heterogeneous features.

---

The score is updated post-rebuttal. See the other comment for the details.

**Strengths:**

* Paper is mostly clearly written.

**Weaknesses:**

* It is very unclear if some distributional patterns from common tabular data can be generalized to other data with completely different distributions. In text, pretraining can learn generic linguistic features such as meaning of the English words or grammars. In tabular data, the core assumption does not hold because it differs drastically between data sources.
* The empirical comparison does not essentially provide evidence that pre-training is the main factor to improve the performance of the downstream tasks.

**Questions:**

* As discussed in the weaknesses section, pretrained tokens from the quantized tabular data do not have a good interpretation of what they are really capturing from. I hope authors can provide discussions and empirical analysis on the tokens that they are capturing.

---

> ### Author Response · Authors · 2023-11-20
> **To v1fL (1/2, Q1)**
>
> Q1: It is unclear if some tabular data distribution patterns can be captured with the LMs, like word meaning or grammars captured in text pre-training paradigm (e.g., auto-encoder LMs with the masked language modeling, auto-regressive LMs with the autoregressive one)?
>
> A1: Thank you for your concern. We will clarify the question from 3 aspects: previous literatures, main contributions, and the potential values of our work.
>
> **Previous literatures**: some previous works also use LM adaption for tabular data related tasks (e.g., GReaT`[1]`, TapTap`[2]` and LUNA`[3]`), and they have proved the capability and feasibility of LMs to model the tabular patterns. Notably, none of these LM-based tabular models is tailored for high-precision tabular prediction tasks, they all directly treat numerical tables as template texts and are designed for generation tasks, such as synthetic table generation, the LM generated tabular data shows great distribution similarity with the original data, implying LM's potential in modeling tabular data patterns`[1]`.
>
> **Main contributions**: our primary endeavor is (1) “making” pre-trained LMs (e.g., RoBERTa) great on tabular prediction tasks; (2) making cross-table pre-training on heterogeneous tabular datasets, such pre-training feasibility is based on the LM ability of associating heterogeneous features in the language space, i.e., when we transform features into texts and maintain LMs' sensitivity to the numerical values, they can transfer tabular patterns across tables. We will introduce our efforts in tailoring LMs to tabular data patterns, and demonstrating their superiority on tabular prediction tasks.
>
> **(1) Technical contributions**
>
> - To make LMs sensitive to numerical values, we propose RMT adaption technique. It is the first attempt to represent relative magnitudes of numerical values in the language space (and has interpretability with clear visualized distribution, please refer to Q2/A2 for details), conveying a numerical processing philosophy in LM adaption to tabular data. RMT significantly outperforms the numerical encoding strategy of existing LM-based tabular models (12.45% average AUC promotion compared to directly treating values as raw strings, in the upper half of Table 2), it enhances the LM sensitivity to the numerical value, which is a common and critical in high-precision tabular prediction tasks (e.g., regression).
>
> - To assist LMs in learning tabular feature organization, we propose the IFA mechanism to explicitly match tabular feature name-value pair for a better contextual understanding. IFA can also significantly save training time (for the analysis of IFA theory computational complexity, please refer to the Q2/A2 of the reviewer AREw, which makes the TP-BERTa reproduction-friendly and easy to follow.
>
> **(2) Empirical contributions**: we conduct extensive evaluations to demonstrate the traditional LMs, with suitable adaptions, has the potential to be comparable with or even exceed the strong baselines on tabular prediction tasks (in Table 1), which indicates LMs are great on tabular prediction and shows a promising new choice for transferring knowledge across tables (it was challenging for the heterogenous nature of tabular features, traditional tabular transfer models `[4, 5]` rely on overlapped features between pre-training datasets and downstream ones, while LMs can inherently associate features in the language space). Besides, we explore the feature appetite of the LMs in handling tabular data, which can help architecture selection based on feature type characteristics in future tabular prediction studies.
>
> **Potential values**: the TP-BERTa is not a “naive SOTA work”, our primary focus lies in two facets: (1) “making”, i.e., conveying the design philosophy of the TP-BERTa (RMT and IFA) to make LMs aware of numerical values and tabular feature organization. The proposed methodology helps LMs to learn tabular patterns and improve efficiency (performance and training speed) on tabular prediction tasks, we hope TP-BERTa can stimulate broad reflection in the edge-cutting fields like LM adaption to tabular data, tabular foundation models and LLM enhancements. (2) the TP-BERTa sends a significant signal that pure NLP LMs can also outperform in high-precision tabular tasks, which provides a completely brand perspective in the tabular transfer learning field.
>
> Overall, we make novel technical endeavor in adapting LMs to tabular patterns and extensively demonstrate their promising tabular prediction performances.

---

> > ### Author Response · Authors · 2023-11-20
> > **To v1fL (2/2, Q2)**
> >
> > Q2: What the pre-trained magnitude tokens really captured? (How to interpret the magnitude tokens)
> >
> > A2: We have added the t-SNE visualization of the TP-BERTa‘s 256 magnitude token embeddings in Appendix F “Interpretability of RMT” (page 14, Fig. 4) of the current PDF revision (using `scikit-learn` package with default function parameters). After pre-training, we interestingly find all 256 tokens are located on a highly regular manifold, even they are randomly initialized in the language space at the beginning. It can be empirically seen the token distribution maintains an intuitive assumption that the embedding of a numerical value should be close to the embedding of a nearby one, i.e., an overall trend that a magnitude bin (token) is close to a nearby one in the language space. This visualization also demonstrates the TP-BERTa is sensitive to the numerical value magnitude and benefits from the captured regular relationship among the relative magnitudes, which interprets its significant progress on the existing LM-based tabular models (e.g., directly treating values as raw strings).
> >
> > ---
> > Reference
> >
> > `[1]` Borisov, Vadim, et al. "Language Models are Realistic Tabular Data Generators." ICLR. 2022.
> >
> > `[2]` Zhang, Tianping, et al. "Generative Table Pre-training Empowers Models for Tabular Prediction." arXiv. 2023.
> >
> > `[3]` Han, Hongwei, et al. "LUNA: Language Understanding with Number Augmentations on Transformers via Number Plugins and Pre-training." arXiv. 2022.
> >
> > `[4]` Wang, Zifeng, and Jimeng Sun. "Transtab: Learning transferable tabular transformers across tables." NeurIPS. 2022.
> >
> > `[5]` Zhu, Bingzhao, et al. "XTab: Cross-table Pretraining for Tabular Transformers." ICML. 2023.

---

### Official Review · Reviewer_AREw · 2023-11-01

**Soundness:** 3 good
**Presentation:** 3 good
**Contribution:** 3 good
**Rating:** 8
**Confidence:** 3

**Summary:**

The paper introduces a novel approach to adapt pre-trained language models for tabular data prediction, presenting the TP-BERTa model based on RoBERTa's architecture. This model employs a unique method termed "relative magnitude tokenization" to transform scalar numerical feature values into a more discrete, high-dimensional token format. This tokenization process enables the language model to comprehend relative value magnitudes within the language representation space. Additionally, the paper showcases the intra-feature attention (IFA) mechanism that fuses feature names and their corresponding values attentively. Comparative evaluations on 145 datasets reveal that the TP-BERTa outstrips conventional tabular deep neural networks (DNNs) and is on par with Gradient Boosted Decision Tree models in typical tabular data settings.

**Strengths:**

1. **Originality:**
    - The approach of "relative magnitude tokenization" (RMT) is an inventive technique for adapting pre-trained language models to tabular data. This method of converting scalar values to a tokenized format to be perceived as meaningful words within the language model's vocabulary stands out as a significant contribution.
    - The intra-feature attention (IFA) module to fuse feature name and value embeddings is another commendable addition to the field. This ensures a more contextual understanding of the feature within the model.

2. **Quality:**
    - The proposed model has been extensively tested against 145 datasets, which is a comprehensive evaluation to validate its efficacy.
    - Superior performance against established models like common tabular DNNs and close competition with GBDTs further establish the quality and utility of the proposed approach.

3. **Clarity:**
    - The paper delineates the methodology with sufficient detail, ensuring understanding and reproducibility.

**Weaknesses:**

1. It would have been beneficial if the paper delved deeper into the limitations and potential pitfalls of the relative magnitude tokenization technique. Understanding how the granularity of this tokenization might impact model performance, especially in cases with intricate numerical nuances, is crucial.
2. Comparisons with Gradient Boosted Decision Trees are noted, but an in-depth discussion regarding scenarios where GBDTs might outshine or underperform against the proposed TP-BERTa would provide readers with a clearer perspective.

**Questions:**

1. How does the TP-BERTa model perform when handling missing, extreme or highly imbalanced values?
2. Could the authors provide insights into the computational complexity introduced by the relative magnitude tokenization and intra-feature attention mechanisms, especially when scaling to larger datasets?
3. Were there specific domains or types of datasets where the TP-BERTa particularly excelled or faced challenges?
4. Can the relative magnitude tokenization be fine-tuned or adaptively adjusted (e.g., bin size) based on the domain or the nature of the data to potentially yield better results?
5. Has there been any visualization or analysis conducted to assess how effectively the proposed model captures the representations of numerical values?

---

> ### Author Response · Authors · 2023-11-20
> **To AREw (1/3, Q1)**
>
> We sincerely thank you for your insightful questions and constructive suggestions to make our work more thorough and comprehensive.
>
> Q1: How does TP-BERTa perform when handling imbalanced datasets or missing values?
>
> A1: In summary, without hyperparameter tuning, TP-BERTa can still hold its superiority in moderate class-imbalance or feature absence conditions. The detail experiment results and analysis are as follows.
>
> **Imbalanced-class scenario**: to inspect the performances on imbalanced datasets, we create pivot tables by filtering datasets whose minor-class proportions, i.e., $p = \frac{\text{min}(\\# \text{positive}, \\# \text{negative})}{\\# \text{samples}}$, are less than 1/3 (32 datasets), 1/5 (18 datasets), 1/8 (12 datasets), 1/20 (4 datasets) from 80 binary classification datasets. We report the average ranks (standard deviations); “(d)” and “(t)” denote default and tuned hyper-parameters.
>
> |             | $p <$ 1/3   | $p <$ 1/5   | $p <$ 1/8   | $p <$ 1/20  |
> |-------------|-----------|-----------|-----------|-----------|
> | XGBoost(d)  | 7.9(4.0)  | 8.1(4.2)  | 8.0(4.1)  | 10.0(2.5) |
> | CatBoost(d) | 8.0(4.3)  | 8.1(4.4)  | 7.3(4.6)  | 5.8(4.5)  |
> | FTT(d)      | 6.6(3.6)  | 6.8(2.9)  | 6.9(3.2)  | 8.3(4.0)  |
> | TransTab(d) | 10.6(4.7) | 10.3(4.4) | 10.2(4.2) | 8.0(4.8)  |
> | XGBoost(t)  | 6.9(4.6)  | 7.0(5.1)  | $\underline{6.0(4.9)}$  | **3.1(1.4)**  |
> | CatBoost(t) | $\underline{6.4(4.2)}$  | 6.4(4.8)  | 6.7(4.7)  | 7.6(5.3)  |
> | MLP(t)      | 8.5(4.3)  | 10.1(4.0) | 10.1(4.6) | 9.0(5.4)  |
> | AutoInt(t)  | 7.4(3.5)  | $\underline{6.0(3.3)}$  | **5.5(3.5)**  | $\underline{4.8(4.8)}$  |
> | DCNv2(t)    | 6.6(4.0)  | 7.2(4.3)  | 8.3(4.4)  | 8.6(4.5)  |
> | TabNet(t)   | 11.2(4.2) | 10.9(4.2) | 10.8(3.9) | 11.0(6.0) |
> | SAINT(t)    | 8.6(3.5)  | 10.1(3.6) | 11.0(2.3) | 11.9(1.7) |
> | FTT(t)      | 6.6(3.7)  | 7.1(3.6)  | 6.8(3.5)  | 8.5(4.1)  |
> | XTab(t)     | 9.0(4.0)  | 7.8(4.2)  | 7.7(4.2)  | 6.8(1.7)  |
> | TP-BERTa(d) | **6.2(4.0)**  | **5.2(3.2)**  | **5.5(3.1)**  | 6.3(3.2)  |
>
> It is obvious that TP-BERTa outperforms baselines in moderate class-imbalance situations. In the extremely imbalanced situations (4 datasets, with $p <$ 0.05), GBDTs showcase dominating performances, but TP-BERTa still outperforms most DNN approaches. Perhaps this is attributed to TP-BERTa leveraging the transferability and semantic understanding capabilities of the language model, thus resulting in consistent performance across various levels of data imbalance. Notably, AutoInt`[1]` exhibits promising performances in the imbalance-class scenario, which is originally designed for click-through rate prediction (a classical binary classification task). We think this may due to its specific combinatorial feature design that helps to capture patterns for minority class by manually combining original features to generate new features. We will sort the pivot tables and analysis on imbalanced datasets into our final version.
>
> **Missing-value scenario**: on the 32 imbalanced binary datasets ($p <$ 1/3), we further conduct missing-value experiments by randomly dropping a proportion of features for each sample (including training, validation and test sets), i.e., `Xs[np.random(Xs.shape) < missing_rate] = np.nan`. We conduct experiment on datasets with missing rates of 0.1 (10 % features dropped) and 0.3 (30 % features dropped) and compare with several representative baselines (default and fine-tuned).  For missing rates exceeding 0.3 (e.g., 0.5 or higher), we observed that the performances of most baselines on various datasets closely resemble random guessing. Thus, we opted to exclude such extreme missing rates from our analysis. We employ the mean value for imputing missing numerical features and utilize the most frequent value for filling in missing categorical features. The average performance rank (standard deviations) among models are reported as follows, “(d)” and “(t)” denote that the default and tuned hyperparameters are used.
>
> |                  | missing rate = 0.1 | missing rate = 0.3 |
> |-------------|--------------------|--------------------|
> | XGBoost(d)  | 4.3(2.1)           | 4.5(2.1)           |
> | CatBoost(d) | 3.9(2.4)           | **3.2(2.0)**       |
> | FTT(d)          | 4.0(1.7)           | 4.2(2.1)           |
> | XGBoost(t)  | 5.0(2.2)           | 4.0(1.7)           |
> | CatBoost(t) | 4.1(1.8)           | 4.3(2.2)           |
> | FTT(t)      | $\underline{3.7(1.8)}$           | $\underline{3.5(1.9)}$           |
> | TP-BERTa(d) | **3.0(1.6)**       | 4.2(1.9)           |
>
> In summary, TP-BERTa inherently has a strong ability to handle imbalanced data and missing values, possibly stemming from the language model's robust generalization. However, it cannot surpass algorithms specifically optimized for such extreme scenarios (e.g.Catboost). We intend to delve deeper into this aspect in our future research. Thank you!

---

> ### Author Response · Authors · 2023-11-20
> **To AREw (2/3, Q1, Q2 & Q3)**
>
> Q2: The insights into the computational complexity introduced by RMT and IFA, especially on the larger datasets?
>
> A2: Assume $F$ is the feature number, $N$ is the data scale, $L$ is the layer number of the LM. In summary, RMT introduces extra CPU computational complexity of $\text{O}(FN\text{log}N)$, **which is equivalent to training an $F$-feature C4.5 decision tree (which is known to be very efficient)**. Adding IFA gives a total LM computational complexity of $\text{O}(NF) + \text{O}(NLF^2)$, while without IFA, the complexity is at least $\text{O}(4NLF^2)$. **Therefore, employing IFA is beneficial for optimizing computational resources.**
>
> **In RMT process**, for a dataset with $F$ numerical features and $N$ samples, we will fit $F$ single-feature C4.5 decision trees (DTs) to the labels $y$ for each numerical feature, i.e., fitting C4.5$^{(i)}(x_i)$ to $y$. Since learning C4.5 DTs is CPU operation, and RMT takes place in the tokenization process, in our implementation it was incorporated with the `Tokenizer` class in the `transformers` package as a preprocessing step. The extra CPU computational complexity is from training $F$ single-feature C4.5 DTs, i.e., $\text{O}(F) \times \text{O} (N\text{log}N \times 1) = \text{O}(FN\text{log}N)$, equivalent to training an $F$-feature C4.5 DT.
>
> **For IFA mechanisms**, given a dataset with $F$ features and $N$ samples, we assume each feature name or feature value takes up 1 token (e.g., “Gender Female”, “Height 183.2”, each feature uses 3 tokens (including one $[\text{CLS}]$ token)), then for each sample the IFA complexity is $\text{O}(F \times 3^2)$, giving a total one of $\text{O}(NF \times 3^2)$. Notably, if introducing IFA, the computational complexity of the subsequent $L$-layer LM process becomes $\text{O}(NLF^2)$, hence giving $\text{O}(NF \times 3^2) + \text{O}(NLF^2)$ computational complexity for the whole process. But if removing IFA, feature-wise $[\text{CLS}]$ tokens can be removed and each feature uses 2 tokens, the whole process complexity is $\text{O}(NL(2F)^2) = \text{O}(4NLF^2)$. It can be seen using IFA asymptotically achieve 4 times better computational complexity when $F$ is very large, such theoretical acceleration effect supports the observation of 32% more average training time (the lower half of Table 2) after removing IFA. Besides, IFA achieves name-value matching before all features are fed into the LM, which may further accelerate convergence and alleviate training burden.
>
> ---
>
> Q3: Are there dataset types or domains the TP-BERTa faces challenges?
>
> A3: As analyzed in A1, for the TP-BERTa, there are still several challenging data scenarios:
>
> (1) In the extreme class imbalance cases, the TP-BERTa exhibits a relative performance decline like other common tabular deep models for the data insufficiency of the minority class, in which case GBDTs with specific design for those issues play an active role.
>
> (2) When facing extreme feature absence situations (a high feature missing rate), the TP-BERTa appears different degrees of performance decline upon the characteristics of missing features.
>
> (3) Besides, since our TP-BERTa relies on the semantic knowledge of LMs to transfer feature patterns from pre-training feature names to downstream ones, it implicitly requires meaningful and clear feature semantics. However, there may exist some tables with unclear feature names or values, e.g., quantum physics experiment tables containing lots of uncommon quantum descriptor feature names and medical domain substituting specific feature values with meaningless encoding to protect patient privacy. At present, the current version of TP-BERTa does not adeptly handle this situation. We will investigate this issue in future work.
>
> We will add the above data challenges into our limitations in the final version. Thank you!

---

> ### Author Response · Authors · 2023-11-20
> **To AREw (3/3, Q4 & Q5)**
>
> Q4: Can the RMT be fine-tuned or adaptively adjusted (e.g., bin size) based on the domain or the data to potentially yield better results?
>
> A4: **Will bin size of RMT affect the performance?** From the comparison experiment of different magnitude token numbers (i.e., bin size) in Section 3.5 (detailed analysis in Appendix C and Table 5), when we decrease bin size from 256 to 128, in 80 binary classification datasets, significant performance decline appears in 26 datasets, while improvement is witnessed in 12 ones. When decreasing from 256 to 32, the quantities are 40 and 13. Overall, the performance changes indicate that each dataset prefer a specific bin size, and gradually increasing bin size to 256 can benefit the most datasets. Actually, this is a similar question in NLP tasks, where different text datasets perfer different embedding size. However, to leverage pre-training benefits, a uniform embedding size is a standard setting, and it also works well.
>
> **Can I manually add some extra relative magnitude tokens in fine-tune step, even though the bin size is kept fixed (i.e, 256) in pre-training? Will manually added bin size in fine-tune step hurt performance?** Although the default TP-BERTa provides 256 pre-trained magnitude token embeddings, these tokens just represent a relative relationship of number magnitudes. Extra randomly initialized magnitude tokens in fine-tune will learn their embeddings locally, and the weights of pre-trained tokens (bins) still keep the relative relationships. It may lead to a slower convergence, but does not necessarily hurt the performance.
>
> **Adaptively adjust RMT according to data characteristics?** In theory, each dataset prefers a specific bin size. However, in our experiment, performance improvement was observed on most datasets when we increased the bin size from 32 to 256, the performance decline empirically appeared in small-scale datasets. Due to the necessity of a fixed bin size during the pre-training step, we choose a common value of 256. Actually, this is a similar question in NLP tasks, where different text datasets have different embedding size (i.e., hidden dimension size) preference. To leverage pre-training benefits, a uniform embedding size is a standard setting, and it also works well.
>
> ---
> Q5: Any visualization or analysis to assess how effectively the TP-BERTa captures the representations of numerical values?
>
> We have added the visualization of the TP-BERTa‘s 256 magnitude tokens (before and after pre-training) by directly applying t-SNE algorithm (using `scikit-learn` package and default function parameters) on their embeddings (in Appendix F “Interpretability of RMT” of the current PDF revision, page 14, Fig. 4). Interestingly, even if we have randomly placed these 256 tokens in the language space at the beginning, after pre-training they captured a clear inherent distribution, all token embeddings lie on a highly regular manifold and successfully maintain an intuitive assumption that the embedding of a numerical value should close to the embedding of a nearby one. This visualization empirically demonstrates the TP-BERTa is sensitive to the numerical value magnitudes and benefits from the captured regular relationship among the relative magnitudes, which interprets its significant progress on the existing LM-based tabular models (e.g., directly treating values as raw strings). We hope the design philosophy of the TP-BERTa can stimulate broad reflection in the field of LM adaption to tabular data, and open up the mind for LLM enhancements.
>
> ---
> Reference
>
> `[1]` Song, Weiping, et al. "Autoint: Automatic feature interaction learning via self-attentive neural networks." CIKM. 2019.
>
> `[2]` Prokhorenkova, Liudmila, et al. "CatBoost: unbiased boosting with categorical features." NeurIPS. 2018.

---

> ### Comment · Reviewer_AREw · 2023-12-01
>
> Thank the authors for the detailed response which well addressed my main concerns! I raised my score accordingly.

---

### Meta-Review · Area_Chair_MKpK · 2023-12-15

**Metareview:**

The paper introduces TP-BERTa, a novel language model designed for tabular data prediction, incorporating relative magnitude tokenization and intra-feature attention mechanisms. The general consensus across reviews is positive. The reviewers appreciate the paper's soundness, presentation, novelty, and contribution. The proposed model demonstrates strong performance in tabular data prediction, outperforming conventional models and showing competitiveness with GBDTs. The presentation is also clear. Areas for improvement include further exploration into limitations and comparisons to previous approaches, as well as investigations into the impact of (pre-training) data diversity.

**Justification For Why Not Higher Score:**

The scope does not seem to be broad enough to garner sufficient attention from a wider audience.

**Justification For Why Not Lower Score:**

Good paper that seems to be worth highlighting at the conference.

---

### Decision · Program_Chairs · 2024-01-16

Accept (spotlight)